# Direct dating of overprinting fluid systems in the Martabe epithermal gold deposit using highly retentive alunite

Jack Muston[1], Marnie Forster[2], Davood Vasegh[2], Conrad Alderton[3], Shawn Crispin[4], Gordon Lister[5]

[1] AngloGold Ashanti, Perth, 2601 Australia

[2] Argon Geochronology and Structural Geology, Research School of Earth Sciences, Australian National University, Canberra, 2601 Australia

[3] C3 Metals, Toronto, Canada

[4] Eurasian Resources Group, Dubai

[5] W.H. Bryan Mining and Geology Research Centre, Sustainable Minerals Institute, The University of Queensland, Brisbane 4068, Australia

*Correspondence to*: Jack Muston <jemuston@gmail.com> and Gordon Lister <g.lister@uq.edu.au>

**Abstract.** The Martabe gold deposits in Sumatra formed in a shallow crustal epithermal environment associated with intermediate mafic intrusions adjacent to an active right-lateral wrench system. Gas/fluid temperatures reached 200-350°C. The structural geology suggests episodic switches in stress orientations during a Plio-Pleistocene seismotectonic evolution. Different mineralisation events may have been associated with oscillations in this earthquake cycle, so samples containing alunite were collected for $^{40}Ar/^{39}Ar$ geochronology to constrain the timing. $^{39}Ar$ diffusion experiments were performed to constrain variation in argon retentivity. The age spectra were produced by incremental step-heating with heating times chosen so similar percentages of $^{39}Ar$ gas release occurred during as many steps as possible. This ensured the detail necessary for analysis of the complex morphology, of these spectra, applying the method of asymptotes and limits, which enabled recognition of different growth events of alunite in overprinting fluid systems. It was possible to provide estimates as to the frequency of individual events and their duration. The heating schedule also ensured that Arrhenius data populated the inverse temperature axis with sufficient detail to allow modelling. Activation energies were between 370 – 660 kJ/mol. Application of Dodson's recursion determined closure temperatures that range from 400 – 560°C for a cooling rate of 100°C/Ma. Such estimates are higher than any temperature to be expected in the natural system, giving confidence that the ages represent the timing of growth during periods of active fluid movement and alteration: a hypothesis confirmed by modelling age spectra using the *MacArgon* program. We conclude that gold in the Purnama pit resulted from overprinting fluid rock interactions during very short mineralisation episodes at ~ 2.25 and ~2.00 Ma.

## 1 Introduction

Gold production on Sumatra significantly increased due to the successful development of the Martabe gold mine, which began production in 2012. The ore system was first discovered in 1997 after a positive result from geochemical surveys conducted on stream sediment within an exploration tenement. This was soon followed by field mapping, rock-chip sampling, and aeromagnetic surveying, which led to the discovery of six deposits within a 7 km by 3 km corridor. The largest of these deposits is Purnama, with

a resource estimate of 4.3 million ounces of gold and 53 million ounces of silver (open file report Agincourt Resources, 2018).

The Purnama deposit near Martabe (Fig. 1) occurs near the intersection of several major structural features (Muston, 2020). These include: i) the Martabe Lineament; ii) the inferred trajectory of the boundary of the rupture segments between the 2004 and 2005 Great Earthquakes; iii) the inferred trajectory of the now partially subducted Wharton Ridge, which separates the Indian and Australian plates; and iv) the right-lateral Sumatran wrench system, a 1600 km long wrench system that runs the entire length of Sumatra (Levet et al., 2003). Other authors have noted the coincidence with the extrapolation of fracture zones such as the Investigator Ridge (e.g., Garwin et al., 2005; Maryona et al., 2014) but these fracture zones link to former transform faults in the subducted part of the Wharton Ridge, and the subducted transforms are still seismically active. The Wharton Ridge is of particular interest because multiple spreading centres were offset by long transform faults. Continual movement on these structures (in conjunction with active volcanism) would have generated fluid pathways, and thus contributed towards the localisation and enrichment of gold deposits in a host rock characterised by intermediate flow domes (hornblende-bearing andesites) and maar-diatreme breccias.

The Sumatran wrench system formed in response to the oblique subduction of the locked Indian and Australian plates (Barber and Crow, 2005), but it is atypical: i) since it was formed during rollback of an adjacent subduction zone; and ii) it was occasionally overwhelmed by the effects of differential rollback of the slab adjacent to different rupture segments of the attendant megathrust. Such relative movement may have driven the development of offset structures (Fig. 1) that cut the Sumatran Fault, preventing strike-slip motion, locking it, and thus temporarily rendering it inactive. Each locking event would have driven episodic switches in stress orientations, and driven pulses of fluid activity related to changes in the movement picture as illustrated in Figure 2. Such events appear to have regularly taken place during the Plio-Pleistocene seismotectonic evolution of the Martabe district.

The cycle appears to begin with differential rollback of the subducting Indian Plate, evidenced by localised curvature of the subduction trench (Fig. 1). Left-lateral motion on a cross-cutting strike-slip fault accommodates this differential rollback, and feeds displacement into the offset structure (Fig. 2a). The location of the inferred strike-slip cross-fault coincides with the boundary separating the rupture segments of the 2004 and 2005 Great Earthquakes (Fig. 1). If ductile failure at depth controlled the active structures, a maximum moment yield condition would control the orientation of the plane of failure, requiring a state of stress as illustrated in Figure 2b. Deviatoric stress intensity would then have been expected to rise, since the cross-over structure would have temporarily locked the Sumatran Fault.

Such a locked state could not have lasted long because renewed motion on the Sumatran Fault then led to rupture, forming relay faults that ironed out these offsets (Fig. 2c). Since simultaneous rifting across the Sumatran Fault took place in association with these relays, this change in the movement picture required a >90° vertical axis rotation of the deviatoric stress axes (Fig. 2d). Even this stress state was transitory, for the final stress state in this sequence switched the orientation of $\sigma_3$ from horizontal to vertical, forming popup structures and thrusts within the restraining bends (Fig. 2e). These last changes in the orientation of the deviatoric stress axes (Fig. 2f) ensured that the orientation of $\sigma_1$ was what was necessary to drive continued right-lateral motion on the Sumatran Fault.

During these oscillations in the earthquake cycle, geometric incompatibilities in the relative motion of intersecting structures would have caused dilation that facilitated fluid pulses. Different episodes of fluid movement might have distinct metallogenic significance, so we set out to explore the spatial and temporal evolution of fluid systems associated with this deposit, aided by $^{40}Ar/^{39}Ar$ geochronology. The new data should allow inference as to the time involved in each mineralisation pulse and provide estimates as to the frequency with which the seismotectonic cycle repeats. The data should also allow insight into the relation between earthquakes and the formation of large epithermal gold deposits.

Extensional faults may structurally control fluid pathways for alteration and mineralisation at deposit-scale. Their orientation implies that the axis of least compressive stress ($\sigma_3$) was parallel to the Martabe Lineament, so mineralisation coincided with NW-SE stretching (Fig. 1). The Sumatran Fault System overall has at least twelve such step-overs, each with offsets greater than 1 km (Sieh and Natawidjaja, 2000), so this may be significant for generating new mineral exploration targets. However, it should be noted that the distinction of different fluid systems is essential when using geochemical data to 'vector' within the alteration systems, since such methods will fail if samples are taken from fluid conduits active at different times, and thus with independent variation of their mineralogy and microchemistry.

## 2 $^{40}Ar/^{39}Ar$ Geochronology

Epithermal gold ± copper ± silver deposits form in shallow crustal environments, at 1-3 km depth (White and Hedenquist, 1990, 1995; Simmons et al., 2005). Low-sulphidation deposits form distal to their source magmas, through mixing and transport by deep groundwater fluids. These are characterised by reduced sulphur species and $H_2S$. In contrast, high-sulphidation deposits form from magmatic fluids, proximal to their source intrusion, and are undiluted by groundwaters. In these low-pH conditions, a suite of alteration minerals is formed (e.g., dickite, alunite, kaolin-dickite, pyrophyllite). The mineral alunite $[KAl_2(SO_4)_2(OH)_6]$ is of particular interest as it is a potassium bearing mineral that has been shown to be a useful $^{40}Ar/^{39}Ar$ geochronometer for dating alteration systems (e.g., Arribas et al., 1995, 2011). It commonly forms in porphyry and epithermal gold systems when hot, highly acidic fluids interact with and alter potassium-feldspars. During such acid-sulphate alteration, a subset of advanced argillic alteration assemblages is distinguished by the formation of alunite (Rye et al., 1992).

Previous work at Martabe has dated four samples of alunite using $^{40}Ar/^{39}Ar$ geochronology and concluded that the alteration system was active over a broad time-period: between $3.30 \pm 0.2$ Ma and $2.00 \pm 0.2$ Ma (Sutopo, 2013). Our study provides an additional ten samples analysed to provide more detail as to the geometry and timing of the alteration systems. Sample locations are shown in Figure 3, and photographs of the samples in Figure 4. The most pristine alunite zones were cut out from drill core, and these small samples were crushed to 420-240 µm before alunite grains were laboriously hand-picked and separated under an optical microscope. The chosen grains had a white-cream colour, soft texture, and anhedral crystal shape with a high (9-10% by weight) $K_2O$ content. The ages were estimated to lie between 3 and 1 Ma, so 100-150 mg of alunite per sample was picked with a purity of 99%, using XRD to verify the purity and mineralogy of the chosen aliquots.

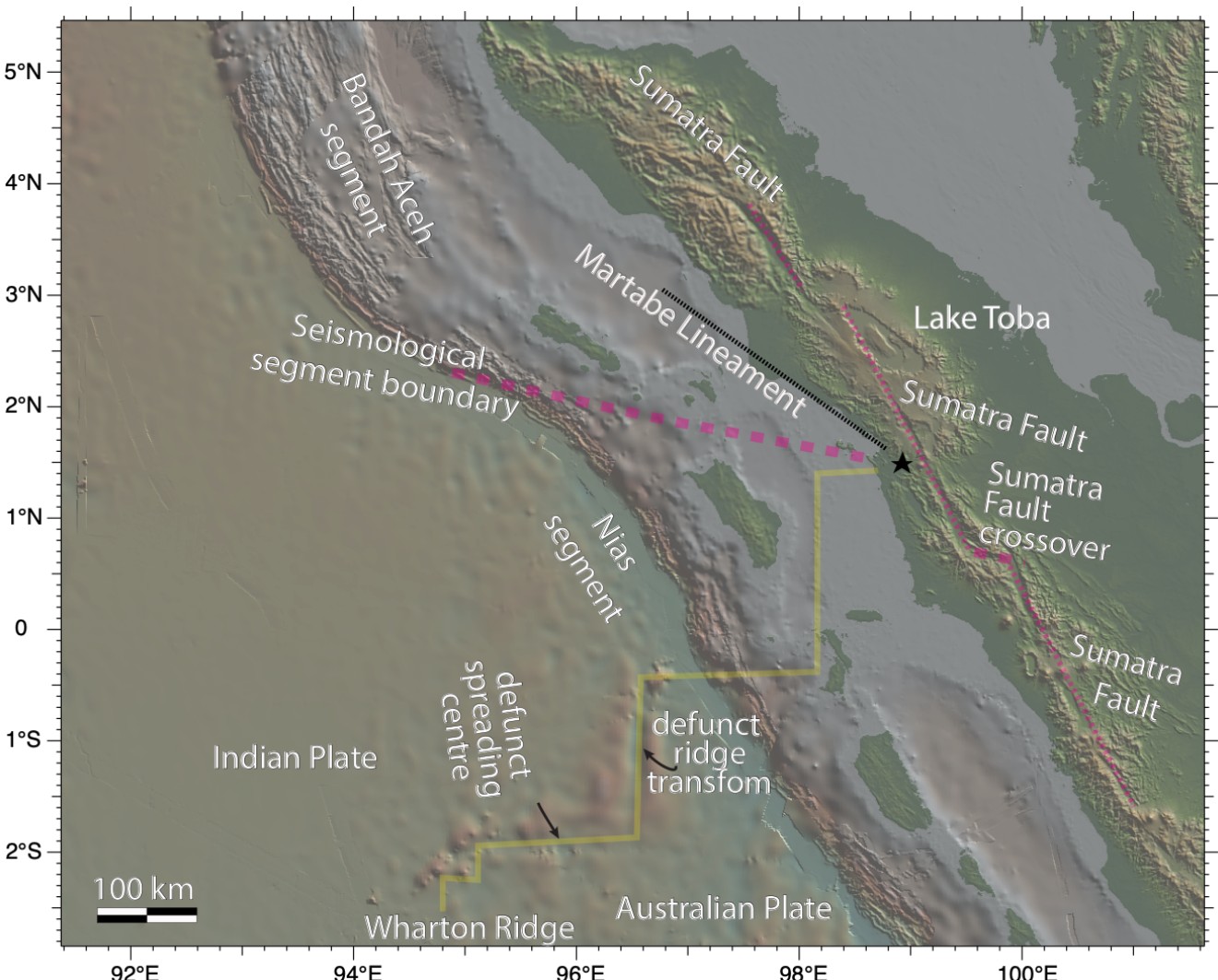

**Figure 1** The Martabe deposits (black star) formed near where the seismological segment boundary between the 2004 and 2005 $M_w$ 9.2 and $M_w$ 8.5 Great Earthquakes intersects the Sumatran Fault System. Other structures also intersect in this region: i) the Martabe lineament with limited vertical block motion; and ii) the defunct spreading ridges and linked transform faults associated with the Wharton Ridge. Swarms and clusters of hypocenters allow definition of its trajectory in 3D, even though the ridge has in part already been subducted (Muston, 2020).

120

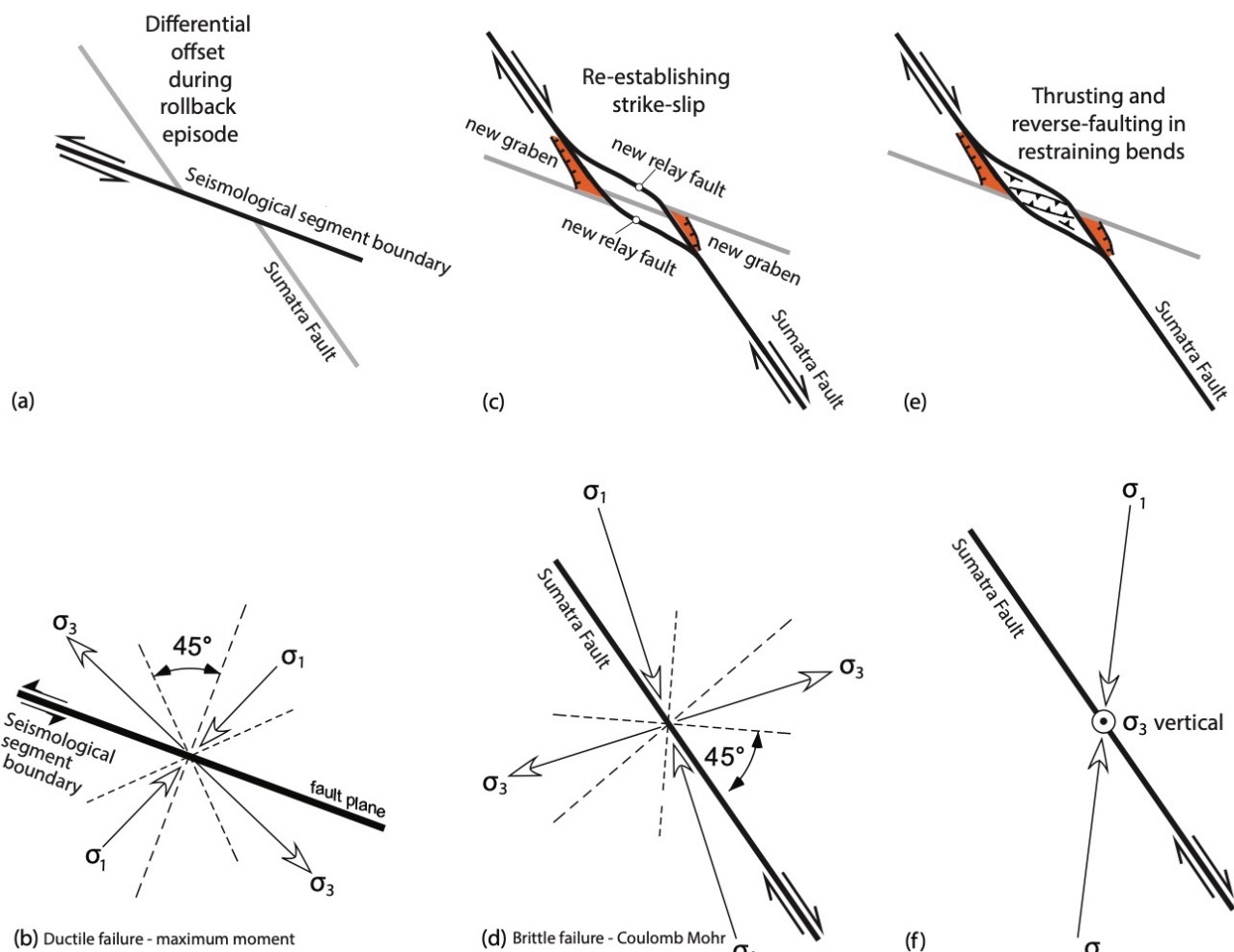

**Figure 2** The movement picture associated with the formation of the Martabe deposits appears to be linked to earthquake cycles, beginning with a period of differential rollback on the Simeulue (or Banda Aceh) segment of the subducting Indian plate (a). The stress state (b) during this time involved NW-SE stretching, with the orientation of faults as predicted by the maximum moment yield criterion. Renewed motion on the Sumatran Fault (c) drove the formation of relay structures, with dilation of the fault strands requiring stress state (d). Finally (e), at the end of this cycle, $\sigma_3$ switched to vertical (f), causing popup structures in the restraining bends, and inversion.

Details of the measurement procedures are provided in the supplementary information, but several aspects of specific interest are noted here. First, although one can rid an alunite sample of unwanted inclusions by using strong acids (e.g., $HNO_3$), such methods can modify the microstructure and thus eliminate useful information. So instead, the samples were cleaned in deionised water, and the detail in the age spectra was used to separate the effects of contamination. This was made possible by numerous (32-35) heating steps in the step-heating schedule, thus providing the necessary detail. Contaminants (probably related to inclusions) overpowered the diminishing alunite signal in the last 5-20% of gas release, however (see later discussion). Second, prior to measurement, samples were dropped into the resistance furnace and heated briefly to 400°C, thus driving off volatile contaminants in fast-diffusion pathways. The temperature was then immediately reduced, and the sample left for a minimum of 12 hours pumping away unwanted gases such as extraneous $^{40}Ar$, air and water. Long periods of cleaning under ultra-high-vacuum (UHV) conditions minimise the effect of such volatiles, especially in the first few steps where low retentivity diffusion domains also release gas. This procedure also reduces sample instability. A third important aspect of the methodology is that the mass of the sample was chosen to be large enough to allow many heating steps. This ensures detail in the age spectra and associated Arrhenius plots, and for each step, more than sufficient release of lattice argon so as to overwhelm any 'blank' caused by adsorption/retention in the furnace. The larger volume also allows 51 measurement cycles (instead of the historical 7) for each step, thereby significantly improving the precision.

Samples were analysed using a multi-collector *ThermoFisher* Argus VI (see information available at URL: http://argon.anu.edu.au/ and in the detail of the supplementary information). The samples were step-heated in a resistance furnace, each step commencing under ultra-high-vacuum (UHV) conditions, thus allowing $^{39}Ar$ diffusion experiments at the same time as $^{40}Ar/^{39}Ar$ geochronology. Correction factors were used to eliminate interference from Ca and Cl, since these elements produce argon isotopes during the irradiation process. Cross-contamination was avoided by protracted cleaning of the furnace between samples, so furnace blanks are consistently reduced to low levels with a return to atmospheric isotope ratios prior to commencing work on the next sample (see supplementary information). Only the Faraday cups were utilised for measurement, thus avoiding the drift and uncertainties associated with cross-calibration of different detector types in the Argus VI mass spectrometer.

Data from the mass spectrometer was analysed using a computer program based on the *Noble* program. This code was designed by the late Professor Ian McDougall (using methods and formulae as documented in McDougall and Harrison (1988, 1999). The *Noble* program was extensively modified by one of us (Davood Vasegh) so as to allow it to be used with the Argus VI multicollector mass spectrometer, importantly including the ability to be able to continually access and interactively interrogate data from shared memory in the otherwise inaccessible QTegra operating system.

After measurement and data reduction, data tables were produced (see supplementary information) and uploaded into the *eArgon* program, which was used for interactive analysis, facilitating: i) application of the method of asymptotes and limits (Forster and Lister, 2010; Forster et al., 2015) to analyse complex age spectra; ii) production and analysis of Arrhenius data (Fig. 5), applying and methods and formulae as set out by Crank (1975) and based on C++ computer algorithms for the analysis of different domain geometries; and iii) identification and mapping of mixing trends between different (microstructural) gas reservoirs using Turner three component inverse $^{40}Ar$ isotope ratio correlation diagrams (e.g., Fig. 6).

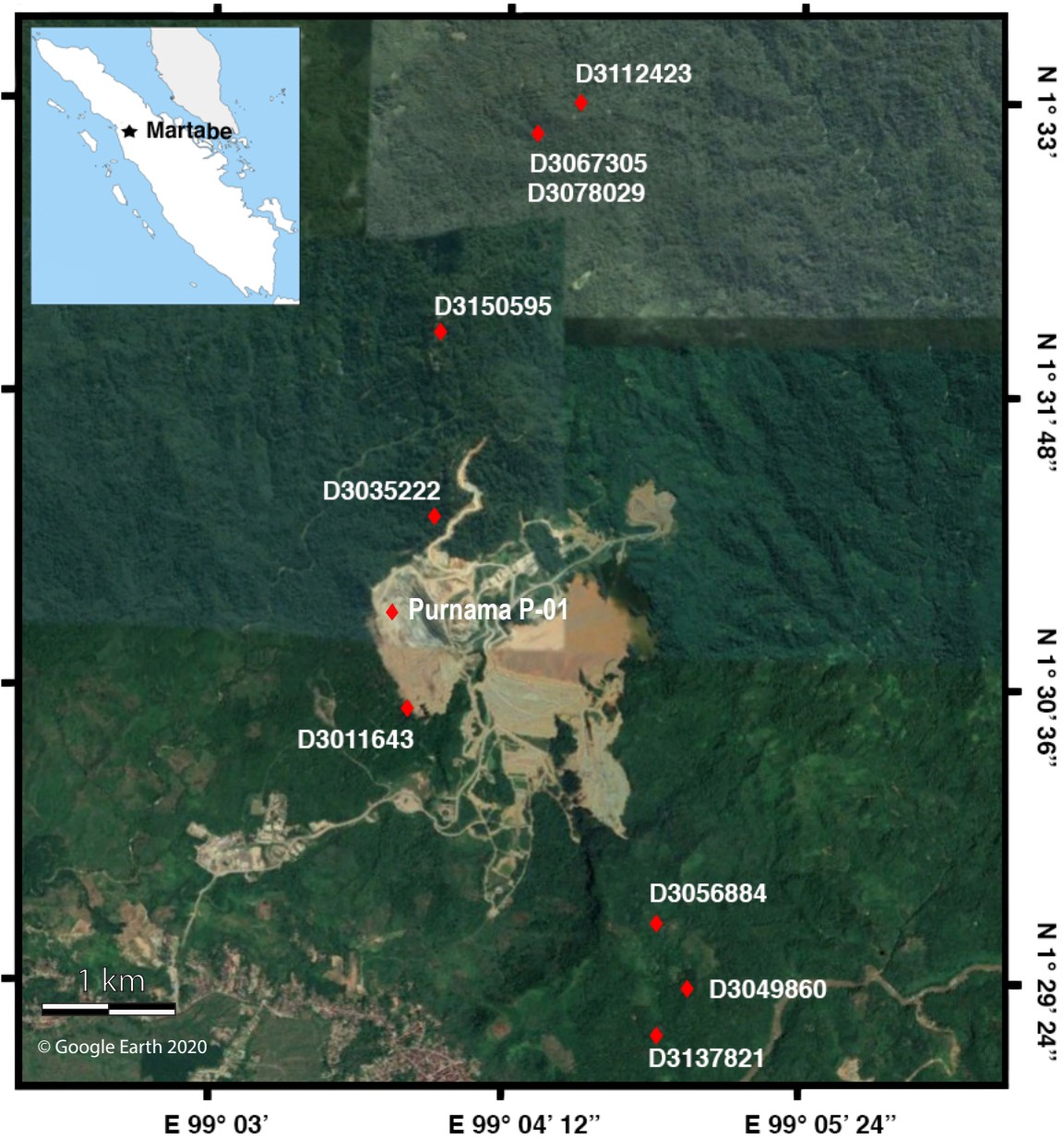

**Figure 3** The Martabe mine, Sumatra, Indonesia with location of the ten alunite samples shown by red diamonds. Two samples were from the same location. The P-01 sample from within the Purnama pit was not precisely located.

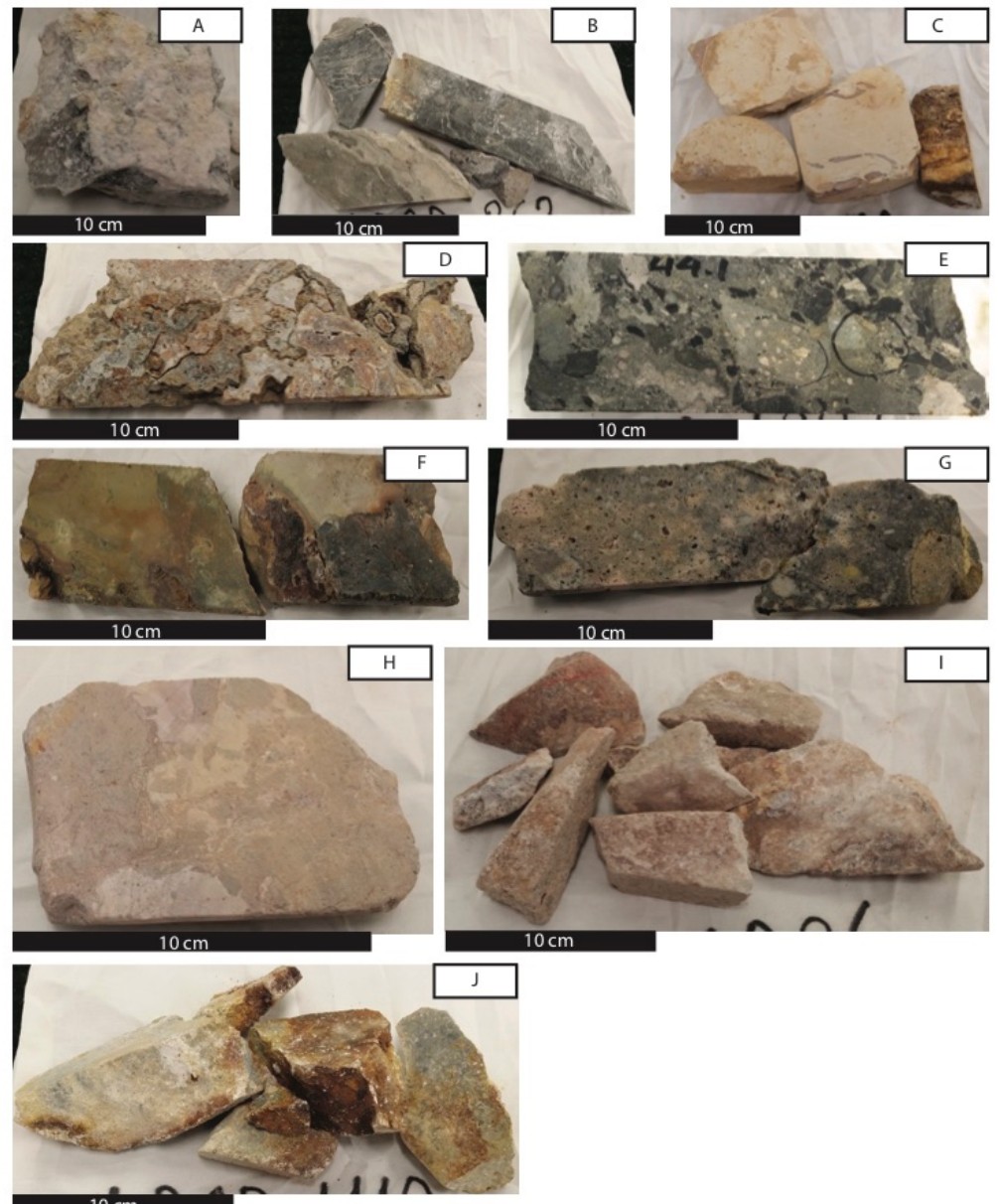

**Figure 4** (A) PURNAMA P-01. Fine grained porphyritic volcanic andesite with pervasive alunite-silica alteration and disseminated haematite. (B) D3011643. Contact between sediment and quartz vein. (C) D3150595. Fine-grained alunite in clay with reddish-brown banding. (D) D3112423. Phreatomagmatic breccia, altered by alunite-dickite-silica. (E) D3078029. Phreatomagmatic breccia, altered by alunite-dickite-silica (high grade ore). (F) D3056884. Crackle sandstone, matrix filled by alunite ± dickite. (G) D3067305. Phreatomagmatic breccia, altered by alunite-dickite-silica. (H) D3137821. Massive sandstone, oxide staining, pervasive alunite-clay alteration. (I) D3035222. Sandstone with alunite vein. (J) D3049860. Phreatomagmatic breccia, altered by alunite-silica. The purity of alunite separated from these samples was confirmed using XRD analysis.

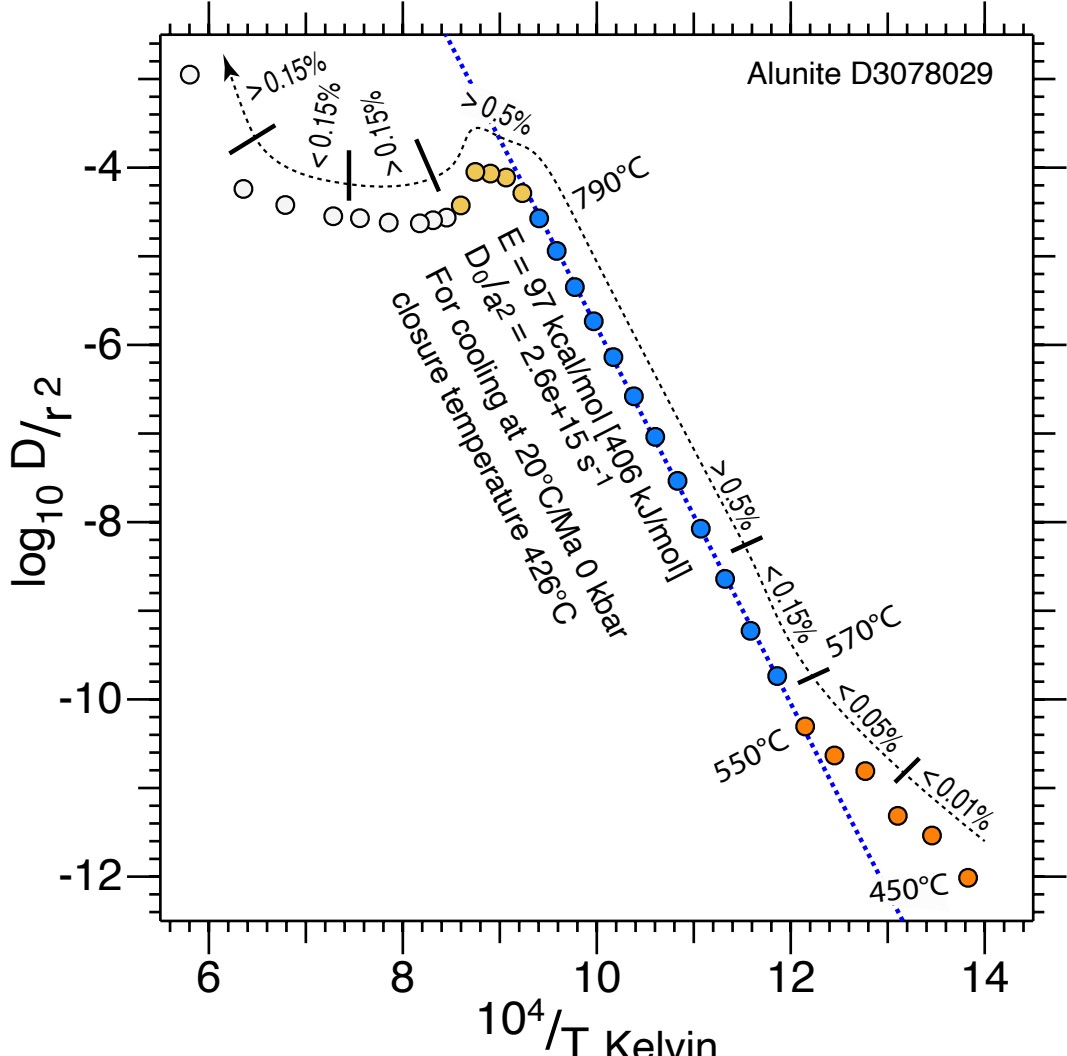

**Figure 5** An extended Arrhenius Plot, showing the application of the Fundamental Asymmetry Principle (Forster and Lister, 2010) in estimating the activation energy and normalised frequency factor for a single diffusion domain. The Arrhenius plot shows measurements with a low initial slope (as also noted by Ren and Vasconcelos, 2019) but in comparison, the change to a higher slope occurs at temperatures 80-100°C higher in our experiments. This implies a difference in material response, either due to intrinsic variation in the physical properties of these alunite separates, or to the effect of extrinsic variables, e.g., the use of strong acids in sample preparation, thinner cadmium shielding leading to more recoil and lattice damage, and/or the effect of the electron beam during an *in situ* experiment. Nevertheless, the initial change in slope is the result of mixing. An increasing proportion of lattice argon is being released as temperature climbs, while a low-volume non-lattice $^{39}Ar$ reservoir is progressively exhausted. Thereafter the same diffusion parameters apply over a considerable temperature range.

Data tables are included in the supplementary information. Decay constants used are those from Steiger and Jäger (1977). All constants are listed in the supplementary information, consistent with requirements of Renne et al. (2009) and Schaen et al. (2021). The sample aliquots measured include multiple alunite grains.

## 2.1 Ultra-high-vacuum (UHV) diffusion experiments

Arrhenius diagrams (e.g., Fig. 5) plot the logarithm of the normalised diffusivity against the inverse of the absolute temperature. We note that we were able to use each heating step as a [39]Ar diffusion experiment because the percentage release of [39]Ar overall allows back calculation of the percentage release of [39]Ar in each step. The normalised diffusivity could then be calculated by applying the analytical equations that link the percentage of [39]Ar release to normalised diffusivity, but first assuming a particular diffusion geometry. The peak temperature attained was accurately determined by using a thermocouple in direct contact with the bottom of the tantalum crucible. Optical calibration of temperature was achieved by using an array of seven metals.

The necessary code for these calculations was documented in their supplementary data by Forster and Lister (2014). The temperature-time curve should approximate a square wave to allow their application, so the resistance furnace must therefore heat and cool as rapidly as is practically possible. Nevertheless, there are significant deviations from a square wave (see supplementary information). Although errors due to this effect are minimised by ensuring that the step-heating sequence involves monotonic rise in temperature, more [39]Ar will be released in each step than would be the case if the temperature rose and fell more steeply. This means diffusivity is always overestimated.

## 2.2 Turner inverse [40]Ar isotope correlation plots

Three-isotope correlation diagrams are widely used in isotope geochemistry. Diagrams plotting [40]Ar/[36]Ar *versus* [39]Ar/[36]Ar were first used by Merrihue and Turner (1966). Turner (1971) later introduced the [36]Ar/[40]Ar *versus* [39]Ar/[40]Ar plot. These later became commonly known as 'normal' and 'inverse' isochron plots. York (1969) developed a robust linear regression method that accounts for errors in both axes, and error correlations, commonly used in association with such plots.

Although a major use of such diagrams is the application of the York (1969) regression statistic, and the identification of 'inverse' isochrons, their main use in this paper is in revealing the pattern of mixing (e.g., Fig. 6). The step-heating schedule was chosen so that only small amounts of gas are released in each step, so the progression of the variation in the isotopic ratios from one step to the next allows ready identification of different gas populations, and the mixing trends between them. Mixing trends are particularly evident in the inverse [40]Ar ratio plots because different reservoirs begin to release gas at different stages during the step-heating experiment. The inverse [40]Ar ratio plots also reveal trends as individual reservoirs are overwhelmed, or diminish, or as one reservoir takes over from another.

The [36]Ar/[40]Ar and [40]Ar/[39]Ar ratios that have been plotted have first been adjusted to remove the effects of isotopic interferences.

## 2.3 Age spectra

The age spectrum is produced by correcting the $^{36}Ar/^{40}Ar$ and $^{40}Ar/^{39}Ar$ ratios (including by the removal of the effects of isotopic interferences), and then linking the assumed atmospheric $^{36}Ar/^{40}Ar$ ratio to the corrected $^{40}Ar/^{39}Ar$ ratio. The intersection of this line with the x-axis of a Turner inverse $^{40}Ar$ isotope ratio correlation plot defines the percentage of radiogenic argon ($^{40}Ar^*$), thence allowing the apparent age to be calculated. We note that there are drawbacks in using apparent ages calculated this way: i) if

there is a significant fraction of excess argon, *i.e.,* $^{40}Ar$ included in the measurement that neither originates from the atmosphere, nor from the release of lattice argon; or ii) if contaminants produce molecules with mass 36 so the corrected $^{36}Ar/^{40}Ar$ ratio is pushed towards (or above) the assumed atmospheric $^{36}Ar/^{40}Ar$ ratio. The calculated apparent age will then drop to zero and the inferred age uncertainty will become untenably large. Once the $^{36}Ar/^{40}Ar$ ratio has risen above atmospheric, no age

can be estimated at all. Isochrons on the age plot are computed using a weighted mean, which takes account of the variance in the age of individual steps using equations as in Mahon (1996).

## 2.4 Plots from nine successful step-heating experiments

Figures 7-15 show combined plots from nine successful step-heating experiments, each with several diagrams created using the *eArgon* program.

Diagram (a) shows an age spectrum, with plateau segments if they are present, and other asymptotes and limits recognised according to the method set out by Forster and Lister (2014). Diagram (b) for each sample is an Arrhenius plot. Distinct diffusion domains are identified, with estimated closure temperatures as shown. Diagram (c) is a comparative radius plot, showing the relative volumes of the different domains, but necessarily (in order that this calculation might be performed) assuming a

constant activation energy based on the slope of a reference line in the Arrhenius plot.

Diagram (d) shows the variation of the percentage of radiogenic argon ($^{40}Ar^*/^{40}Ar$), noting the limitation that this is computed by determining the $^{40}Ar^*/^{39}Ar$ ratio based on extrapolation from the assumed atmospheric $^{40}Ar/^{36}Ar$ ratio. The estimated percentage radiogenic argon thus drops rapidly to zero if excess mass 36 is introduced into the mass spectrometer, e.g., as the result of small amounts of

260 Cl from contaminating inclusions bypassing the getters [see discussion].

Diagram (f) for each sample shows the Cl/K ratio inferred using the pattern of $^{38}Ar$ release, while diagram (e) shows the Ca/K ratio inferred using $^{37}Ar$. The effect of inclusions is evident in the last steps, where the amount of alunite lattice argon being released is negligible, thus allowing estimates as to impurity composition. Diagram (g) for each sample is an inverse $^{40}Ar$ isotope ratio correlation plot.

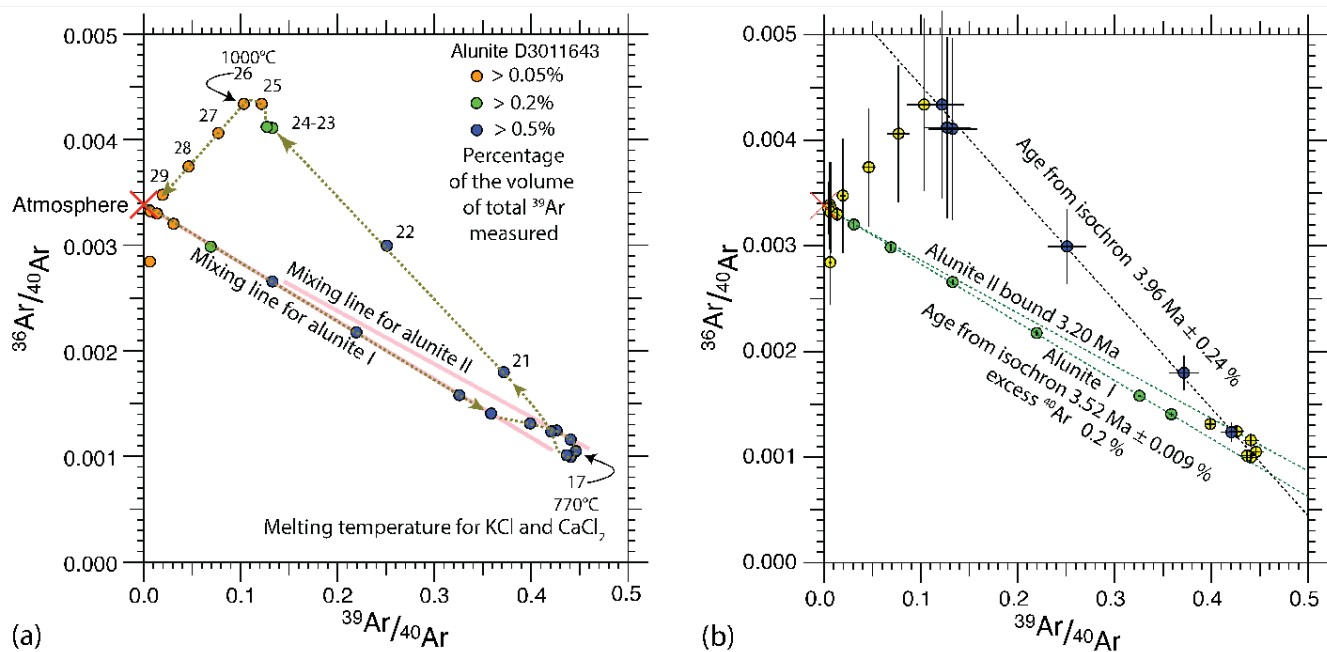

**Figure 6** Turner's inverse $^{40}$Ar isotope ratio correlation diagram, showing the mixing trend along a well-defined inverse isochron for alunite domain I, followed by mixing with alunite domain II, then a return to the mixing line for domain I, before looping off towards the inclusion reservoir. Steps are numbered to show the progression of the step-heating experiment along the mixing trends. Each sample shows elements of the trends identified here. In (b) the inverse isochrons, bounds and 2σ uncertainties in the corrected isotope ratios, estimated by recursion using the York-Mahon-Trappitsch regression and applying the associated uncertainty statistics. Notably, this sample shows how isotope ratios invariably return to atmospheric ratios by the end of each step-heating experiment.

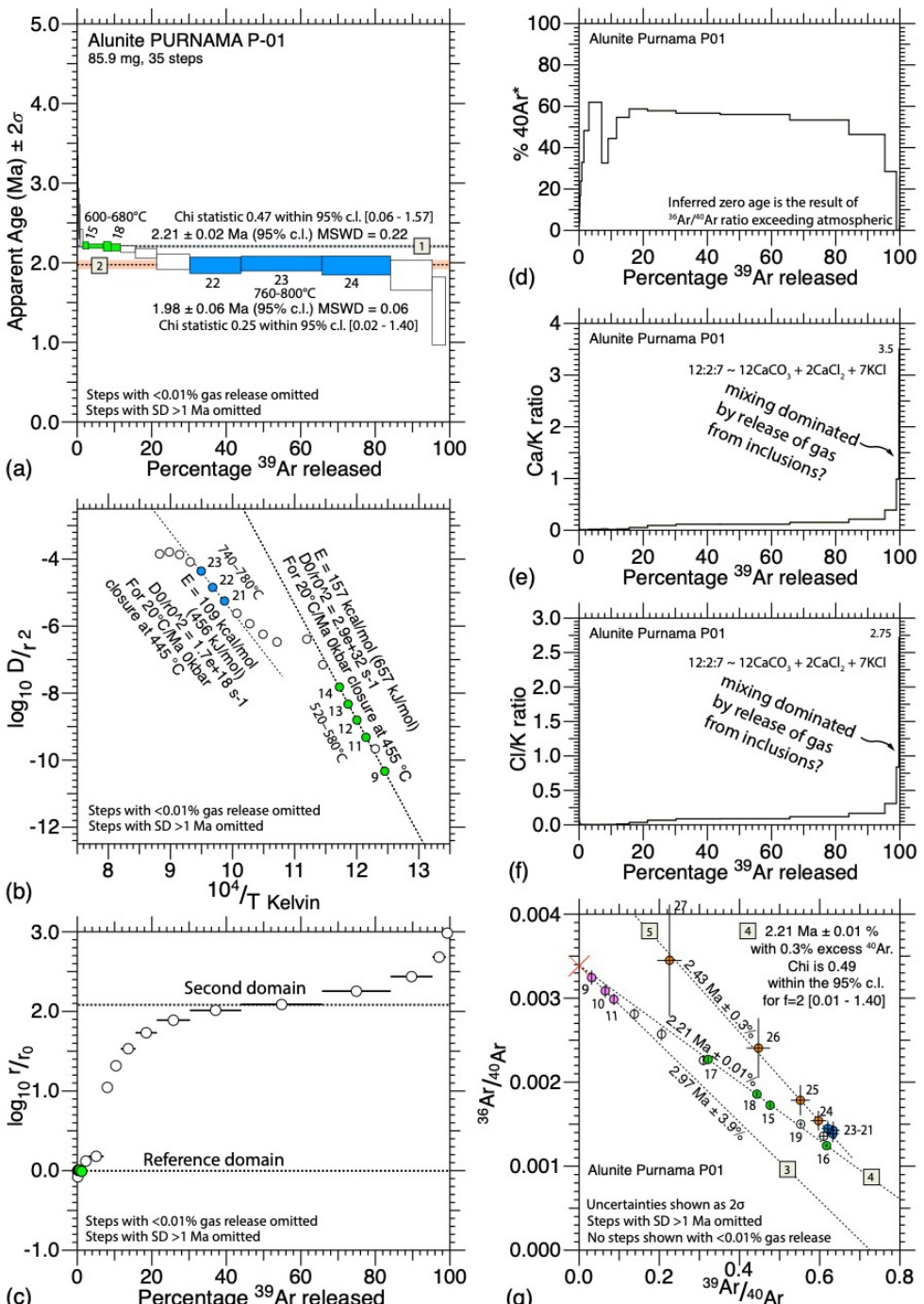

**Figure 7** Data from Purnama-P01, plotted using *eArgon*: (a) age spectrum with two plateau segments; (b) Arrhenius plot with two diffusion domains; (c) comparative radius plot, showing mixing; (d) percentage radiogenic argon drops when excess mass 36 enters the mass spectrometer; (e-f) Ca/K and Cl/K ratios show jumps with late-stage release from inclusions; and (g) Turner's inverse [40]Ar isotope ratio correlation diagram, with three inverse isochrons.

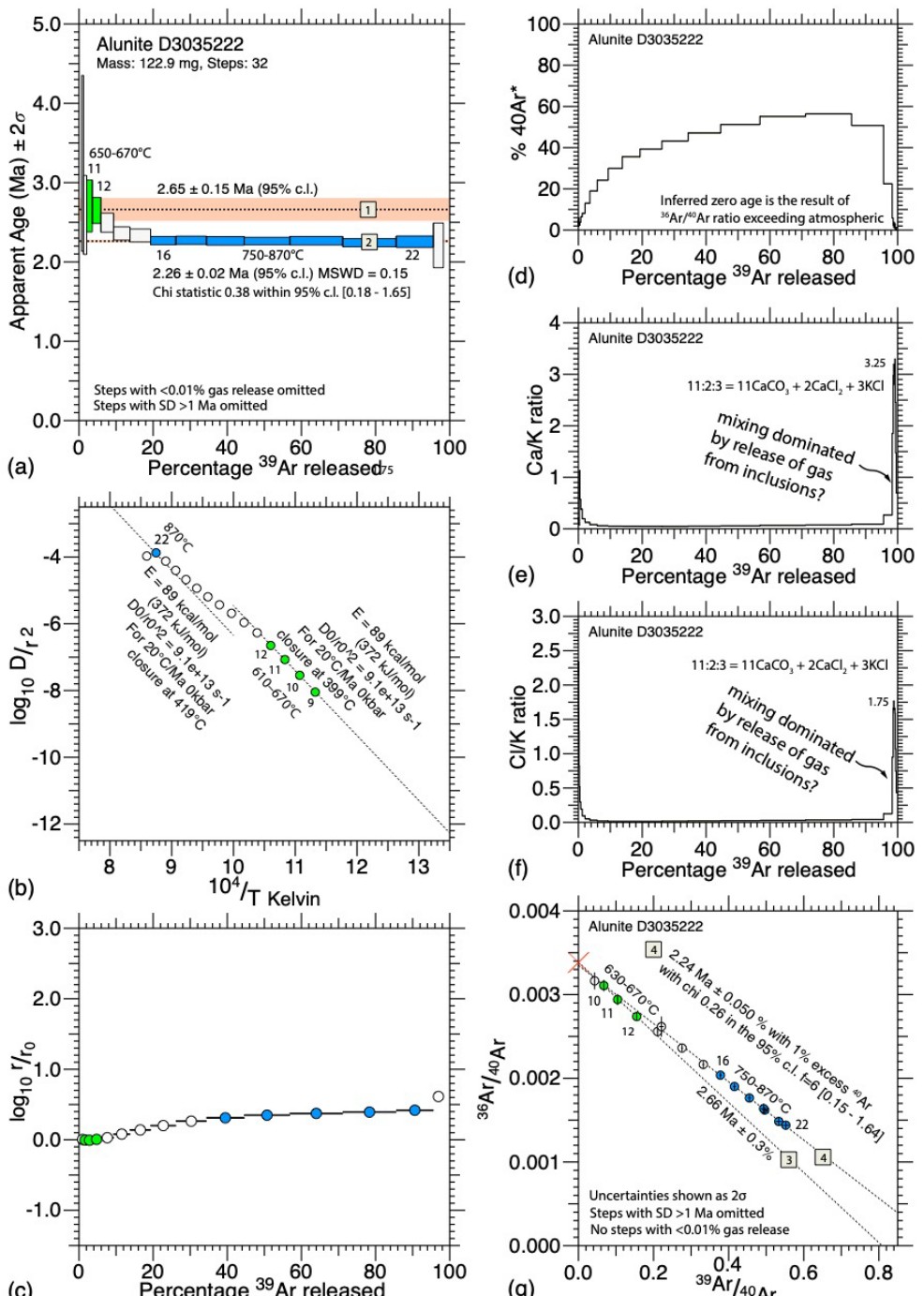

**Figure 8** Data from D3035222: (a) age spectrum with two plateau segments; (b) Arrhenius plot with two diffusion domains; (c) comparative radius plot, showing mixing; (d) percentage radiogenic argon drops at the end because excess mass 36 enters the mass spectrometer; (e-f) Ca/K and Cl/K ratios showing jumps consistent with late-stage release from inclusions; and (g) inverse $^{40}$Ar isotope ratio correlation diagram, showing two inverse isochrons.

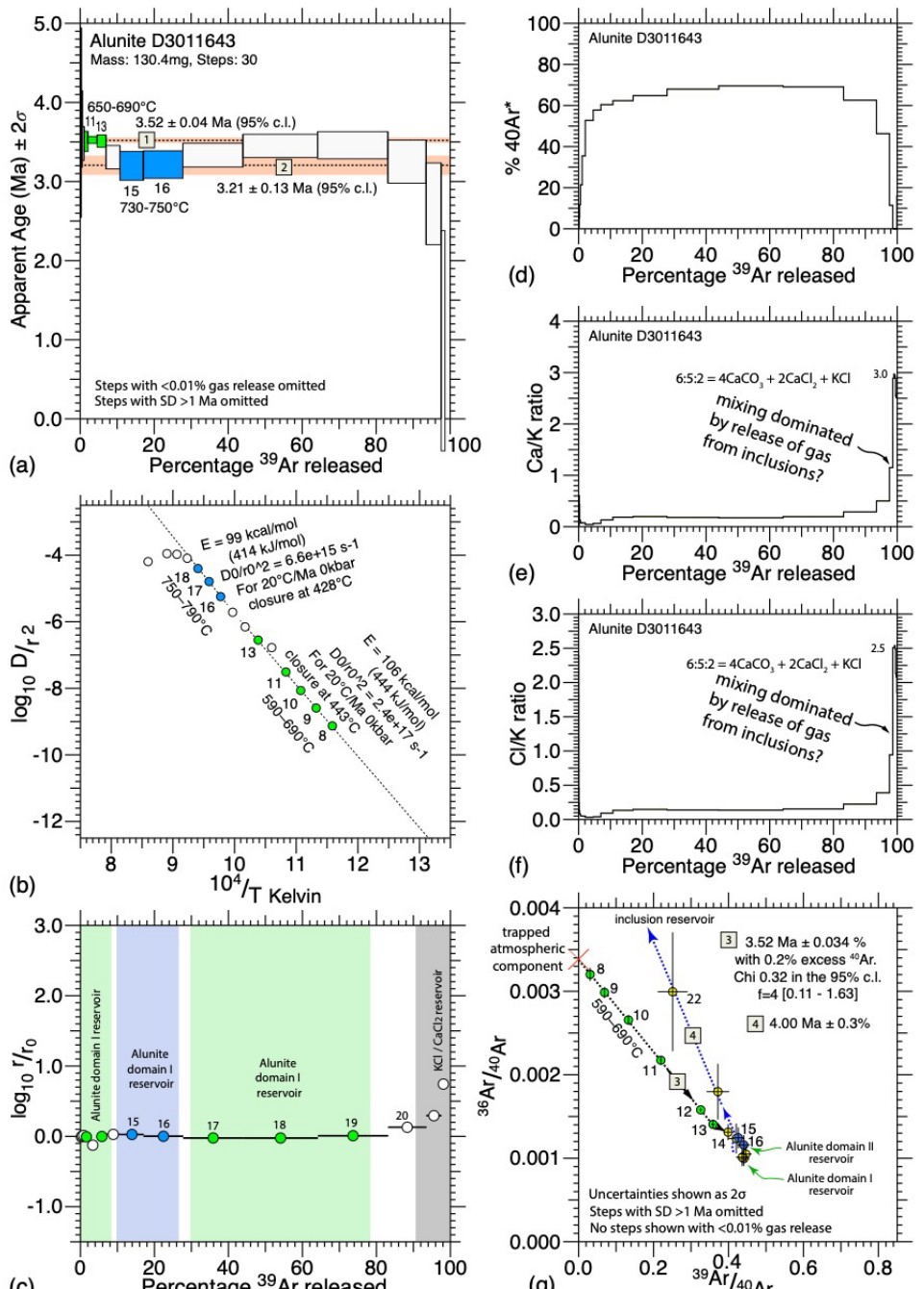

**Figure 9** Data from D3011643: (a) age spectrum starts with a plateau segment (defining domain I), drops to a lower limit (domain II), and then again mixes with domain I; (b) Arrhenius plot shows two diffusion domains, which are similar in retentivity, thus explaining the mixing curve; (c) a comparative radius plot, which shows the relative volumes; (d) radiogenic argon plot; (e-f) Ca/K and Cl/K ratios showing jumps consistent with late stage release from inclusions; and (g) inverse $^{40}$Ar isotope ratio correlation diagram, showing inverse isochron for domain I, mixing with domain II, a return to the mixing line for domain I, before looping off towards the inclusion reservoir.

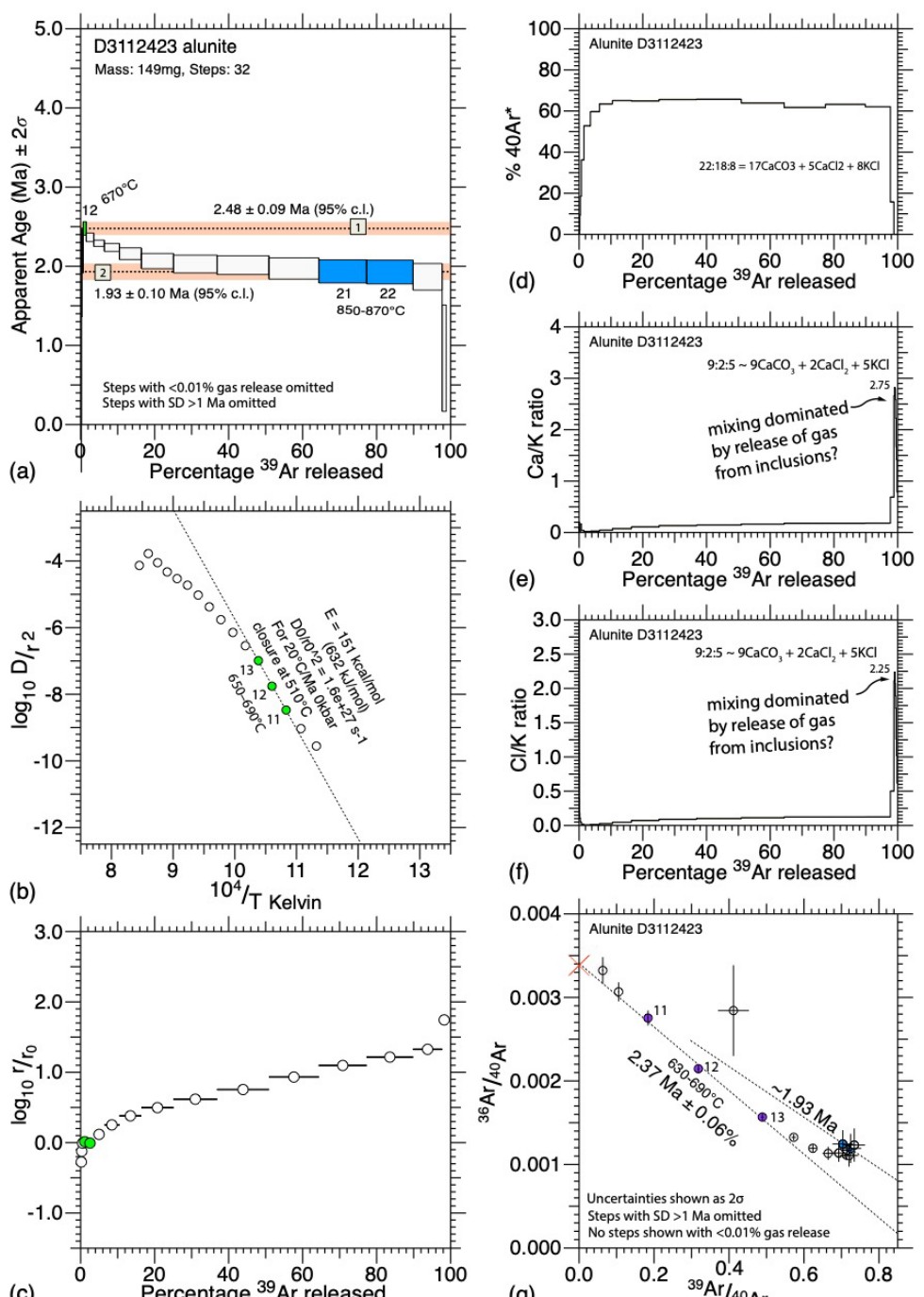

**Figure 10** Data from D3112423: (a) age spectrum with upper and lower limits; (b) Arrhenius plot allows one estimate of diffusion parameters; (c) a comparative radius plot; (d) the radiogenic argon plot; (e-f) Ca/K and Cl/K ratios showing jumps consistent with late-stage release from inclusions; and (g) inverse [40]Ar isotope ratio correlation diagram, which shows one inverse isochron, for domain I, and the mixing line for domain II.

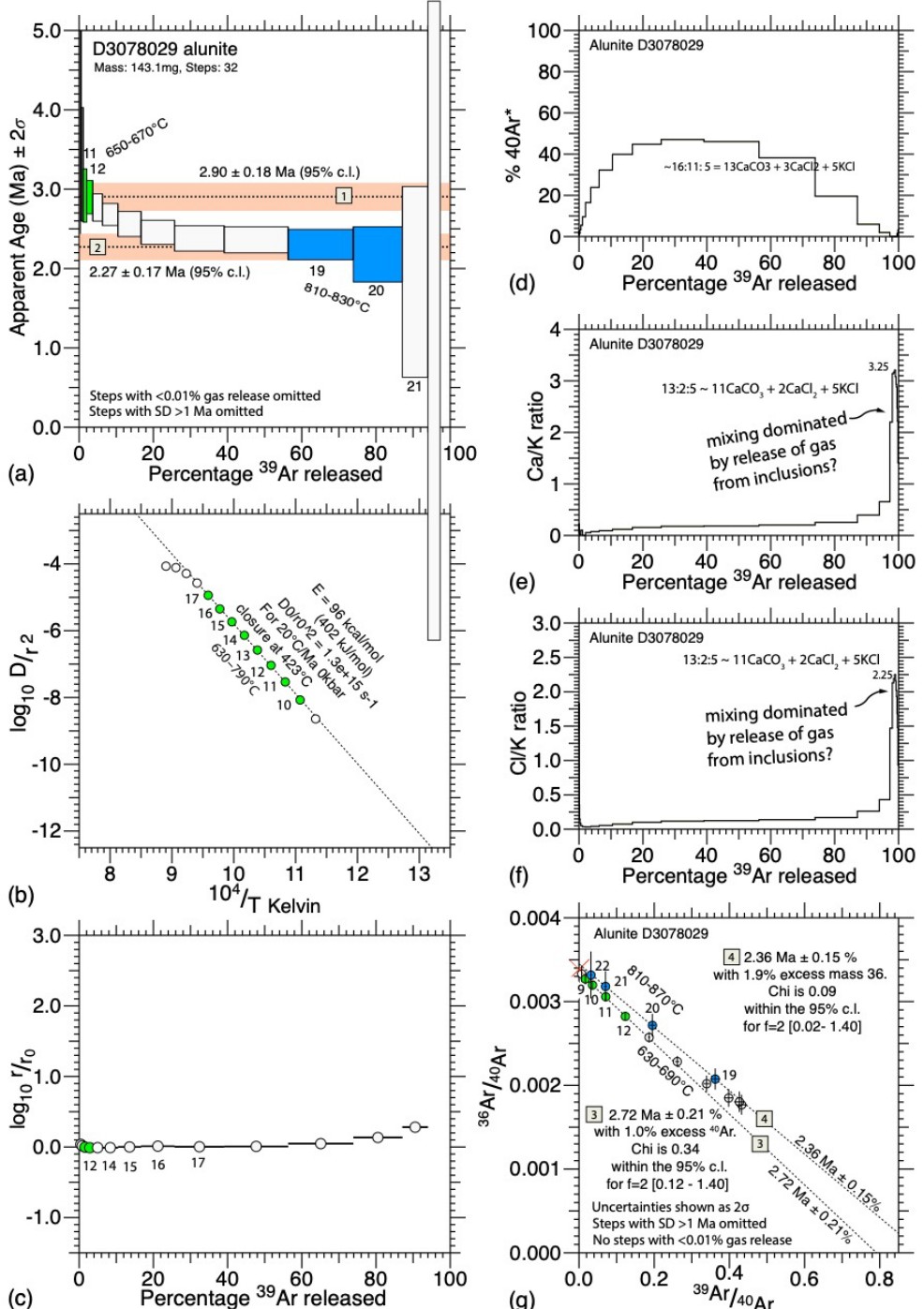

**Figure 11** Data from D3078029: (a) age spectrum with limits; (b) Arrhenius plot with one diffusion domain; (c) comparative radius; (d) radiogenic argon; (e-f) Ca/K and Cl/K ratios showing jumps consistent with late-stage release from inclusions; and (g) inverse $^{40}$Ar isotope ratio correlation diagram, with two inverse isochrons.

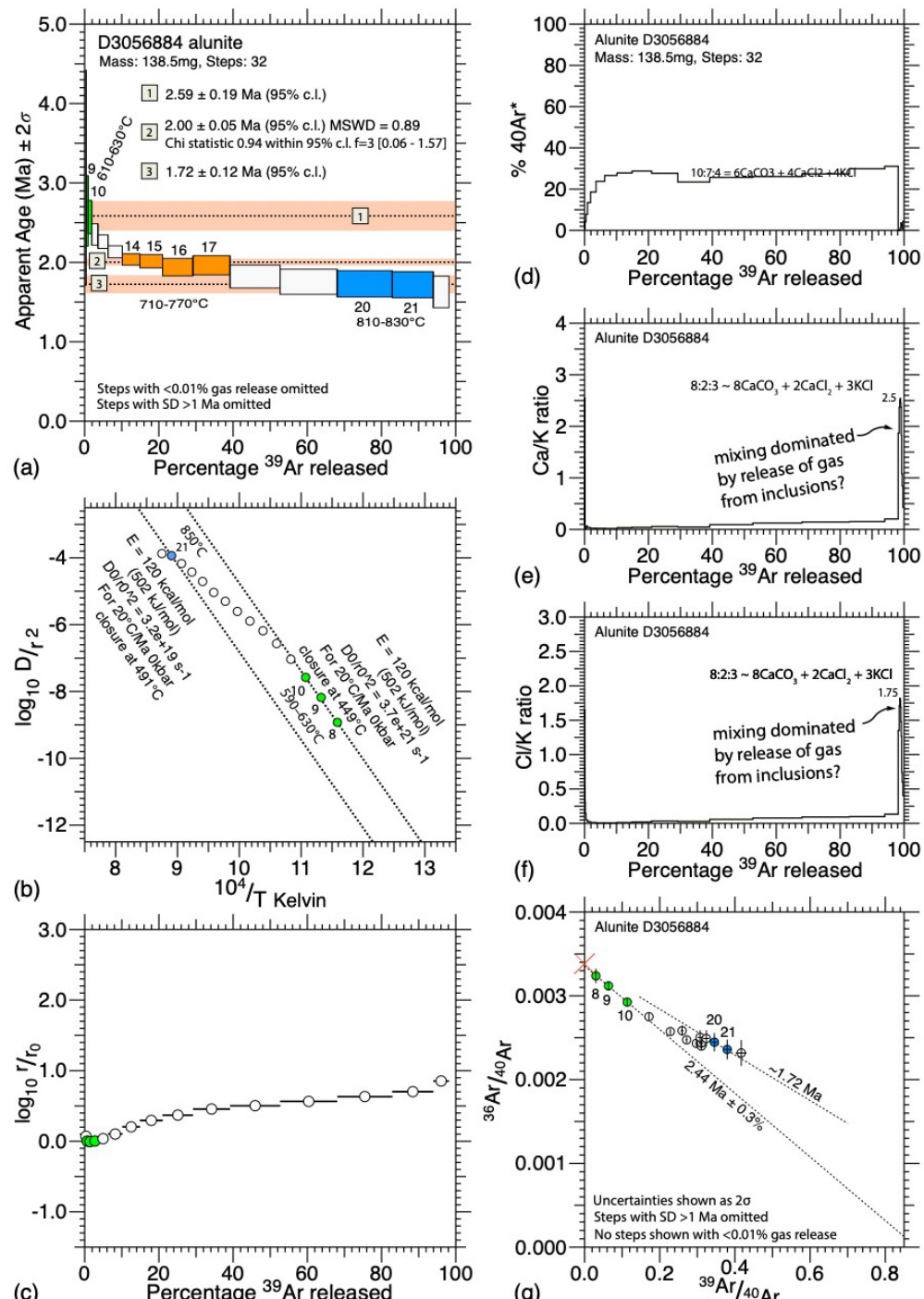

**Figure 12** Data from D3056884: (a) age spectrum with limits and a central plateau segment; (b) Arrhenius plot with one distinct diffusion domain, and then mixing over a range; (c) comparative radius plot, which shows the relative volumes; (e-f) Ca/K and Cl/K ratios showing jumps consistent with late-stage release from inclusions; and (g) inverse 40Ar isotope ratio correlation diagram, with two inverse isochrons.

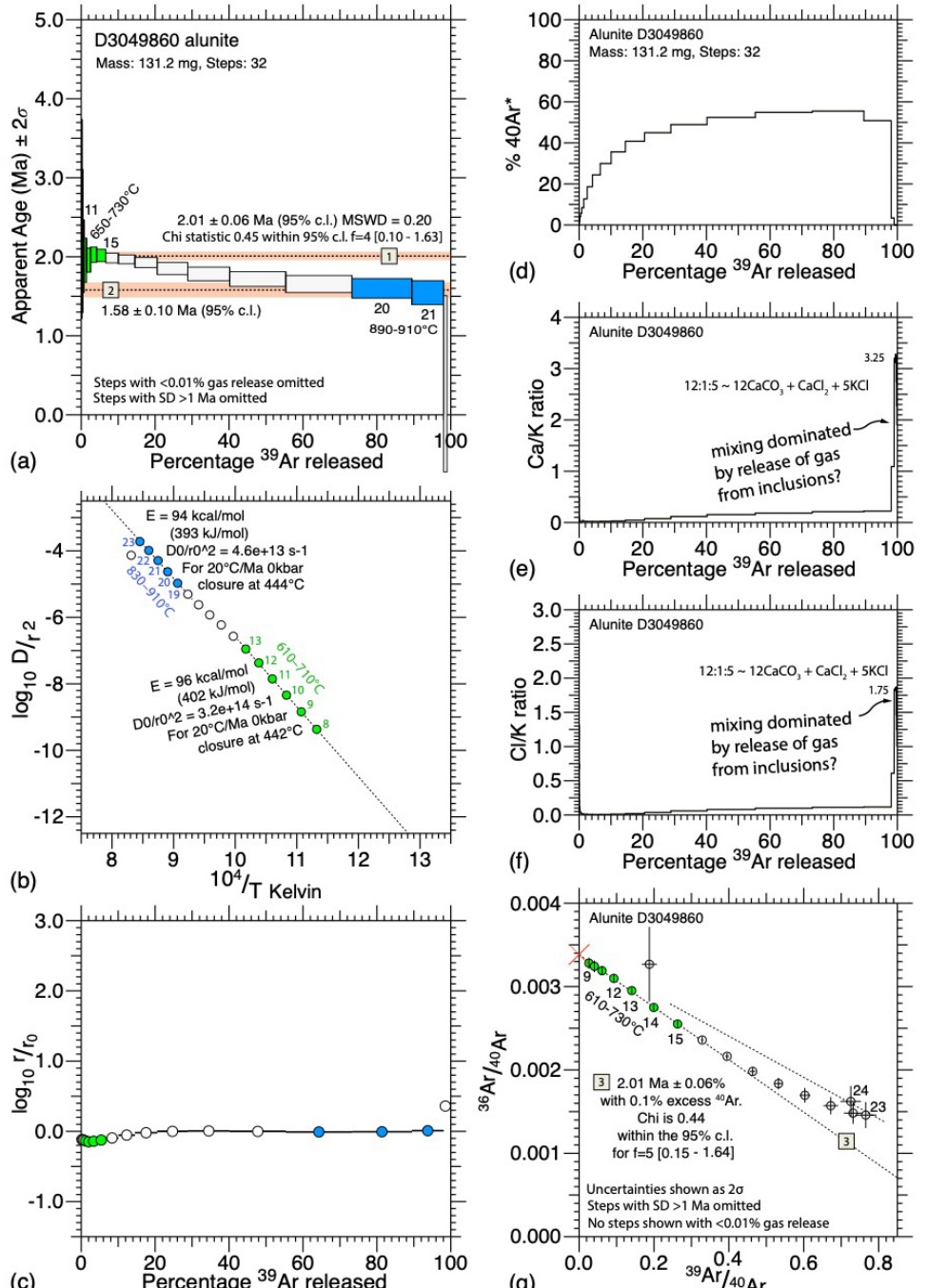

**Figure 13** Data from D3049860: (a) age spectrum with plateau segment followed by a lower asymptote; (b) Arrhenius plot, with similar diffusion domains; (c) comparative radius plot, showing relative volumes; (d) radiogenic argon plot; (e-f) Ca/K and Cl/K ratios showing late-stage release from inclusions; and (g) inverse [40]Ar isotope ratio correlation diagram, with one inverse isochron, and then mixing toward the lower asymptote.

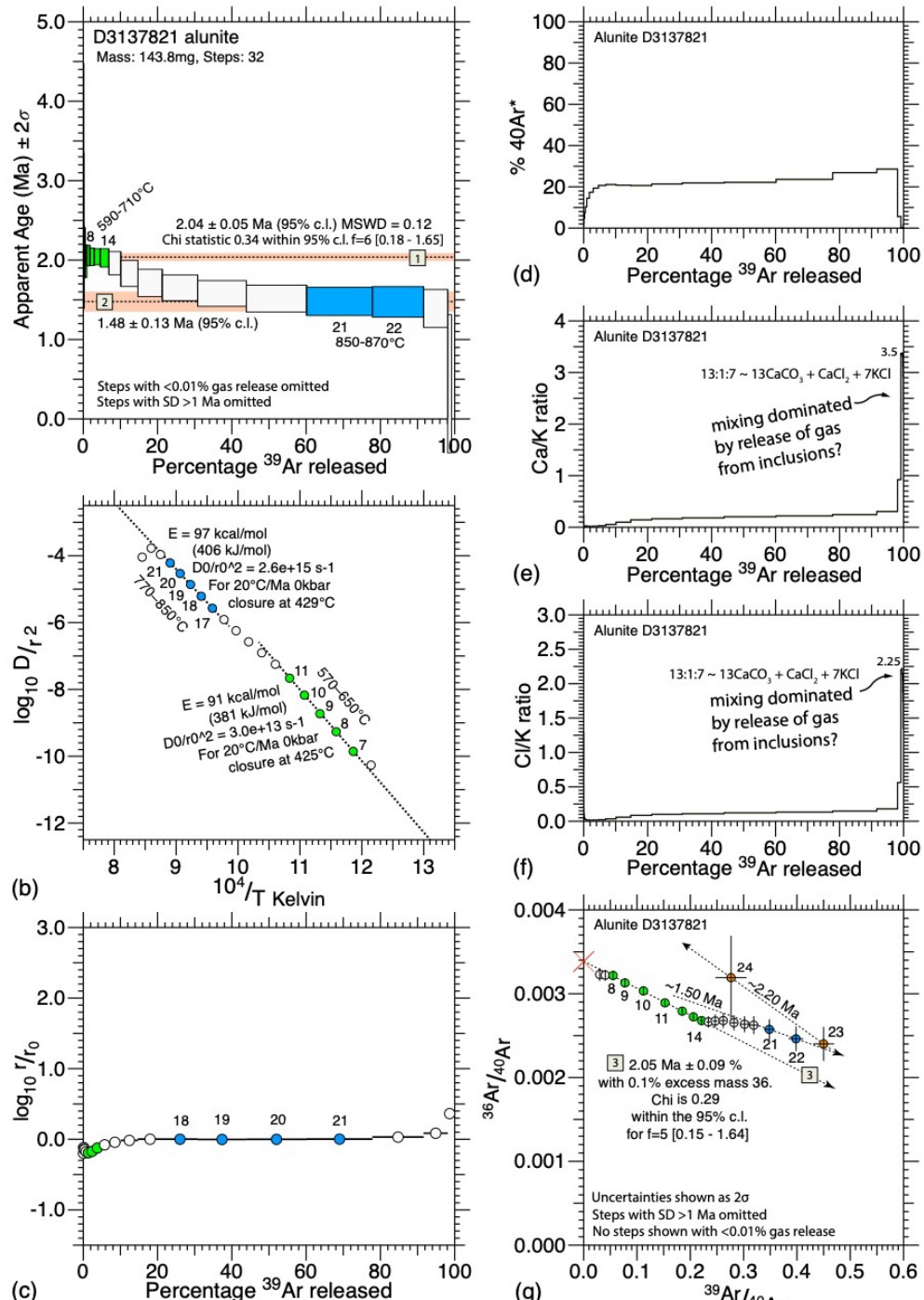

**Figure 14** Data from D3137821: (a) age spectrum with minor plateau segment and lower asymptote; (b) Arrhenius plot, with two domains; (c) comparative radius plot; (d) the radiogenic argon plot; (e-f) Ca/K and Cl/K ratios showing jumps consistent with late-stage release from inclusions; and (g) inverse 40Ar isotope ratio correlation diagram, which shows two inverse isochrons, one not apparent in the age plot.

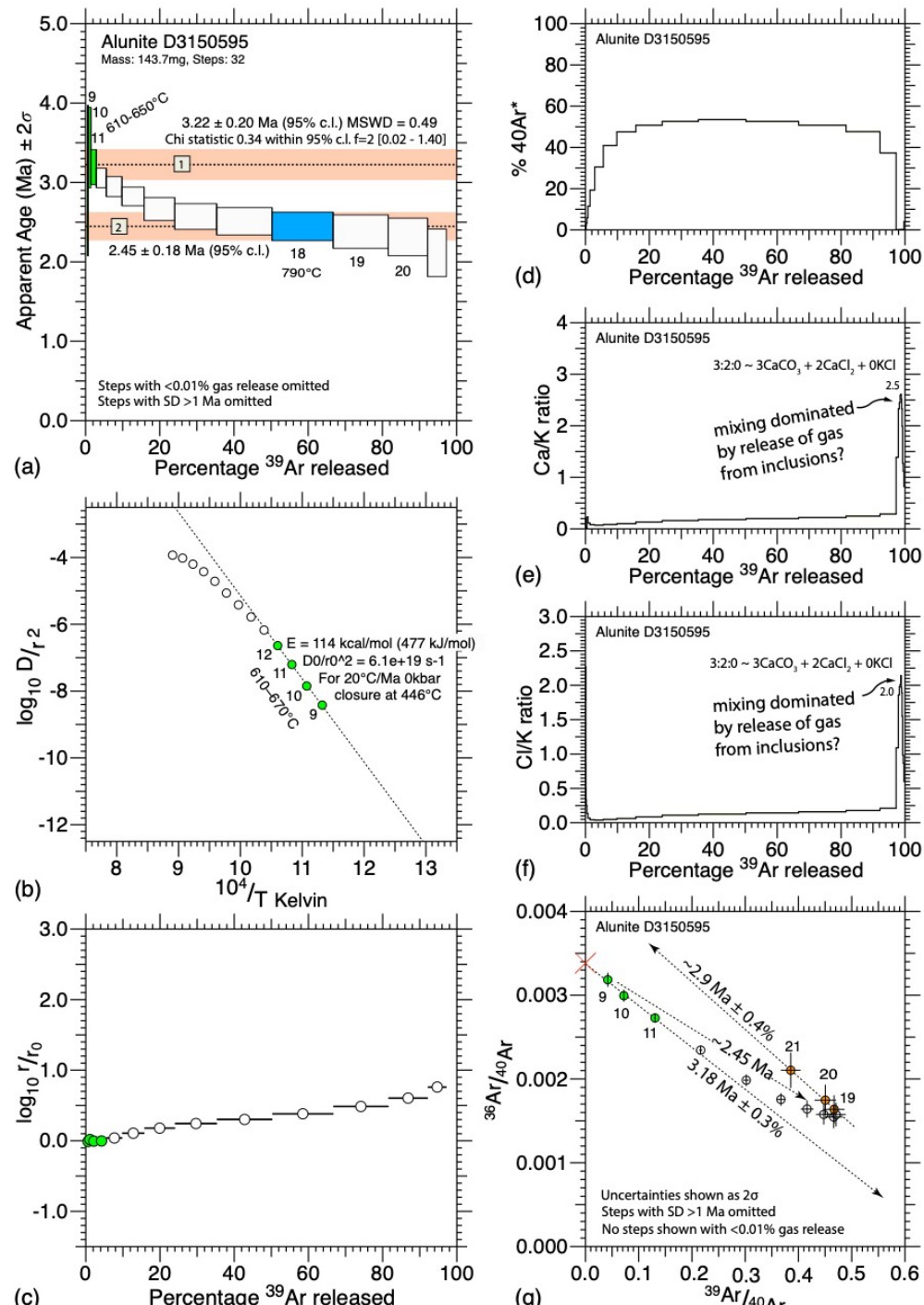

**Figure 15** Data from D3150595: (a) age spectrum with asymptote and limit; (b) Arrhenius plot, with one diffusion domain; (c) comparative radius plot; (d) radiogenic argon plot; (e-f) Ca/K and Cl/K ratios with late-stage release from inclusions; and (g) inverse $^{40}$Ar isotope ratio correlation diagram, with one inverse isochron, then mixing toward the limit shown in the age plot, and finally in the last few percent of gas release, the effect of the inclusions.

## 3 Application of the method of asymptotes and limits

The method of asymptotes and limits (Forster and Lister, 2004) was developed to allow analysis of the morphology of a collection of apparent age spectra. In essence the method advocates analysis of an age spectrum as though it was a mixture released from multiple gas reservoirs. A collection of ages can thereby be defined: representing asymptotes and limits, as well as plateau segments. Each age thereby determined can be considered as a successful result from a Bernouilli trial, with a method of recording the counting statistic then needing to be developed. The results are shown in Table I, considering only the ages from the plateau segments, asymptotes, and limits in Figures 7-15.

One way to recognise Frequently Measured Ages (FMAs) in the collection of age spectra is a counting method that uses a histogram, with individual cells incremented each time an age falls within its purview. Alternatively, as done in *eArgon*, a point may be added to a cumulative probability diagram, with a Gaussian scatter determined by the accuracy of the original measurement, or by some other (arbitrarily determined) value. Each estimate is weighted equally, since the method does not set out to determine the volume fractions that contribute to each estimate. The same uncertainty can be applied to each estimate as was recorded in the age spectrum (Fig. 16a). Alternatively, the individual Gaussian peaks can be summed with a standard deviation for each estimate equal to $2\sigma$, *i.e.*, using the values listed in Table I. Another option is that (as in Figs. 16c and 16d) the same standard deviation is forced onto each individual Gaussian peak: in (c) equal to 0.075 Ma; and in (d) 0.1 Ma. Figure 16c appears optimal for only two estimates had a greater scatter, while the remainder would otherwise scatter considerably less than the value forced upon them. This allows the effect of the precision of individual estimates to be separated from the FMA counting statistic, since in a Bernouilli trial what is important is that we consider only whether an age estimate is defined, or not. Ages defined with considerably better precision than the other ages are otherwise able to obscure the existence of valid estimates in the counting statistic. For Figure 16d the standard deviation applied is equal to 0.1 Ma, which is greater than for all estimates considered, with the result thus blurred. Note that the probability in this diagram is relative to the peak at ~2.0 Ma defined by five estimates.

Gaussian statistics assume a mean, and a scatter thereabouts. Another way to consider probability is to use a simple metric as below. Poisson statistics asks the question as to whether a particular distribution of values can be attributed to a random process. Thereby one can assess the likelihood that measured values have fallen randomly. We assume local growth at the time in question. With n=19 measurements, and each growth event able to occur (randomly) anywhere in a time interval of 3 Ma, we can use a simple Poisson statistic $[P = (rt)^k . e^{-rt}/k!]$ to estimate the probability that one or more events occur in the same time range: where k = 0, 1, 2, 3, 4, 5 with k! the factorial, r the rate at which n events occur, and t = 0.2 Ma is the time interval sampled. For this sample set the estimated probabilities are ~[28.2%, 35.7%, 22.6%, 9.5%, 3%, 0.8%]. More samples need to be measured to take such an analysis further, but for now one can assert that there is a reasonable chance that events inferred at ~2.00, 2.25 and 2.5 Ma are unlikely to have been the result of random coincidence.

# 4 Discussion

## 4.1 Timing episodes of mineralisation in the Martabe district

To determine the periods of alunite growth, each age determined by the method of asymptotes and limits was plotted using a Gaussian distribution to smooth the data. Distinct periods of alunite growth could thereby be distinguished across the 10 samples measured in this study (Fig. 16). The oldest events recorded occurred between ~3.5–2.9 Ma. The dominant events occurred at ~2.25 and 2.00 Ma. Younger growth events occurred at ~1.70 and 1.50 Ma. However, since the data are few, insufficient estimates exist to allow inference that all inferred events will become frequently measured ages (FMAs). Only the events at 2.5, 2.25 and 2.00 Ma had enough data to allow the conclusion that the confluence of ages was unlikely to be the result of random coincidence.

Sample D3011643 produced older events which overlap in timing with the emplacement of the dacite $3.8 \pm 0.5$ Ma (U-Pb on zircon) and hornblende-bearing andesite $3.1 \pm 0.4$ to $2.8 \pm 0.3$ Ma (U-Pb on zircon) flow dome complexes (Sutopo, 2013). The clustering of ages and the overlap with flow-dome formation gives us confidence that the interpretation of age spectra using the method of asymptotes and limits is not simply an artefact of contamination or excess argon.

Since the alunite measured was retentive, the FMAs recognised must therefore be interpreted to reflect the existence of several distinct and separate very short-lived periods of alunite growth during the history of alteration and mineralisation. The asymptotes and limits recognised are consistent amongst the sample set because different short-lived episodes of mineral growth have taken place at distinctly different times. Microchemical or microstructural variation might be the reason that different growth episodes produced alunite with different diffusion parameters. The frequency with which these growth events occur appears to lie between 250-300 ka, with individual events taking far less time to complete.

A map of the interpreted fluid systems is shown in Figure 17. More geological information is needed to constrain the alunite alteration systems further, but this is difficult due to the area being covered by dense rain forest and restricted zones. Worse, any attempt to 'vector' towards an orebody using geochemical methods is potentially doomed to failure, since individual fluid pulses permeated different channels. Geochemical samples needed for vectoring should have been taken from the same channel. The Purnama pit exhibits the influence of the 2.00 Ma fluid system overprinting the 2.25 Ma fluid system, potentially explaining the higher grades.

## 4.2 The role of the plateau in age spectra

The age spectra measured are not simple plateaux (cf Fleck et al., 1977; Jourdan et al., 2004; Sharp and Renne, 2005; Schaen et al., 2021) and some authors (e.g., Li et al., 2020) argue that an age spectrum is not reliable unless it approximates a plateau. Li et al. (2020) discuss why a plateau might be absent and recommend that such 'imperfect' results be considered less reliable. Yet, while it is true that there are many circumstances in which a plateau will result in a measured age spectrum, there is nothing in the theory and practice of argon geochronology that requires an age spectrum to be a simple plateau. There are many circumstances in which a plateau will not be evident, nor should have been expected.

There are underlined methodological reasons that can lead to data that approximate a plateau, e.g., the use of laser-rastering or the use of strong acids to remove parts of a microstructure that appear to be altered. There

are also theoretical scenarios that require age data to approximate a plateau e.g., rapid cooling from temperatures that were sufficiently high as to prevent (or remove) prior accumulation of radiogenic argon, or a single growth event at temperatures that ensured diffusion was insignificant for a particular mineral. However, the common definition of a plateau (e.g., Schaen et al., 2021) is not compatible with a holistic approach that recognises that plateaux may result from microstructures that represent

relatively small fractions of the sample volume. There is thus no reason to set an arbitrary limit on the percentage of gas required to define a plateau segment (Fig. 18). Numerical modelling (Fig. 19) demonstrates the existence of plateau segments with far less than 50% of the total gas release.

There are many publications in which small plateau segments have been ignored or passed over, although they may well be significant. For example, an audit of Ren and Vasconcelos (2019) reveals

similar spectra, although the lack of detail in the published laser results would have obscured the smaller plateau recorded in their figure 4b. Note that their first plateau segment demonstrates release from a younger-grown alunite generation that had lesser retentivity, whereas in the results we present the reverse seems to apply: the younger-grown alunite generation had greater retentivity.

The issue is that analysis of an argon spectrum that resembles a plateau is straightforward, but the same

is not true for the analysis of an argon spectrum that results from progressive release of gas from different microstructural reservoirs. Such circumstances require a rethink as to how age spectra are analysed, moving away from the notion that: i) *a priori* a single age is to be expected; and that ii) the scatter of ages thus obtained should define a normal distribution.

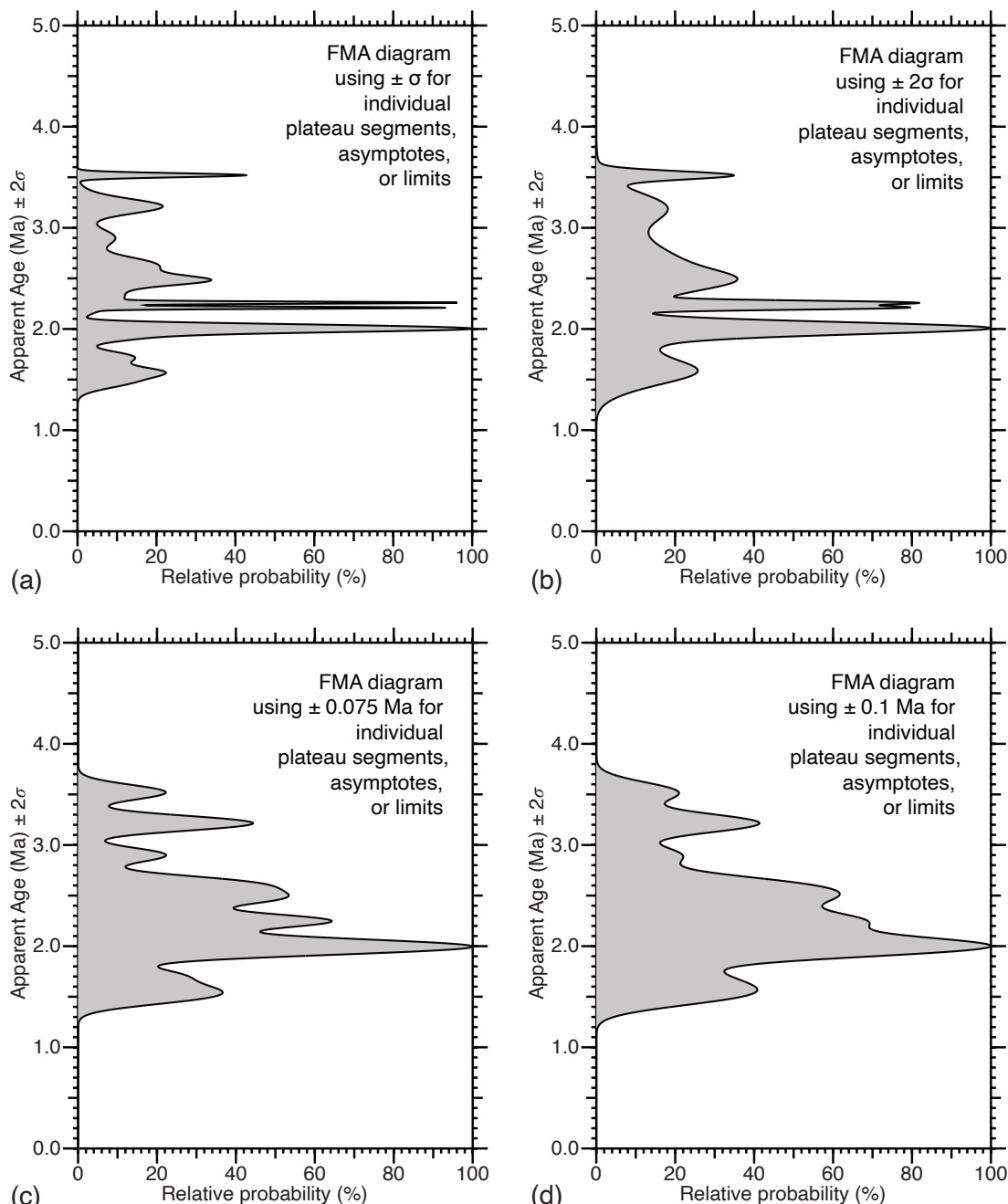

**Figure 16** Gaussian distributions used to develop a relative probability curve showing Frequently Measured Ages (FMAs) based on the application of the method of asymptotes and limits to the measured age spectra. The data utilised is from individual age plots: in (a) with the same uncertainties (σ) as individually estimated for each plateau segment, limit, or asymptote; in (b) by doubling these uncertainties, and using 2σ to scatter
each estimate; while in (c) and (d) imposing a fixed uncertainty on each estimate, with the value as shown

| Age (Ma) | Uncertainty (2σ Ma) |
|---|---|
| 3.22 | 0.20 |
| 2.45 | 0.18 |
| 1.48 | 0.13 |
| 2.04 | 0.05 |
| 2.01 | 0.06 |
| 1.58 | 0.10 |
| 2.59 | 0.19 |
| 2.00 | 0.05 |
| 1.72 | 0.12 |
| 2.90 | 0.18 |
| 2.27 | 0.17 |
| 2.48 | 0.09 |
| 1.93 | 0.10 |
| 3.52 | 0.04 |
| 3.21 | 0.13 |
| 2.65 | 0.15 |
| 2.26 | 0.02 |
| 2.21 | 0.02 |
| 1.98 | 0.06 |

**Table 1** Gaussian distributions used to develop a probability curve showing Frequently Measured Ages (FMAs) based on the application of the method of asymptotes and limits to the measured age spectra. The data utilised is as shown on individual age plots: Figure 16a utilised uncertainties individually estimated for the scatter of the mean of each plateau segment, limit or asymptote; while Figures 16b-16d imposed a fixed uncertainty on each plateau segment, limit or asymptote.

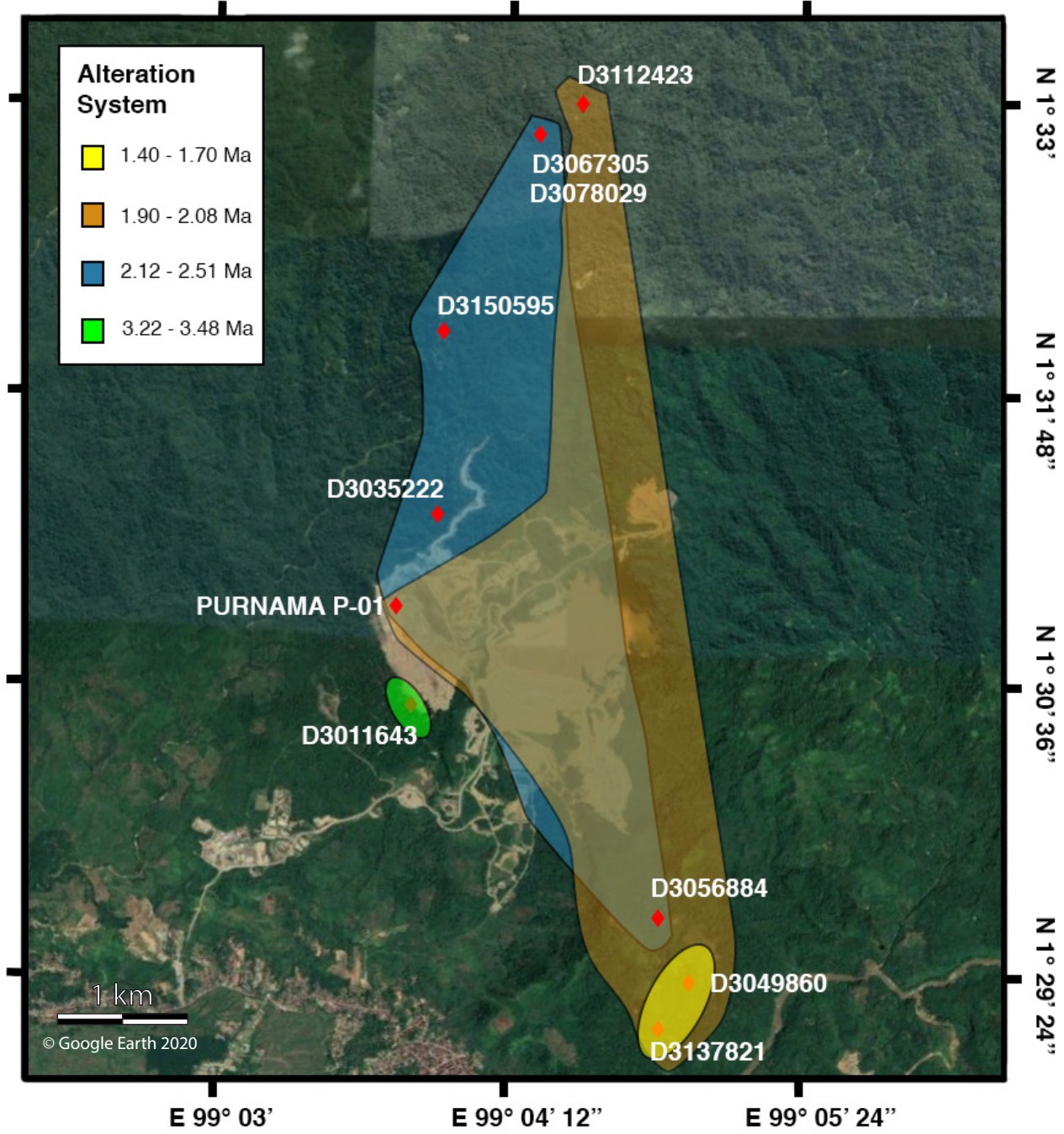

**Figure 17** Map of Martabe gold field with alteration systems mapped. Alteration systems outlined according to the sample location and period of alunite growth. The two older events are grouped together.

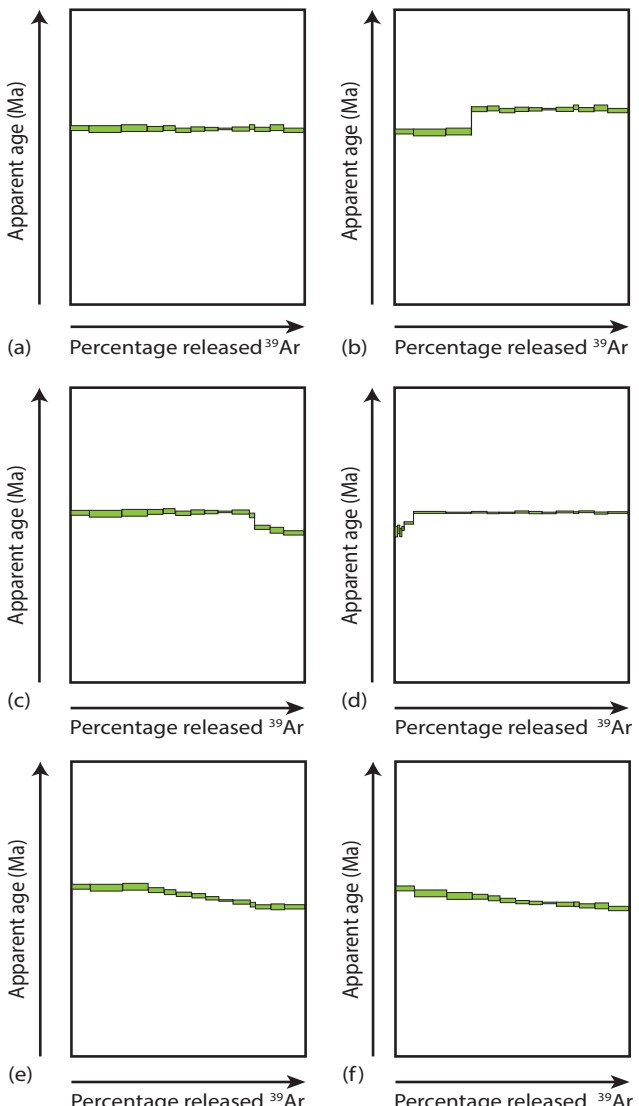

**Figure 18** An age spectrum has a morphology: In (a) a plateau formed because the mineral has a single "growth age", or very rapidly cooled from elevated temperatures so it retained all radiogenic argon subsequently produced, yielding a "cooling age"; while in (b) a mineral aliquot has two growth ages, with the second period of growth having produced a microchemistry or a microstructure that was less retentive of argon in the mass

spectrometer; compared with (c) when the second period of growth produced a microchemistry or a microstructure that was more retentive of argon in the mass spectrometer, but not so different as to prevent mixing between the two domains. In (d) the second period of growth involved only a small volume fraction and more scatter was produced in the apparent age during measurement. A mini-plateau can be recognised. In (e) while in (f) we highlight issues when an age spectrum involves progressive mixing of gas from different

reservoirs. Plateaux segments are still evident in (e) while in (f) the upper and lower limits of a staircase spectrum are all that remains to hint at the existence of distinct microstructural reservoirs formed at different ages.

## 4.3 How do complex age spectrum result?

Complex age spectra typically result when there are different types of gas reservoirs in the aliquot of mineral grains analysed during a step-heating experiment. Typically, this occurs because: i) there is more than one microstructural domain; and ii) the microstructure/microchemistry of individual domains requires different diffusion parameters, and these govern differing patterns of $^{39}$Ar release during the experiment; or iii) when one domain degasses differently because of microstructural change and/or breakdown as temperatures increase in sequential steps. In this case, because so many heating steps were undertaken, it was possible to compare the age spectra with graphs showing the variation of different isotopic ratios, specifically: i) the Ca/K and Cl/K ratios, which showed a gradual increase during the experiment, until a sudden rise during the final steps; ii) the Arrhenius plots and the linked log $r/r_0$ plots, which allow recognition of distinct diffusion domains, their relative volumes, and estimates of the relevant diffusion parameters; and finally iii) the inverse $^{40}$Ar isotope ratio correlation plot which allows recognition of inverse isochrons, but more importantly, which allows the progression of mixing between different gas reservoirs during the experiment to be monitored and examined.

Statistical models that rely on the assertion that a single age should be present are overly simplistic, and do not reflect the observed pattern of age variation in published spectra from very many samples, including those reported in this paper. For these reasons, a theory of how to analyse complex age spectra caused by mixing was developed by Forster and Lister (2004). Application of their method of asymptotes and limits has been shown to allow analysis of even the most complicated of age spectra. Frequently Measured Ages (FMAs) estimated from such analyses consistently allow the recognition of (and the replication of estimates for) the timing of potentially significant events in the history of the rocks studied. Nevertheless, independent arguments for the significance of each FMA need to be given.

Because the experiments were designed to allow many (32-35) steps, we were able to compare diffusion parameters inferred for steps in one part of an age spectrum with parameters inferred for other segments of the same age spectrum. In almost all cases, observed differences in the inferred diffusion parameters explicitly relate to variation in the morphology of the measured age spectra. Multiple plateau segments and/or asymptotes are therefore a natural complication which is to be expected.

This is not to say that the inferred diffusion parameters are necessarily those that applied in the natural environment. Such an assumption would require metastability during the furnace step-heating episodes, while in fact the material being measured may have undergone significant microstructural modification, phase transformations, metamorphic reactions, or melting and/or breakdown of the crystalline structure. But since domains in the age spectra correlate with domains in the Arrhenius plot, we can be certain that: i) a basic mapping of age and material properties had already been established in the natural environment, and that: ii) this mapping persisted during the UHV diffusion experiment. In the present case, since the domains in the age spectra appear to reflect growth ages, we could identify different and distinct periods of hydrothermal mineral growth in the natural environment. Overprinting fluids active in the formation of Martabe Gold deposit produced alunite in several growth generations, with variation in mineral geochemistry or microstructure ensuring slightly different diffusion parameters. These differences enabled discrimination of these alunite generations during our UHV diffusion experiments.

## 4.4 Dealing with the natural complexity of an age spectrum

The otherwise hidden detail of an age spectrum can be liberated by carefully choosing the heating steps to allow the maximum number of meaningful episodes of gas release, noting that temperature and time for each heating step must be chosen to ensure as uniform an aliquot (of the $^{39}$Ar released in each heating step) as is possible. Too few steps and the morphology will be utterly obscured.

At the same time, the temperature variation in the sequence of heating steps must ensure that the distribution of points on the inverse temperature axis of an Arrhenius plot is as uniform as possible. Steps that allow meaningful release of $^{39}$Ar aliquots vary with the applied temperature. Steps that allow a uniform distribution on an Arrhenius plot vary with the inverse of temperature.

The age spectra illustrated do have some common aspects: i) they are not simple plateaux; ii) their morphology typically involves an upper limit OR a plateau segment in the first few percent (up to 10%) of $^{39}$Ar release; iii) the spectra display an asymptotic decrease in age towards a lower limit (which is sometimes a plateau segment); iv) the last few percent of $^{39}$Ar release is marked by a reduction towards (or past) zero apparent age. This variation can be readily explained. However, first note that if two generations of growth produced microstructures and microchemistries that were identical, their ages would uniformly mix. The two generations would not be evident in the age spectrum. Only if there is a difference in retentivity will a carefully designed step-heating experiment with precise temperature control have the potential to discriminate between the ages of the different growth generations. Progressive mixing might prevent all but a few steps from defining the "end member" ages (Fig. 18e), or worse, progressive mixing might result in an age gradient with only the upper and low limits of age having the potential to be statistically significant (Fig. 18f).

Irrespective of whether there was loss of argon from the mineral lattice as the result of diffusion in the natural environment, what is recorded in the mass spectrometer depends critically on the pattern of argon gas release from the different microstructural domains, governed by the difference in their diffusion parameters. The final part of the age spectrum is obscured, potentially because of mixing with a gas reservoir derived from inclusions [Fig. 6 and see later discussion]. This was made evident by using the Turner inverse $^{40}$Ar isotope ratio correlation plot. Because the step-heating schedule produces so many well-populated steps, it was possible to use such plots to track the progression of mixing of different gas reservoirs. In Figure 6, the first data points reflect the release of air, with successive data points moving progressively towards the gas reservoir defined by the lattice of domain A. Thereafter there is a short period during which heating steps dominantly release gas from domain B, but thereafter the trend returns to that defined by the mixing line between air and domain A. This is followed by a sharp loop and the plot records mixing with gas from a small-volume reservoir that may be defined by KCl and CaCl$_2$ salts. These begin to melt around 770°C. More complex stoichiometry might be involved, including with carbonate, but there is no data. Such interferences could have been removed by washing samples in acid baths, but we did not undertake such effort, because of likely damage to existing microstructure. Instead, we relied on the detail provided by heating schedules that produced many steps, thereby enabling recognition of mixing lines.

The low volume 'inclusion' reservoir affects only the last steps in the sequence, reducing their apparent ages. Typically, those steps are associated with age errors two orders of magnitude or more times the

average, and therefore they were not considered (even though we can recalculate the ages of the affected steps using the $^{39}Ar/^{40}Ar*$ intercept defined by the inverse isochron along the mixing trend for this late-stage low relative volume reservoir).

Rather than using acids to dissolve parts of the microstructure prior to measurement: i) prior to measurement, samples were rid of volatiles by prolonged warming under ultra-high-vacuum (UHV) conditions at temperatures below the point at which any significant percentage of the $^{39}Ar$ stored in the mineral lattice begins to be released; ii) by using carefully chosen temperature-time schedules (determined *a posteriori* by dint of experience with any given sample set) to ensure that as uniform as possible a release of $^{39}Ar$ in each heating step takes place; iii) while, at the same time, a distribution of temperature is maintained (amongst the steps that allow meaningful release of $^{39}Ar$ aliquots) to ensure as uniform as possible a distribution of points on the inverse temperature axis of an Arrhenius plot; iv) taking note of the above, choosing sample size (dependent on age and potassium content) so that the chosen steps for a particular experiment are able to liberate very many different age data points.

In this way we were able to provide unprecedented detail in the measured age spectra. The use of a properly calibrated resistance furnace is essential in that such equipment offers rigorous (and documented) temperature control, while forced-injection of chilled cooling water through the assembly at the end of a period of heating (in this case 15 minutes) ensures the rapid cooling that is essential in allowing the temperature-time history in each measurement step to better approximate a square wave. Otherwise, it is difficult to justify the use of analytical solutions in inferring diffusivity $(D/a^2)$ where 'a' is the radius or dimension of the diffusion domain. The production of detailed age spectra is essential since without many steps, it is difficult to consider the effects of such variables such as: i) argon recoil during neutron irradiation, during the early steps of the UHV diffusion experiment; or ii) the effect of mixing with impurities such as KCl and $CaCl_2$ in inclusions (in the case of the present results) during the final steps of the UHV diffusion experiment. Thus, when complex age spectra are obtained, meaningful analysis can only be provided by relying on the detail provided by numerous heating steps.

## 4.5 Examining mixing trends in context

It is important to analyse asymptotes and limits in the apparent age spectra in context. Contamination in the early steps of the step-heating sequence is evident because release of $^{39}Ar$ is concordant with the predictions of diffusion modelling, while the release of $^{40}Ar$ follows an entirely different metric. This suggests that excess $^{40}Ar$ was present, and that it was released via fast-diffusion pathways during these initial steps. Such degassing from these fast-diffusion pathways is dynamic since the permeability that allows escape of non-lattice $^{40}Ar$ appears to have been created during step-heating. At the same time, the concordant release of $^{39}Ar$ continues unchanged, implying that the dimensions of diffusion domains in the lattice is also not changing, which in turn requires the diffusion distance to be less than the spacing between fast diffusion pathways being created in the crystalline alunite. Such fast-diffusion pathways are most likely related to cracks and fractures.

We also highlight the importance of conjoint analysis when correlating trends between mixing lines on a Turner inverse ratio plot, or when examining mixing between individual plateau segments, or trends towards individual asymptotes or limits in apparent age plots. In Figures 6 and 9, for example, the

intermediate asymptote [2] correlates with the start of significant mixing with the gas from the inclusion reservoir, as inferred using the Turner plot. We note that since the observed correlations reflect variation in apparent age, they cannot result from artefacts introduced during the step-heating diffusion experiments. A similar variation in retentivity must have existed in the natural environment.

      The step-heating schedule ensured that only small amounts of gas were released in each step, so the
progression of the variation in the isotopic ratios from one step to the next allows ready identification of different gas populations, and the mixing trends between them. Typically, two or three alunite lattice gas reservoirs can be identified, with mixing initially moving away from the atmospheric reservoir towards the alunite I inverse isochron. The mixing trend then moves towards alunite II and sometimes back again to alunite I (e.g., Figs. 6 and 9) before the diminishing yield from the alunite aliquot is
overcome by release of gas from inclusions. The subtleties of these mixing trends can be understood if they are examined in the context of the variation in the associated Arrhenius data, since there is often only a small difference between the inferred activation energy and normalised frequency factor.

      The different diffusion parameters have quite different effects on the pattern of gas release, As shown by Forster and Lister (2014) frequency factor tends to increase rapidly in conjunction with an increase
in activation energy, narrowing the temperature window during which most of the gas is released from the corresponding domain. Thus, the age can decrease as alunite II degasses, because it formed at a younger time, but then as this reservoir depletes, the age can once again increase as degassing from alunite I once again becomes dominant (e.g., Fig. 9).

      This means that care needs be taken in estimating relative volumes of alunite I and alunite II. Typically,
this pattern is evident in the comparative radius plot (e.g., Fig. 9c) when this plot is examined conjointly with the apparent age spectrum (Fig. 9a), the Arrhenius plot (Fig. 9b) and the Turner inverse $^{40}$Ar correlation plot (Fig. 9g). As previously noted, the last steps of each of the heating schedules are discounted because significant contamination took place involving an 'inclusion' reservoir. This late-stage mixing trend is evident in all Turner inverse ratio plots but is especially evident in Figure 6. Here
it can be shown that the inclusion (?) reservoir is likely not of zero age, but the estimates as to its age using an inverse isochron have considerable uncertainty and are thus not further considered.

      This low volume gas reservoir was most likely released from included minerals that contain $CaCO_3$ + $CaCl_2$ + KCl (Figs. 7-15). The key to our reasoning is that: i) small volumes are involved (<2% of the total $^{39}$Ar released); ii) the mixing with the unidentified reservoir begins at the KCl or $CaCl_2$ melting
temperature; and iii) the Ca/K ratio in the unidentified reservoir approaches integral values (e.g., 6:5 for Ca/K whereas the Cl/K ratio is 5:1 which can be achieved by a molal composition of inclusions equivalent to $4CaCO_3$ + $2CaCl_2$ + KCl). Alternative contaminants include clays such as kaolinite or dickite, but these compositions do not typically include K, Ca or Cl. Clay breaks down at >1000°C, coincidentally at the temperatures at which the inclusion reservoir is most evident. If calcite ($CaCO_3$),
sinjarite ($CaCl_2$), sylvite (KCl) or chlorocalcite ($KCaCl_3$) mineralogy was present in such material, and cryptocrystalline, it would have escaped notice. It is also possible that the minerals listed above were present in microcracks, rather than existing as discrete inclusions.

## 4.6 Inverse isochrons on inverse isotope ratio correlation plots

The Turner inverse [40]Ar isotope ratio correlation plot in *eArgon* uses the 'new' York regression as set out by Mahon (1996). This author corrected the formulae given by York (1969). However, even Mahon (1996) had minor errors in the published formulae, and additional corrections were necessary in determining the uncertainty statistics, as noted by Trappitsch et al. (2018) and included in his Python code. The York-Mahon-Trappitsch regression uses a recursion that allows an estimate as to the uncertainty on the y-axis, expressed by the percentage excess [40]Ar [or in the other direction, excess mass 36]. Similarly, the York-Mahon-Trappitsch regression allows an estimate for the uncertainty on the x-axis. Based on the variation in the [40]Ar*/[39]Ar ratio, this translates into a percentage age variation.

The inverse [40]Ar isotope ratio correlation plots published here include inverse isochrons (and any associated statistical information) to define the age of the gas reservoir. For isochrons defined by more than three points, Pearson's chi statistic is shown. This is the square root of the mean square weighted deviation (MSWD) and it is assessed based on whether or not it is within the 95% confidence range calculated for n-2 degrees of freedom. Each inverse [40]Ar isotope ratio correlation plot provides regression data for individual isochrons, but only for isochrons with more than three points. We note that minor plateaux, asymptotes and limits recognised on the age plot can invariably be correlated with isochrons in these Turner plots. We note that the inverse isochrons so identified are typically defined by relatively few points, and that the degree of freedom (n-2) rarely exceeds six. Hence the expected variation in Pearson's chi statistic (Wendt and Carl, 1991) was utilised to assess whether the scatter remains within the 95% confidence limit (when n>3). However, whereas these authors were limited by manual use of numerical approximations, *eArgon* computes the gamma function directly: using the *tgamma()* function in the *<cmath>* C++ library. This enables the Wendt and Carl (1991) statistical analysis to be improved by determining the maximum likelihood for the chi value (estimated using a golden ratio method) based on the number of degrees of freedom, and then numerically estimating the likely standard deviation to be expected about this value. The values plotted are corrected for isotopic interferences occasioned by the presence of Ca and Cl, utilising the correction factors obtained by including the relevant salts in the canister during neutron irradiation.

Excess [40]Ar could be inherited by contamination or it could be introduced to the system by fluids at the grain boundary (Kelley, 2002). This can be detected from inverse isotope correlation diagrams by determining the initial [40]Ar/[36]Ar ratio of the sample (McDougall and Harrison, 1999). However, this method cannot readily be used for samples with multiple growth generations (e.g., Kunk et al., 2005).

## 4.7 Modelling the effect of different Arrhenius data used to estimate argon retentivity

Our diffusion experiments produced Arrhenius data that required theoretical closure temperatures between 400°C and 560°C for modest cooling rates (Figs. 7-15). With such retentivity, in the natural environment, radiogenic argon would have been fully retained in the alunite lattice, even during later hydrothermal events. Based on alteration assemblages, deposits such as Martabe form at temperatures >~200°C (White and Hedenquist, 1990; Simmons et al., 2005). This is compatible with estimates based on the trapping temperature for fluid inclusions (Saing et al., 2015) although peak temperatures may be higher, e.g., 300°C based on mineral assemblages (Simmons et al., 2005) or 350°C based on fluid

inclusion homogenisation temperatures (Saing et al., 2015). But even with such peak temperatures, if our estimates as to the diffusion parameters are correct, alunite is still highly retentive of argon and thus should reliably allow $^{40}Ar/^{39}Ar$ geochronology to time different periods of ore deposition in the same epithermal system, if peak temperatures are only briefly attained.

In contrast, the same statement cannot be made if alunite is less retentive of radiogenic argon, as would be the case for estimates for the closure temperature in the ~240-320°C range as reported by Love et al. (1998), Landis et al. (2005), Arribas et al. (2011), Waltenberg (2012), and Ren and Vasconcelos (2019). With such unretentive parameters, if temperatures were as high as surmised, evidence for older events would have been erased. Even the basic morphology of age spectra as we present here could not exist. Could this mean that the Martabe alunite is more retentive than alunite found elsewhere? This seems unlikely, as other authors have also recognized multiple growth events in the same epithermal system (e.g., Deyell et al., 2005). Moreover, age spectra with multiple plateau segments have also been measured, e.g., figure 4b in Ren and Vasconcelos (2019). Since geochronological evidence for different growth ages is preserved, closure temperature in these earlier works has been underestimated.

One issue could be the effect of microstructural change (either in the natural environment or in the laboratory) so that the results obtained did not reflect the effects of diffusion alone. In other words, perhaps resetting has taken place as the result of microstructural change, including by grain size reduction, e.g., by buckling and misorientation of crystallites in a polycrystalline nanofilm irradiated by the beam of a transmission electron microscope as reported by Ren and Vasconcelos (2019). In the natural environment, recovery and/or recrystallisation during later thermal events may occur, or solution/dissolution processes that in effect regrow alunite, so that older ages do not survive.

To bring clarity to this issue, the *MacArgon* program (Baldwin and Lister, 1996) was used to quantitatively forward model the effect of specific temperature-time paths (Fig. 19a). The T-t paths shown are bounded by the 200-260°C temperature range suggested by Saing et al. (2015) for the formation of the Purnama deposit. The effect of argon loss during temperature spikes was separated from the effect of argon loss due to ambient conditions by modelling two scenarios: i) the ambient temperature set at 100°C (Figs. 19b, 19c); and ii) the ambient temperature set at 200°C (Figs. 19d, 19e). The same MDD model alunite geometry was used throughout: i) alunite I approximated by 5 fractal iterations of a spherical geometry, with $r/r_0 = 2.5$ and $v/v_0 = 0.9$; ii) alunite II a single more retentive spherical domain. Diffusion parameters are shown in Figure 19 [and in the supplementary information].

The observed spectra cannot be replicated using the unretentive diffusion parameters published by Ren and Vasconcelos (2019). The modelling shows unequivocal evidence for argon loss (Figs. 19b and 19d) and that the morphology of any measured age spectrum would have been smoothed should these parameters have been relevant. In contrast, the retentive diffusion domains suggested by our Arrhenius data allowed a good fit to the measured spectrum (Figs. 19c and 19e). Hence, the measured spectra reflect the effect of growth events alone, without partial loss of $^{40}Ar$ in the natural environment.

## 4.8 Comparison of different Arrhenius data used to estimate argon retentivity

Because of the discrepancy above, we need to reflect on possible explanations as to why other authors might have obtained low retentivity estimates for the diffusion parameters. One reason might be that we

analysed the Arrhenius plots from our alunite step-heating experiments, ensuring that our estimates were compliant with the Fundamental Asymmetry Principle (or FAP) recognised by Forster and Lister (2010), while previous authors did not. It is necessary to apply the FAP if a multi-domain diffusion (MDD) model is assumed, noting this can still be done even if the dominant release of $^{39}$Ar does not always come from diffusion domains with the same activation energy. Complications ensue if the 'dominant' activation energy varies between one part of an Arrhenius plot to another, e.g., see Cassata and Renne (2009) and the subsequent discussion by Lovera et al. (2015) but this can be dealt with by local application of the FAP (e.g., as in Forster et al., 2015). Also note that, while anisotropic diffusion can also cause an upward bend in slope between one part of an Arrhenius plot to another (Huber et al., 2011), the FAP can still be applied to estimate the end-member properties.

The analysis of Arrhenius data does become more difficult if different regions on the Arrhenius plot are dominated by release of $^{39}$Ar from domains with different activation energies. The problem is that, between the different regions on the plot, there will be an intervening zone of mixing. It is an issue therefore that Ren and Vasconcelos (2019) interpreted slopes determined in a potential zone of mixing as representative of a distinct diffusion domain. It is a mistake to consider that there should be a one-to-one correspondence between 'slope' on an Arrhenius plot at any particular point and any assumed activation energy at that point. Ren and Vasconcelos (2019) may thus have erred in attempting to separate a distinct activation energy when in fact $^{39}$Ar release was transitional between domains with quite different activation energies. Each point on an Arrhenius plot sums contributions from all the diffusion domains that are present, and this hinders comparison of points 1:1 with steps in other plots.

The Ren and Vasconcelos (2019) data was re-examined, and dubious estimates were eliminated. Only those values of the diffusion parameters that were FAP compliant were plotted (Fig. 20). From this diagram it is evident that the most accurate estimates of activation energy from the Ren and Vasconcelos (2019) experiments coincide with our own. However, the diffusion dimension in our experiments is ~700 times greater than that which applied in the Ren and Vasconcelos (2019) experiments. This could be explained by the episodes of grain size reduction these authors observed, noting these may be artefacts caused by nano-scale thin slices of the alunite lattice buckling and kinking in the beam during transmission electron microscopy. Their higher estimates for normalized frequency factor are thus to be expected. The material in our experiments was not subject to such grain size reduction, explaining the difference in estimated diffusion dimension.

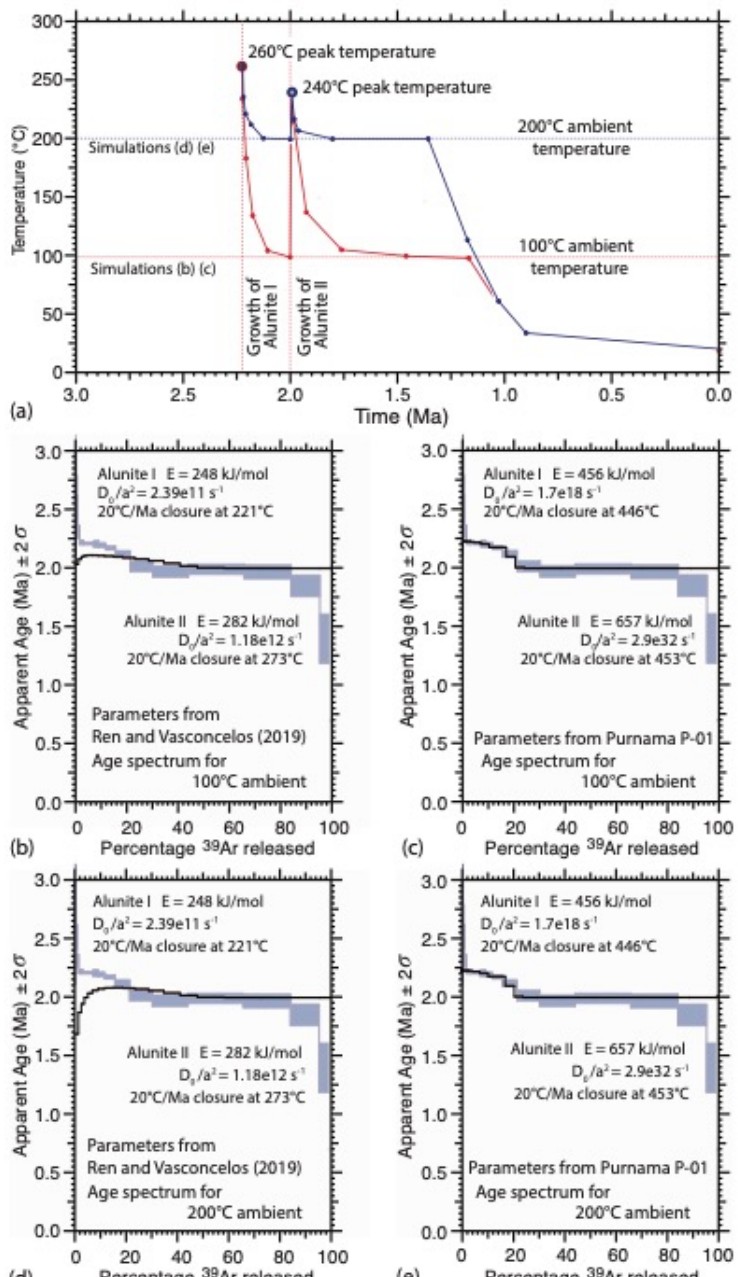

**Figure 19** *MacArgon* used to simulate the effect of different MDD models (with Alunite I versus Alunite II in the ratio 1:5). Temperature-time paths (a) have 260 and 240°C spikes: (b) and (c) dropping back to a 100°C ambient; and (d) and (e) back to a 200°C ambient temperature. The Ren and Vasconcelos (2019) parameters are not sufficiently retentive, causing significant argon loss at peak temperatures (b) and at ambient temperatures >150°C (d). In contrast our parameters allow accurate simulation of the observed age spectra (c, e) though we ignore the effect of the inclusion reservoir in the last 20% of gas release.

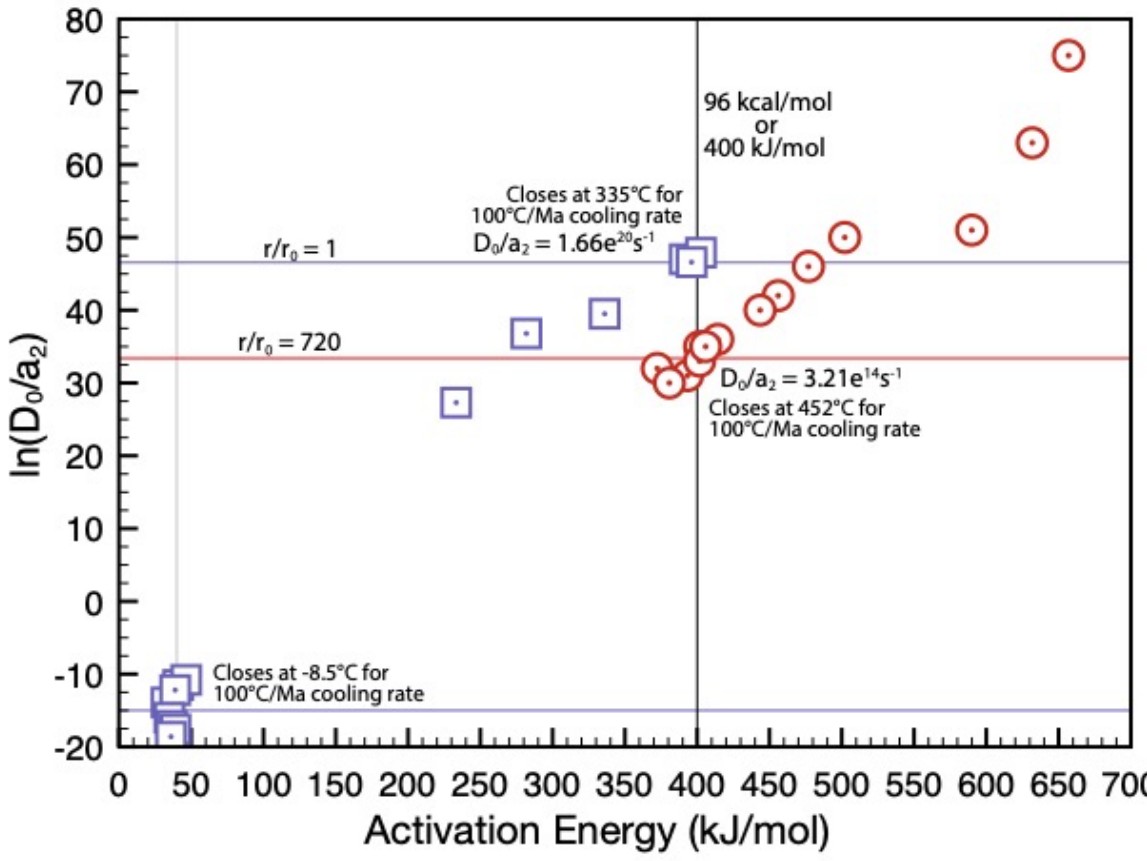

**Figure 20** A plot of activation energy against the natural logarithm of the normalised frequency factor. This shows all FAP compliant estimates of diffusion parameters made by Ren and Vasconcelos (2019) [blue squares], enabling comparison with estimates made here [red circles]. The activation energy is the same. Normalized frequency factors differ so either the samples used in the Ren and Vasconcelos (2019) suffered a grain size reduction during the experiment, or the diffusion dimension was inherently much smaller.

## 4.8 Fundamental limitations of UHV $^{39}$Ar diffusion experiments

Increasing temperature during a step-heating experiment will eventually cause microstructural change, phase transformations, mineral decomposition and/or melting. Ren and Vasconcelos (2019) suggest that a phase transformation occurs at ~450°C which is the temperature when our experiments started. Such changes could render our estimates for the diffusion parameters invalid. However, whereas the microstructural modification described by Ren and Vasconcelos (2019) substantially reduced the diffusion dimension (Fig. 20), there was no change in the estimated activation energy, no change in the lattice parameters, and thus no evidence for a phase transformation. Nevertheless, the potential for changes of material structure during the experiment remains a drawback for the application of data from UHV diffusion experiments. It might have been better if we had conducted our diffusion experiments at lower temperatures, for longer time periods. However, as shown in the supplemental data very little gas was released below 450°C, so it would not have been possible to accurately apply the equations relating diffusivity to the percentage partial loss of $^{39}$Ar.

We re-examined the release of gas at the lower temperatures used by Ren and Vasconcelos (2019). The percentage release of $^{39}$Ar documented by these authors allows an estimate of the relevant diffusion parameters, and Arrhenius data that require closure below ~10°C. Thus, any stored argon would have already been released, prior to measurement. Hence, this was not lattice argon. Escape of $^{39}$Ar from this low retentivity reservoir must have been the result of a dynamic process occasioned by the conditions of the experiment. The Arrhenius data may estimate the kinetics for grain boundary diffusion, e.g., as would be the case if $^{39}$Ar produced by recoil during irradiation was progressively expelled as the result of mechanical distortions caused by the initial heating steps. These distortions may have created fast diffusion pathways that enabled this early release of $^{39}$Ar. Since very small volumes of gas are involved, depletion of this non-lattice $^{39}$Ar reservoir explains the steepening of the gradient on the Arrhenius plot.

An alternative explanation for the Ren and Vasconcelos (2019) data is therefore that the estimates made in the transition zone leading to the first release of lattice $^{39}$Ar were overwhelmed by the effects of mixing with non-lattice $^{39}$Ar. The intermediate diffusion parameters preferred by Ren and Vasconcelos (2019) are thus artefacts, and the upward slope on their Arrhenius diagrams reflects the prior existence of a more retentive diffusion domain. Mathematically there is no coincidence between temperature, diffusivity, and the average slope of a progression of individual points on an Arrhenius plot. All domains contribute in greater or less amounts to the release of $^{39}$Ar used to estimate the bulk diffusivity.

Cassata et al. (2013) also interpreted inflections in the Arrhenius plot as the result of structural transitions in the mineral due to laboratory heating. They argued that thermal expansion can occur on short timescales, and lattice deformation can be caused by the transition from triclinic to monoclinic that can pose an activation energy barrier and exhibit hysteresis on retrograde heating. But while such effects may well have been taking place in the mass spectrometer, one would not then expect a corresponding variation in the age spectrum. Age variation can only reflect processes that occurred in the natural environment. Since there are variations in the age spectrum that correspond to the observed changes in the Arrhenius plot, Cassata et al. (2013) cannot be correct [see Lovera et al., 2015].

Many of our estimates for the diffusion parameters are made using data that extend over a considerable range of temperature (e.g., Fig. 5) so material properties are unlikely to have changed in that range. The

scattering of orientation and the reduction in grain size observed by Ren and Vasconcelos (2019) was coincident with the beginning of a change of slope on the accompanying Arrhenius plot. Our results similarly demonstrate a change in slope on the Arrhenius plot (e.g., Fig. 5), but this occurs at temperatures 80-100°C hotter than for the samples examined by Ren and Vasconcelos (2019). If indeed a microstructural change is occurring in our samples, one might therefore surmise that it began to occur at higher temperatures than in the Ren and Vasconcelos (2019) samples. This might reflect the different ways that samples were prepared (e.g., we did not use acid to clean the aliquot). There may also be differences in the way samples were irradiated (e.g., we used 1.0 mm thick cadmium liners that minimized the effect of thermal neutrons, thus ensuring minimal recoil and the least lattice damage during irradiation). Finally, there are also differences in the way samples were examined (e.g., we did not excite our sample in the beam of an electron microscope). It might also be that there is a systematic variation in material properties, with more retentive alunite grown at higher temperatures in the natural environment. Unretentive parameters cannot explain the observed age spectra (Fig. 19), so our inferences as to the diffusion parameters for alunite may in fact be correct.

## 4.8 Further work

There is a need to examine the microstructural evolution of the alunite at Martabe in the context of the new age data presented in this paper. The theory and practice of argon geochronology also needs some attention, in particular requiring recognition that the Schaen et al. (2021) criteria for the definition of a plateau fail when it comes to the analysis of the morphology of a complex age spectrum. There needs to be a focus on the analysis of mixing, with new developments such as that proposed by Carter et al. (2020) ensuring future innovation.

## 5 Conclusion

Through utilisation of a suitable heating schedule, determined *a posteriori*, we were able to degas alunite samples without losing too large a percentage of gas in a single step. This allowed extraction of information from both the age spectra and the Arrhenius plots. The detail allowed recognition of mixing trends, and application of the method of asymptotes and limits. Frequently Measured Ages (or FMAs) could be recognised, and these were shown to have significance using a Poisson statistic.

The recursive formula in Dodson (1973) allows the accurate calculation of closure temperature for any specific cooling rate and/or ambient pressure. Given the parameters listed by Ren and Vasconcelos (2019), hypogene alunite spheres would exhibit closure temperatures in the range 234-287°C. Such retentivity might allow alunite to retain growth ages in hypogene environments, but this lesser retentivity would result in age spectra that exhibited typical diffusional loss profiles in epithermal systems. Such spectra are not observed, so their diffusion parameters cannot apply.

With the diffusion parameters determined based on our analysis of our Arrhenius plots, we were able to accurately replicate the morphology of the observed spectra, by forward modelling using the *MacArgon* program. This is to be expected since our data put the closure temperature of alunite ranging between 400°C and 560°C, which is above the ~200-260° range in temperature expected for the formation of the Martabe deposits. This result, and the heterogeneity in age, gives confidence that the inferred ages from

$^{40}Ar/^{39}Ar$ geochronology time the growth of alunite and that these are not cooling ages. Our Arrhenius data suggests that alunite is considerably more retentive of argon than previous estimates would allow.

Since the alunite measured was retentive, the FMAs recognised reflect distinct and separate short-lived periods of alunite growth during the history of alteration and mineralisation around the Martabe deposits. The asymptotes and limits recognised are consistent amongst the sample set because different short-lived episodes of mineral growth took place at distinctly different times. We do not know the reason that different growth episodes produced alunite with different diffusion parameters, however.

Although our map of the extent of the overprinting fluid systems lacks detail, it is evident that mineralisation in the Purnama pit is the result of two specific fluid events, one enriching the other. These growth events took place at ~2.25 and 2.00 Ma. The separation in timing may reflect long term periodicity in the earthquake cycle. The frequency with which these growth events occur is between 250-300 ka, providing some constraint as to the duration of these long-term oscillations.

Only when samples come from the same fluid system would 'vectoring' using variation in mineral compositions allow determination of the direction of fluid movement towards or away from an orebody.

## Data availability

All necessary data is provided in the supplementary information.

## Author contribution

The paper includes modified information from the MPhil thesis by Muston (2020). Lister undertook modelling and simulation and was responsible for manuscript resubmission and reviewer discussion. All authors contributed to data analysis, discussion and reviewing of the manuscript.

## Disclaimer

This article is based in part of the MPhil Thesis of the first author submitted to the Research School of Earth Sciences (RSES), Australian National University.

## Conflicts of Interest

The authors declare that they have no conflicts of interest but note the software utilised was written by one of the authors (Lister) and that the applications used are commercially available from the AppStore.

## Funding Statement

Australian Research Council Linkage Project LP130100134 "Where to find giant porphyry and epithermal gold and copper deposits" with additional support from PT Agincourt Resources.

## Acknowledgments

PT Agincourt Resources, Martabe Mine, Sumatra, Indonesia, provided logistic support for this project. Shane Paxton and Sareh Rajabi assisted with mineral separation. The paper benefitted from the efforts of the Editor, Klaus Mezger, the Associate Editor, Darren Marks, and several anonymous reviewers.

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
