# Peer review of "Direct dating of overprinting fluid systems in the Martabe epithermal gold deposit using highly retentive alunite"

_Geochronology, 2021_

## Editor Comment (EC1)

The manuscript by Muston et al present 40Ar/39Ar furnace step-heating and 39Ar diffusion data from alunite in ten samples from the Martabe epithermal gold deposit in Indonesia. They use their closure temperature constraints to conclude that gold in the Purnama pit was the result of fluid rock interactions from ~ 2.25 and ~2.00 Ma.

This manuscript is filled with numerous misleading statements, but more importantly it is plagued by a lack of up to date knowledge of geochronologic methods and flawed data interpretations. For these reasons, which are explained in much more detail below, I strongly urge that this manuscript be rejected.

Lack of knowledge of published studies and use language that could mislead a non-specialist:
Line 56: There are numerous other examples of 40Ar/39Ar dating of alunite. All published data on alunite do not need to be cited here, but other work besides Arribas et al should be cited. Vasconcelos 1999 provides a nice overview of dating supergene minerals. It should be cited here.

Line 224: "These are the first detailed ultra-high-vacuum (UHV) furnace-step-heating results for alunite that have been reported in the literature." This is not true. Polyak et al 1998 Science paper used a furnace to date Carslbad alunite. See another paper by Lin et al 2017. There are numerous other examples. This study is not the first to use a furnace on alunite. The text repeatedly touts the detailed 30+ step heating experiments.  However, upon closer investigation, in most of the experiments in Figs 3-10, the first 12-20 steps comprise only 5-10% of the 39Ar released and the remaining 12-15 steps makeup the rest of the experiment. They have essentially done 12-15 step experiments, which are the exact type of experiments that the authors denounce in this manuscript. Thus, the repeated reference to detailed step heating experiments needs to be downplayed. Moreover, there is a whole section of the manuscript (Discussion 4.5) that sheds a negative light on laser analyses.  On line 145, the text states that the "detailed heating schedule was chosen so that only small amounts of gas are released in each step…..The resultant age spectrum allows vastly more information to be ascertained than any laser spot analysis can provide." This statement is extremely misleading.  I encourage the authors to look at any of the recent work by Anthony Koppers and colleagues on Cretaceous seamounts. Those incremental heating experiments have 30-40 steps and show sufficient detail. There are numerous other examples of detailed laser experiments also. These experiments are typically done on way less than the 100-150mg of material used by the authors in this study.  The entire section 4.5 discounting laser step heating experiments should be omitted because a non-specialist may read this and think that they must use a furnace to step heat alunite and that is simply not true. There are numerous examples of laser step heating of alunite that yield plateaus that make geologic sense.

Lines 124-126: "The method utilised allows protracted cleaning of the furnace between samples, so furnace blanks are consistently reduced to low levels. Corrections are done by interpolation, but in general the blanks are so low that this is not essential."
What is meant by non-essential? Has a blank correction been performed? What is the size of the 36Ar signals relative to the blanks? How many blanks are measured over the temperature range of a typical experiment. The disadvantage of furnace step heating is you cannot assess the blank variability during the experiment and must interpolate blanks before and after the experiment when the furnace is clean. If 100 to 150 mg of alunite is being step heated, the blanks are undoubtedly going to vary during the experiment. Significantly more detail needs to be provided on number of blanks performed, how they vary with temperature, how big they are compared to the signal for each isotope, etc.

Paragraph beginning on line 345: There is more text with a negative tone towards laser methods here. This whole paragraph is misleading and not needed. For example, the text says, "It is rare for a laboratory to have the technical capability to conduct step-heating experiments". Almost all 40Ar/39Ar labs have the capability to conduct step heating experiments. These types of statements are potentially very dangerous to a non-specialist who may be interested in using geochronology for a project.

Flawed data interpretations:
There is no doubt that some samples produce complex apparent 40Ar/39Ar age spectra, which may be due in part to multiple episodes of alteration. The authors propose to see through this complexity by using the asymptotes and limits method of Forster and Lister 2004. When conducting incremental heating experiments, by far the most common convention is to use the plateau age, isochron age, or in some cases an integrated age with caution if the sample does not form a statistically acceptable plateau. The authors have chosen not to use any of these methods and have opted for their own method. The text needs to clearly state that they are not using any common data interpretation methods and have chosen to use the asymptotes and limits method, which is not a conventional approach.

Line 231: "All of the age spectra show evidence of contamination in the first steps, with high initial ages that decrease, often to a small plateau age segment." First of all, these first few steps comprise 5-10% of the 39Ar released. It is by no means a plateau. The word plateau should be omitted throughout the manuscript when talking about these mini segments. The authors are interpreting a few overlapping steps for 5% of the released gas to have significance. There is no age significance to this. Just like there is no significance to the ages later on in most of the experiments that overlap but are sloping downward and are only two steps (blue steps). The probability analysis of all the apparent ages is deeply flawed as many of the steps do not reflect the actual age of the alunite. These apparent ages may be affected by processes such as excess Ar, recoil, K loss, all of which are minimally discussed.

The spectra for each sample look very similar. Almost all of them get younger in the last 20% of the experiment. Coincidentally the Ca/K increase during this time. Yet there is very little mention of recoil. The authors instead attribute the decrease in age at the end of the experiment to mixing with a KCl/CaCl2 gas reservoir derived from inclusions. It is unlikely that all of the samples would be plagued by inclusions. Images of inclusions from each sample would help this argument if they are available. 39Ar recoil needs to be discussed as an alternative.

The isochrons have no statistics (no intercepts with uncertainties, no MWSD, etc) and thus they have little value as presented. The authors state that isochrons are not preferred for these types of samples. The reader cannot evaluate if the isochrons are of value until all of the data associated with them are provided.

There is a whole section on how the authors don't agree with plateau criteria. The authors are entitled to their own opinion. However, numerous published studies have show that alunite can produce plateaus (Corral, 2021, Pan et al 2019, Ren and Vasconcelos, 2019, Mote et al 2001, and many more). Coincidentally most of these studies used a laser. It may be that the Martabe alunite are indeed very complicated. If they are going to argue for multiple generations of fluid interaction, there should be some chemical or mineralogic evidence to support this (overgrowths, zones, etc). The argument cannot be made solely based on complicated Ar spectra.

Line 96-100: These sentences read like they are straight out of a textbook and should be omitted or drastically streamlined. Most readers of GChron are going to be aware of what a fluence monitor is and why it is used.

There are 14 Figures. Ten of them look exactly the same except the data within them are different.  I urge the authors to present a few examples and put the others in a supplement. This will reduce the redundancy of the figures.

---

## Author Comment (AC1)

**REJOINDER**

**Response to reviewer I**

*Gordon Lister (on behalf of the author team)*

Comments by the reviewer are in *red text in italics*. My response is shown in black beneath.

Overall response to reviewer I

Reviewer I is apparently of the opinion that argon spectra should be plateaux, and if not, then the data should be rejected. Yet there is nothing in the theory and practice of argon geochronology that requires an age spectrum to be a simple plateau. It is true that there are many circumstances in which a plateau will result. Equally true, however, is that there are many circumstances in which a plateau will not be evident, nor should one have been expected to have been present. Therefore, the opinions of this reviewer are contentious, and they need to be considered in that light. We are at odds with the overall tenor of this review.

Whereas the analysis of an argon spectrum that resembles a plateau is straightforward, the same is not true for the analysis of an argon spectrum that results from progressive release of gas from different reservoirs. Such circumstances require a rethink as to how complex age spectra are analysed, moving away from the notion that: i) *a priori* a single age is to be expected; and ii) the scatter of ages will define a normal distribution. Such statistical models are overly simplistic, and do not approximate the observed pattern of age variation in many samples, including those reported in this paper. For these reasons, a theory of how to analyze complex age spectra caused by mixing was developed by Forster and Lister (2004). Application of their "method of asymptotes and limits" has been shown to allow analysis of even the most complicated spectra. Frequently Measured Ages (FMAs) as estimated from such analyses have consistently allowed the recognition of (and the replication of estimates for) the timing of potentially significant events in the history of the rocks studied.

*The manuscript by Muston et al present 40Ar/39Ar furnace step-heating and 39Ar diffusion data fromalunite in ten samples from the Martabe epithermal gold deposit in Indonesia. They use their closure temperature constraints to conclude that gold in the Purnama pit was the result of fluid rock interactions from ~ 2.25 and ~2.00 Ma.*
*This manuscript is filled with numerous misleading statements, but more importantly it is plagued by a lack of up to date knowledge of geochronologic methods and flawed data interpretations. For these reasons, which are explained in much more detail below, I strongly urge that this manuscript be rejected.*

It is one thing to claim that a particular group of authors is not up to date with their methods and analysis. It is another thing to reject data that does not conform to the current paradigm. Complex spectra need to be analysed and the results put on display. Therefore, this paper should be published. The submitted paper is a rare example of the detail involved in application of the "method of asymptotes and limits": a method that has been in use for more than a decade, and which has allowed steady progress in developing understanding as to the factors underlying the development of complex argon spectra. The paper is a case study of the issues associated with application of the methods of asymptotes and limits to a real-world problem.

*Suggested remedy*: An additional figure has been produced, in order that the reading public might be reminded of the attendant issues when an age spectrum involves mixing of gas from different reservoirs.

This figure is shown overleaf (on page 2 of this response).

[Figure]

**Figure 1** An age spectrum has a morphology: (a) a plateau forms in the ideal case when the mineral has a single "growth age", or very rapidly cools from elevated temperatures so it retains all radiogenic argon produced subsequently producing a "cooling age"; (b) a mineral aliquot with two growth ages, with the second period of growth producing a microchemistry or a microstructure that is less retentive of argon in the mass spectrometer; (c) the second period of growth producing a microchemistry or a microstructure that is more retentive of argon in the mass spectrometer, but not so different as to prevent mixing between the two domains; (d) the second period of growth involved only a small volume fraction, and produced more scatter in the apparent age during measurement.

Should the mini-plateau be discarded as statistically insignificant? In a Gaussian probability plot, this might be the conclusion: because increased scatter in the mini-plateau reduces the intensity of the age step. If Poisson statistics are utilised, a different conclusion might be reached, because it is highly improbable that a random process will produce several steps with the same age, no matter what the percentage of gas released overall in comparison.

If the two generations of growth produced microstructures and microchemistries that were identical, the age would uniformly mix, and the two generations would not be evident in the age spectrum. If the difference in retentivity is small, a carefully designed step-heating experiment with precise temperature control has the potential to discriminate between the ages of the different growth generations: but (e) mixing might prevent all but a few steps defining the "end member" ages; or worse (f) progressive mixing will result in an age gradient with only the upper and low limits of age having the potential to be statistically significant, and then if, and only if, the same result can be obtained time after time again in different step-heating experiments with different aliquots.

These considerations gave birth to the concept (Foster and Lister 2004) that complex age spectra could be analysed according to a method of asymptotes and limits. Frequently Measured Ages (or FMAs) have the potential to be statistically significant, and worthy of further effort. We need to determine the FMAs and then attempting to elucidate their significance in terms of either the microstructural or the thermal evolution of a sample.

*Lack of knowledge of published studies and use language that could mislead a non-specialist:*
*Line 56: There are numerous other examples of 40Ar/39Ar dating of alunite. All published data on alunite do not need to be cited here, but other work besides Arribas et al should be cited. Vasconcelos1999 provides a nice overview of dating supergene minerals. It should be cited here.*

There is no issue with incorporating additional references, although the reference requested is not at all relevant to the paper. But it is a useful reference for those working on alunite in weathering profiles. The manuscript can be amended so that it is incorporated.

P. M. Vasconcelos, K-Ar AND 40Ar/39Ar GEOCHRONOLOGY OF WEATHERING PROCESSES. Annu. Rev. Earth Planet. Sci. 1999. 27:183–229, doi:10.1146/annurev.earth.27.1.183

*Line 224: "These are the first detailed ultra-high-vacuum (UHV) furnace-step-heating results for alunite that have been reported in the literature." This is not true. Polyak et al 1998 Science paper used a furnace to date Carslbad alunite. See another paper by Lin et al 2017.*

This sentence could be amended to remark instead that previously published age spectra used far fewer steps in the step-heating schedule than shown here, *e.g.,* in the papers cited by the reviewer. Therefore, these previous data lack resolution as to the shape of the spectrum and hence were not referred to as "detailed". The detail in an age spectrum is important, for if it is not sufficient, it will not be possible to separate the effect of 'mixing' as opposed to the effects of random fluctuations of age, involving larger gas percentages.

*There are numerous other examples. This study is not the first to use a furnace on alunite.*

The reviewer is correct, but on the other hand, neither did the submitted manuscript claim to be the first paper to use step-heating. Nevertheless, this remark can be clarified, and the paper be amended to cite other papers, for example: POLYAK, V.J., MCINTOSH, W.C., GÜVEN, N. AND PROVENCIO, P. Age and Origin of Carlsbad Cavern and Related Caves from 40Ar/39Ar of Alunite. Science 279, 1919 (1998), DOI: 10.1126/science.279.5358.1919

*The text repeatedly touts the detailed 30+ step heating experiments. However, upon closer investigation, in most of the experiments in Figs 3- 10, the first 12-20 steps comprise only 5-10% of the 39Ar released and the remaining 12-15 steps makeup the rest of the experiment. They have essentially done 12-15 step experiments, which are the exact type of experiments that the authors denounce in this manuscript. Thus, the repeated reference to detailed step heating experiments needs to be downplayed.*

The reviewer is not correct. The manuscript needs to make these points better perhaps: i) whereas only small gas volumes may be released in those first steps, they are key in terms of developing an Arrhenius diagram spanning as large a temperature range as possible; ii) while it is not necessary to consider such low volume release steps when defining the slope on the Arrhenius plot, they are of interest and have been considered.

Consider the Arrhenius data in conjunction with the age spectrum: typically, the first 6-8 steps have not been illustrated, because the volume of $^{39}$Ar released in each of those steps was less than 0.15% of the total $^{39}$Ar released in the diffusion experiment. This is a somewhat arbitrary but conservative cutoff because meaningful information still appears to be present, despite the low volume fraction involved, at least for steps that release > 0.01% of the total $^{39}$Ar volume. Similarly, in considering the York Plot, there seems to be an abundance of potentially useful information in these low release steps. Thus, although low-release steps were not plotted on these diagrams, their inclusion would have revealed that they were not inconsistent with the later release pattern. Importantly these early results can be compared with those from later in the UHV diffusion experiment when higher temperatures were involved. The results are consistent, confirming that the low release steps were significant. An expanded Arrhenius Plot is shown for one sample on page 12.

The reviewer should also note that the analytical solutions relating fractional loss to normalized diffusivity require: i) accurate knowledge of the gas release percentage in each step, which brings into question the accuracy of experiments that deal with very small fraction of the total gas release; and ii) heating methods that force temperature-time variation during an individual heating step to accurately approximate a square wave.

The reviewer mistakenly asserts that "*the first 12-20 steps comprise only 5-10% of the $^{39}$Ar released and the remaining 12-15 steps makeup the rest of the experiment*". This is also not correct, as is evident in the York Plots and Arrhenius plots that have been provided. In each case it is the last 9-10 steps that are not shown, because the errors in these steps have become unacceptably large and/or because the percent of the total $^{39}$Ar released in an individual step has become small (<0.05%).

An additional figure is suggested, for inclusion in the supplementary data, to make it clear how the volume of gas released in each step varies with temperature. At temperatures below 450°C very small volumes of gas will be released. Heating times could be lengthened, but then the likelihood of microstructural change is also increased. Significant errors in diffusivity estimates are also more likely once the volume of $^{39}$Ar released decreases below 0.05% of the total volume present in the sample.

[Figure]

$^{39}$Ar Release Pattern vs Temperature

*Moreover, there is a whole section of themanuscript (Discussion 4.5) that sheds a negative light on laser analyses.*

A previous reviewer questioned our efforts with the furnace, which led to these comparisons. We emphasize that it is important to differentiate laser heating with little or no temperature control from step-heating in which there is some measure of temperature control. There is a need for more accurate reporting of methodology. Nevertheless, our intent was not to shine a negative light on laser analysis in general.

*On line 145, the text states that the "detailed heating schedule was chosen so that only small amounts of gas are released in each step ….. The resultant age spectrum allows vastly more information to be ascertained than any laser spot analysis can provide." This statement is extremely misleading.*

I am not sure why the reviewer thinks that incremental heating experiments are equivalent to laser spot analyses, unless the reviewer's intent is to imply that step-heating can include laser spot analyses where the same spot in a microstructure is reheated several times.

Laser spot analysis refers to the practice of using a laser while exploring a microstructure while zapping individual locations to yield a single age analysis. This is not step-heating. Nevertheless, we can amend the offending remarks, or delete them, since they detract from the point and focus of this paper.

The way forward is to emphasize that more precise descriptions of methodologies are in order. Clearly, based on the comments by this reviewer, "step heating" has come to mean many different things. The manual roaming of a laser over a sample with gradually increasing intensity and/or duration of heating should *sensu stricto* not be described as step-heating. Nor should 'rastering' be used as a terminology to describe anything other than a machine-driven operation, since otherwise the trajectory of the laser is arbitrary and undefinable.

For the avoidance of doubt, and to avoid unintentionally misleading the "non-specialist" as an alternative we suggest "temperature-controlled step heating" to be used as a terminology to describe step-heating experiments that have the capacity to accurately measure the temperature of the sample during each heating step. In this case, as a matter of routine, we recommend that data such as temperature during each step and the duration of the heating step should be included in published datasets.

*I encourage the authors to look at any of the recent work by Anthony Koppers and colleagues on Cretaceous seamounts. Those incremental heating experiments have 30-40 steps and show sufficient detail.*
The revised manuscript will remove unintended criticism of laser-driven "incremental step heating" noting that a laser can be used to heat a receptacle or crucible so that sequential temperatures increment, and so that the rise and fall of temperature prior to each heating step approximates a square wave.

We will report Koppers et al. 2004 as demonstrating experiments with "sufficient detail" to allow the application of the method of asymptotes and limits (e.g., when multiple limits are present in the age spectra), or in other cases, analysis of a plateau as though a single Gaussian distribution is present.

Anthony A. P. Koppers, Robert A. Duncan, Bernhard Steinberger, 2004. Implications of a nonlinear $^{40}Ar/^{39}Ar$ age progression along the Louisville seamount trail for models of fixed and moving hot spots. Geochemistry, Geophysics, Geosystems, doi:10.1029/2003GC000671

*There are numerous other examples of detailed laser experiments also. These experiments are typically done on way less than the 100-150mg of material used by the authors in this study. The entire section 4.5 discounting laser step heating experiments should be omitted because a non-specialist may read this and think that they must use a furnace to step heat alunite and that is simply not true.*
The overall message seems to be not to detract from the main thread of the paper by digressing into comparisons with other methodologies. A laser can indeed be used to step-heat a crucible.

*There are numerous examples of laser step heating of alunite that yield plateaus that make geologic sense.*
Of course, an argon spectrum can be a plateau, and we have said nothing different. A plateau might result because a single growth age was involved, as per the additional figure we have included on page two, or because of rapid cooling. But is also the case that laser heating with no attempt to ensure uniformity of temperature may yield a pseudo-plateau because the degassing was homogenized.

We have seen that it is possible to step-heat with a laser in such a way that different grains get subjected to different temperatures, for different times, and in a random and non-uniform way. The net result can be a plateau, but such a plateau is methodological rather than reflecting any intrinsic property of the sample. It is misleading to report such experiments as "step heating" and that is the point we are trying to make. A methodological plateau may result because of chaos and random mixing caused by the way grains were heated, rather than the plateau being an intrinsic property of the aliquot of mineral grains analysed.

There are ways in which to step heat with a laser so that precise temperature control is assured, in which case it matters not whether a resistance furnace is used or not. We agree, but that is not the point we try to make. The problem is that the use of the term "step heating" has become loose and can mean too many different things. We will go over this aspect of the paper carefully, to try and improve the message.

*Lines 124-126: "The method utilised allows protracted cleaning of the furnace between samples, so furnace blanks are consistently reduced to low levels. Corrections are done by interpolation, but in general the blanks are so low that this is not essential."*

*What is meant by non-essential? Has a blank correction been performed? What is the size of the 36Ar signals relative to the blanks? How many blanks are measured over the temperature range of a typical experiment. The disadvantage of furnace step heating is you cannot assess the blank variability during the experiment and must interpolate blanks before and after the experiment when the furnace is clean. If 100 to 150 mg of alunite is being step heated, the blanks are undoubtedly going to vary during the experiment. Significantly more detail needs to be provided on number of blanks performed, how they vary with temperature, how big they are compared to the signal for each isotope, etc.*

The procedure is detailed in the supplementary information. The manuscript text will be modified.

The reviewer is correct in that a blank cannot be measured while the sample is still in the furnace, but this is true no matter how a step-heating experiment is conducted (either using a furnace or a crucible heated with a laser) so the question of 'inheritance' from heating step to heating step remains an open one. However, since each step is small, inheritance from the previous steps will merely smooth observed variations.

By undertaking many steps we were able to routinely demonstrate that the experiment starts and ends with isotopic ratios typical of air, so we can assume that: i) the furnace has been cleaned by the end of the experiment; ii) inheritance between experiments is not greater than the blank. An additional figure has been included to make this point (shown below and a York Plot tracking isotopic ratios, as on page 33 of this rejoinder). Note that small samples as suggested by the reviewer often yield variability and error because of their size. Larger sample sizes guarantee a high signal to noise ratio once significant release of gas commences.

[Figure]

*Paragraph beginning on line 345: There is more text with a negative tone towards laser methods here. This whole paragraph is misleading and not needed. For example, the text says, "It is rare for a laboratory to have the technical capability to conduct step-heating experiments". Almost all 40Ar/39Ar labs have the capability to conduct step heating experiments.*

The basic requirement for a "temperature-controlled step heating diffusion experiment" is that Arrhenius data can be derived from each step of the experiment. The manuscript does not need to comment on why this capability appears to be rare, except to note that if a laboratory does not have the capacity to produce Arrhenius data, there are a limited number of reasons why this would be the case.

These reasons include:
- I) temperature control was not precise, so step-heating data would lack precision;
- II) volume calibration was not undertaken, so sample variability could not be considered;
- III) the use of the "step heating" epithet was inappropriate (e.g., manually roaming a laser with gradually increasing intensity and/or duration of heating is not step-heating).

*These types of statements are potentially very dangerous to a non-specialist who may be interested in using geochronology for a project.*
There are ways in which to step heat with a laser so that precise temperature control is assured, in which case it matters not whether a resistance furnace is used or not. We agree, but that is not the point we try to make.

The problem is that the use of the term "step heating" has become loose and can mean too many different things. We will go over this carefully, to try and improve the message. To avoid misleading a non-specialist, we suggest a need for more careful use of terminology. Methodology needs to be properly explained, to avoid mismatch between expected outcomes and the results that will eventually be obtained. *Caveat emptor.*

*Flawed data interpretations:*
The reviewer failed to demonstrate (or identify) flaws in our methods.

*There is no doubt that some samples produce complex apparent 40Ar/39Ar age spectra, which may be due in part to multiple episodes of alteration. The authors propose to see through this complexity by using the asymptotes and limits method of Forster and Lister 2004.*
We have demonstrated that use of the method of asymptotes and limits (Forster & Lister 2004) suggest the presence of Frequently Measured Ages (FMAs) which in themselves then need independent assessment. This was made possible because these step-heating experiments were able to produce detailed spectra, because they had many steps, and the multitude of steps enabled use of "the method of asymptotes and limits".

The analysis presented in this paper goes further: the inferences made from the age spectra using "the method of asymptotes and limits" are then examined in the context of the Arrhenius data, which precisely defines different domains with distinct diffusion parameters (at least in the mass spectrometer), and in the context of the York Plots, which show the progression of mixing between distinct reservoirs as the experiments proceed.

These independent parameters were then related back to the interpretation of the age spectra, making it likely that the observed variation reflects distinct diffusion domains (in different growth generations) in the natural environment. Irrespective of the estimates provided by the UHV diffusion experiments, these different domains had sufficiently different diffusion properties to allow sequential release and progressive mixing of argon gas from these different reservoirs during the step-heating experiment, thus reflecting real age variation.

*When conducting incremental heating experiments, by far the most common convention is to use the plateau age, isochron age, or in some cases an integrated age with caution if the sample does not form a statistically acceptable plateau.*
The notion that argon geochronology should yield a single age is not a convention. It is an assertion, a mistaken belief that can be propagated only by rejecting any dataset that does not conform to this model.

*The authors have chosen not to use any of these methods and have opted for their own method. The text needs to clearly state that they are not using any common data interpretation methods and have chosen to use the asymptotes and limits method, which is not a conventional approach.*
Again, this statement is misleading and incorrect. It has no theoretical or practical justification.

We did not opt to use one optional method over preferred use of a 'plateau' method. If a spectrum does not define a plateau, then the use of methods that require a plateau to be present is not an option. Since the spectra in question are not plateaux, they cannot be analysed using conventional methods that assume plateaux should be present. The analysis of mixtures requires an alternative method, *e.g.*, as applied here.

*Line 231: "All of the age spectra show evidence of contamination in the first steps, with high initial ages that decrease, often to a small plateau age segment."*

The reviewer is only in part correct. The initial $^{39}$Ar data are precisely as to be expected if release of argon is proceeding via solid-state diffusion. However, the pattern of $^{39}$Ar release in these early steps in part mimics that in the steps that follow and is discordant to the pattern of $^{40}$Ar release. The fact that the $^{40}$Ar is discordant points to the presence of extraneous argon, which can indeed be regarded as a 'contaminant'. Only once the heating steps have reduced this 'contamination' of non-lattice argon, do the age data become useful.

*First of all, these first few steps comprise 5-10% of the 39Ar released. It is by no means a plateau.*

The reviewer is bound by a definition that *a priori* prevents the recognition of "mini-plateaux".

*The word plateau should be omitted throughout the manuscript when talking about these mini segments. The authors are interpreting a few overlapping steps for 5% of the released gas to have significance. There is no age significance to this.*

The manuscript uses the term 'plateau' in reference to segments, as per the definitions ensconced in Forster & Lister (2004). If the use of the term is confusing the revised manuscript could refer to 'plateau segments' though in truth this changes nothing and is an entirely semantic remedy. Nevertheless, the revised manuscript will explain that an arbitrary definition of a plateau is limiting and unnecessary. Based on "the method of asymptotes and limits" there are indeed plateau segments, and these yield ages often in accord with other Frequently Measured Ages (or FMAs). The theory and practice of argon geochronology can only be improved by attention to such data, and not by their exclusion from the published literature by those who insist that data and interpretations that deviate from those appropriate to a simple plateau are somehow flawed.

*Just like there is no significance to the ages later on in most of the experiments that overlap but are sloping downward and are only two steps (blue steps).*

The reviewer's misuse of statistical arguments reflects a particular bias, as does the recommendation made that 'disturbed' spectra are not to be published. We disagree with the latter course, preferring instead to try to understand the reasons for complexity. Probability can be calculated, but only on the basis of an assumed statistical model. If there is good reason to assume **a priori** that a sample should exhibit a single age, then the scatter of the mean (for each individual measurement) can be compared using Gaussian statistics. In this case there would be no meaning to such a statistic. It is improbable that multiple steps should reflect the same age unless there is some statistical significance, and this remains true no matter what percentage of gas is released.

*The probability analysis of all the apparent ages is deeply flawed as many of the steps do not reflect the actual age of the alunite. These apparent ages may be affected by processes such as excess Ar, recoil, K loss, all of which are minimally discussed.*

The reviewer asserts that many of the steps do not reflect the actual age of the alunite. The issue is that *a priori* the reviewer assumes that each sample has a single age.

This sentiment becomes evident by virtue of the claim that the "*probability analysis of the apparent ages is deeply flawed as many of the steps do not reflect the actual age of the alunite*". Logically speaking this is an example of circular reasoning. The reviewer asserts the sample has a single 'actual age' and therefore the analysis is flawed because the assumption is not supported by the data. This is not a good way to analyze data.

An alternative approach is illustrated in the figure on page two of this rejoinder.

We do discuss the issues of excess argon (as recognized in the York Plots). We also discuss (and dismiss) the effects of recoil, which principally affects only the initial steps of a step-heating experiment, because thermal activated neutrons are involved.

*The spectra for each sample look very similar. Almost all of them get younger in the last 20% of the experiment. Coincidentally the Ca/K increase during this time. Yet there is very little mention of recoil. The authors instead attribute the decrease in age at the end of the experiment to mixing with a KCl/CaCl2 gas reservoir derived from inclusions. It is unlikely that all of the samples would be plagued by inclusions. Images of inclusions from each sample would help this argument if they are available.*

The ages decrease at the end of the step-heating experiment. The York Plots show the location of this gas reservoir, which requires K, Cl and Ca to be present in small quantities (<1%). The alunite may contain cracks or inclusions with $CaCO_3$, $CaCl_2$ and KCl. The reservoir does not have a zero age, but the age plot behaves as though it does because calculation of the apparent age spectrum assumes mixing with air, and the gas reservoir that we think results from the inclusions plots above the assumed value for the atmospheric $^{36}Ar/^{40}Ar$ ratio.

The shape of the age spectra require mixing with small quantities of gas released from this reservoir, and we can use an 'inverse isochron' to estimate the age of the potassium bearing mineral within. This was made evident by analyses of individual York plots, e.g., in the additional figure in the next section.

Since the effect begins once the melting temperature (770°C) of KCl or $CaCl_2$ has been reached, we surmise these minerals may have been present in small quantities, in inclusions. Corrections for isotopic interferences have been carried out (see supplementary information) yet these effects remain. Therefore, we conclude that chlorine has been released from the sample and found its way into mass spectrometer, thereby artificially increasing the $^{36}Ar$ signal. Yet the getters were clean, and copper and silver wool had been included in the extraction line. The three Zr-Al getters included one large volume CP50 getter, and two AP10 getters, one hot and one cold. Further investigation into this matter is warranted. Observed Ca/K and Cl/K ratios can be explained by combining $CaCO_3$, $CaCl_2$ and KCl (*e.g.*, in the ratio 2:2:1 in the example illustrated on page 33).

More information as to the nature of inclusions would have been beneficial, but these were difficult samples, and it was difficult to separate the alunite. The available samples were small and once the limited areas of localized alunite had been removed, no alunite areas remained to be used to make thin sections of key areas.

*39Ar recoil needs to be discussed as an alternative.*

The effect of recoil is likely to be found in the early steps. We did not find significant effects. Neither did Ren & Vasconcelos (2019).

*The isochrons have no statistics (no intercepts with uncertainties, no MWSD, etc) and thus they have little value as presented. The authors state that isochrons are not preferred for these types of samples. The reader cannot evaluate if the isochrons are of value until all of the data associated with them are provided.*

This is a case of circular reasoning. An inverse isochron is no more than a test of the assertion that a group of steps define a single age. However, the supposed process of evaluation includes the requirement that the sample yields a single age. Providing detail such as intercepts and uncertainties is just another way of asserting belief in a model that *a priori* assumes the existence of a single age population with scatter normalized according to a normal distribution. We re-iterate that we illustrate nothing more than bounding lines that help us understand the physics of mixing gas released sequentially from different microstructural reservoirs.

There is no guarantee that an alignment demonstrates the existence of an isochron. The nature of mixing is *a priori* asymptotic. Therefore, we have avoided the misleading imposition of use of Gaussian statistics in this regard. By defining a line of best fit between a set of points that defines a limit or an asymptote, one has to be careful that one does not simply impose an age on a set of points with isotope ratios that vary systematically because of the way that mixing between reservoirs was taking place.

Nevertheless, the manuscript will be revised to include estimates of these uncertainties, and goodness of fit. We use the revised York (1969) regression that was proposed by Mahon (1996) and implemented in python by Trappitsch (2018). Although our C++ and Objective-C code was independently developed, we have followed Trappitsch (2018) in correcting a mistake in one of the equations written down by Mahon (1996), and in respect to the resolution of typographical ambiguities in Mahon (1996).

*There is a whole section on how the authors don't agree with plateau criteria. The authors are entitled to their own opinion.*
Some age spectra have successions of segments that resemble the shape of a plateau. But there are also a multitude of age spectra that contain "mini-plateaux" or which have more complex morphologies.

Such complex age spectra are typified by the existence of asymptotes and limits (suggesting mixing of gas released from different reservoirs is an important process). Hence such spectra need to be analysed differently (and not by the imposition of Gaussian statistics which *a priori* requires scatter about a single age).

*However, numerous published studies have show that alunite can produce plateaus (Corral, 2021, Pan et al 2019, Ren and Vasconcelos, 2019, Mote et al 2001, and many more).*
There is no reason why an age spectrum should not have a plateau morphology. All that is required is a single growth age, or a rock that was hot and which thereafter cooled very rapidly.

On the other hand, we need to avoid describing data too simply. By describing the Ren & Vasconcelos (2019) data as 'plateaux' the reviewer thereby defeats his/her own argument.

The data published by Ren & Vasconcelos (2019) include spectra with multiple plateau segments, for example resembling spectrum (b) in the figure on page two of this rejoinder.

*Coincidentally most of these studies used a laser.*
It is perhaps no more than a worrying coincidence that most of these studies used a laser.

We have drawn attention to the possibility that a plateau can also be methodological in its origin. We have seen that it is possible to step-heat with a laser in such a way that different grains get subjected to different temperatures, for different times, and in a random and non-uniform way. The net result can be a plateau, but such a plateau is methodological rather than reflecting any intrinsic property of the sample.

*It may be that the Martabe alunite are indeed very complicated. If they are going to argue for multiple generations of fluid interaction, there should be some chemical or mineralogic evidence to support this (overgrowths, zones, etc). The argument cannot be made solely based on complicated Ar spectra.*
The research would have been much improved if we had indeed been able to provide additional information as suggested. But there is no logical reason why such an argument should not be made based on the analysis of these complicated argon spectra, as presented here. Our hypothesis can now be tested.

It may be indeed the case that the Martabe alunite grew in multiple events, and therefore that these fluid systems were thus geochemically at least in part, independent. This is perhaps why attempts to vector fluid flow based solely on geochemistry (in the absence if geochronology) were doomed to failure.

Line 96-100: These sentences read like they are straight out of a textbook and should be omitted or drastically streamlined. Most readers of GChron are going to be aware of what a fluence monitor is and why it is used.
These sentences are not necessary: we will eliminate them.

*There are 14 Figures. Ten of them look exactly the same except the data within them are different. I urge the authors to present a few examples and put the others in a supplement. This will reduce the redundancy of the figures.*
The reviewer asserts that ten figures are exactly the same. True, in each of those figures, the same set of plots is presented. For ease of comparison the computer program has set the axes to have the same range. So, the figures do look exactly the same, if one considers only the graph frames and the labelling on the axes.

However, in terms of their content, the figures are different. The data presented in each figure is different, and each figure contains interesting if subtle variations. The figures deserve publication, in particular because this paper is a rare example showing the detail involved in the application of the method of asymptotes and limits.

**Response to reviewer II**

Comments by the reviewer are in *red text in italics*. My response is shown in black beneath.

Overall response to reviewer II

This reviewer rightly emphasizes the importance of the Ren & Vasoncelos (2019) paper. This work appeared after our original paper was submitted. Nevertheless, we agree, we should have considered the implications of the results in more detail. Therefore, the revised manuscript will be modified.

Overall, the results obtained by these authors requires discussion on three points:

i)     despite repeated claims made by this reviewer, Ren & Vasconcelos (2019) did not demonstrate a phase transition. They demonstrated a scattering of orientation and a sharp reduction in grain size that was more or less coincident with the beginning of a change of slope on the accompanying Arrhenius Plot.

Our results also demonstrate a change in slope on the Arrhenius Plot, but this occurs at temperatures 80-100°C hotter than for the samples examined by Ren & Vasconcelos (2019). If indeed a microstructural change is occurring in our samples, one might therefore surmise that it began to occur at higher temperatures than in the Ren & Vasconcelos (2019) samples (see additional figure below).

ii)    Ren & Vasconcelos (2019) made some mistakes in analyzing their Arrhenius data, in particular because no attempt was made to apply the Fundamental Asymmetry Principle (or FAP). This means that the estimates obtained by these authors will be overwhelmed by the effects of mixing. Therefore, it is possible that the values 'preferred' by Ren & Vasconcelos are artefacts. Mathematically there is no coincidence between temperature, diffusivity, and the average slope of a progression of individual points on an Arrhenius plot. Estimates of diffusion parameters made on this basis should be discarded.

In addition, we note that the potential for significantly over-estimating diffusivity because the time-temperature variation using in a diffusion experiment differs from that expected in a Dirac function. Ren & Vasconcelos (2019) acknowledge the limitations of their experimental setup in this regard. The higher estimates they obtained for normalized frequency factor are thus at least in part to be expected.

iii)   Reviewer II argues that phase changes or microstructural changes may potentially occur during a step-heating experiment, with which sentiment we agree, but Forster & Lister (2015) already showed that this possibility does not preclude local application of the FAP.

Significantly, one can graphically assess FAP violations in the diagrams published by Ren & Vasconcelos (2019) and exclude these data in making a comparative analysis of the results in their paper with those that we have obtained (see additional figure below).

It is evident that the most accurate estimates of activation energy coincide, *i.e.*, Ren & Vasconcelos (2019) obtained the same results as we did. However, the diffusion dimension in our experiments is ~700 times that which applied in the Ren & Vasconcelos (2019) experiments.

Since the episodes of grain size reduction that they observed may be the effects of nano-scale thin slices of the alunite lattice buckling and kinking in the electron beam during transmission electron microscopy, it is possibile that our results were not subject to such grain size reduction, thus explaining the observed difference in diffusion dimension.

We repeat that there is the potential for significantly over-estimating diffusivity if the time-temperature variation used in a diffusion experiment differs from that expected in a Dirac function. Ren & Vasconcelos (2019) acknowledge the limitations of their experimental setup in this regard. The higher estimates they obtained for normalized frequency factor are thus at least in part to be expected.

[Figure]

Figure shows all FAP compliant estimates of diffusion parameters made by Ren & Vasconcelos (2019) [blue squares], to be compared with estimates made here [red circles]. The activation energy is the same. Normalized frequency factors differ so either the samples used in the Ren & Vasconcelos (2019) suffered a grain size reduction during the experiment, or the diffusion dimension was inherently much smaller.

[Figure]

Figure shows an extended Arrhenius Plot, showing the same change of slope as documented by Ren & Vasconcelos (2019) but at temperatures 80-100°C hotter than in their experiments. We suggest the change in slope is the result of mixing, with an increasing proportion of lattice argon as release from a non-lattice [39]Ar reservoir causes it to be progressively exhausted.

***General Comments (made by reviewer two)***

*This manuscript describes temperature-controlled incremental heating $^{40}Ar/^{39}Ar$ results forten samples of alunite (ideal endmember: $KAl_3(SO_4)_2(OH)_6$) from the Martabe epithermal Au deposit in Sumatra, Indonesia. While the quality of the language and figures in the paper is generally high, I have several concerns that range in severity from fundamental to minor with regards to the applicability of the results and the clarity with which topics are discussed. I will briefly summarize my chief concerns here, which are described in greater detail in the following sections of this review along with more moderate and minorconcerns that mostly revolve around improving evidence for some ancillary interpretations (e.g., Cl contaminants hosted in fluid inclusions) and increasing the clarity of the paper.*

We are grateful for the efforts of this reviewer, in particular because this stimulated additional analysis.

*First, the authors report using heating schedules starting at 450 C in all of their experiments, despite citing recently published work by Ren and Vasconselos (2019) who demonstrated with ultra-high vacuum (UHV) transmission electron microscope experiments that alunite undergoes a structural transformation at temperatures of 430–460°C. Below these temperatures under UHV, alunite retains its natural crystal structure; but, above these temperatures, it becomes a polycrystalline aggregate with a potential change in structure due to dihydroxylation to $KAl(SO)_4$ (alum) + $Al_2O_3$ aggregates.*

The starting temperature in our experiments was chosen because very low $^{39}Ar$ release percentages were observed at lower temperatures. We have included a figure in the supplementary material to demonstrate this aspect. In retrospect, even so, it would have been better to have commenced the heating schedule at 400°C, and then incremented each step by 10°C.

Nevertheless, in our results, we note alignments in the Arrhenius Plots that can be discerned even at the low $^{39}Ar$ release percentages that occur in the first ten steps of our experiments (with each increment of temperature equal to 10°C). Even though low gas release percentages were involved, these data extrapolated smoothly into data measured at higher temperatures, and the results obtained were similar (or the same) as reported by Ren & Vasconcelos (2019) for the few Arrhenius plots that these authors analysed in ways consistent with the Fundamental Asymmetry Principle (or FAP). Other estimates by these authors could well be the result of mixing, or incorrect because the line of best fit did not comply to the FAP. In any case, many different (and indeterminate) estimates of activation energy can be made in the zone where the effects of mixing between two subsequently released dominant gas reservoirs are evident.

We note that Ren and Vasconcelos (2019) report only ONE sample that broke down into different phases.

The other samples changed their microstructure (into nano-crystalline aggregates) but retained the same lattice parameters (e.g., d-spacing), which we suggest implies microstructural disordering, shifting relatively unaffected domains relatively to each other such as occurs during a Penrose tiling. The lattice may simply have kinked, thus enabling argon to be retained (as observed). This is not a phase change, at least as far as the term is applied in the materials science literature. Presumably the diffraction pattern changed into circular arcs because electron channeling is affected by the breakdown into nano-crystals, with the effects of misorientation exacerbated by the trigonal (not hexagonal) symmetry of the R3m space group.

It is certain that not all our samples have behaved in the same way, and the same can be said for samples analysed by Ren & Vasconcelos (2019). The behaviour they report from a nano-slice under an electron beam is not of necessity what will happen in an unexcited sample with considerably larger dimensions.

Nevertheless, although these authors made some mistakes in how their Arrhenius data were analysed, when the debatable results are set aside, the results are in the same range (with activation energies in the range 90-110 kJ/mol). This further seems to suggest that the transformation witnessed by these authors is

microstructural, and not a phase change as asserted by the reviewer. The d-spacing did not change, but like in a powder camera, the Bragg scattering defined circular arcs, implying a scattering of orientation. This is perhaps not surprising given that a nanofilm was being examined and energised in the electron beam.

*Ren and Vasconselos documented an increase in activation energies and pre-exponential constants (e.g., $ln(D_0/a^2)$) above this phase transition temperature (430–460 C), which can appear as upwards curvature on Arrhenius plots, and concluded that lower temperature data must be used to derive diffusion kinetics representative of natural alunite. Therefore, the results presented in this paper may not be relevant to the natural conditions of the Martabe epithermal Au deposit. Repeated experiments that include lower temperatures are required to evaluate this possibility for the Martabe sample set.*
Ren and Vasconcelos (2019) report only ONE sample that broke down into different phases. The other samples changed their microstructure (into nano-crystalline aggregates) while retaining the same d-spacing. This is not a phase change, at least as far as the term is applied in the materials science literature.

The reviewer should note that the FAP is developed for a MDD model that maintains constant activation energy across all diffusion domains. Forster et al. (2015) <http://dx.doi.org/10.1080/08120099.2015.1114524> detailed how 'local' application of the FAP is implemented and showed how variation in activation energy explains upward curvature in the Arrhenius Plot. The FAP must be applied 'locally' on either side of the mixing zone where one slope changes to the other. We need not repeat this information: the revised manuscript will draw attention to the 2015 paper.

*Second, the Fundamental Asymmetry Principle (FAP) is not described in the context of multi-domain diffusion (MDD) theory with sufficient clarity, and there is only a single statement noting a fundamental limitation of the method: that it cannot be applied in the presence of microstructural changes. Because the results presented by the authors are all effectively at higher temperatures than the phase change documented by Ren and Vasconselos, the data appear to follow expected MDD behavior in Arrhenius plots and the FAP was applied in analyzing the data.*
The revised paper will direct the reader to the papers where 'local' application of the FAP has already been described in detail. In our analysis of our Arrhenius data, the FAP was locally applied, and consistent results were obtained. Ren & Vasconcelos (2019) did not apply this principle in their analysis, so many of their results are debatable. Mixing can be called upon to produce the estimates shown by red lines in their figure 6, in most cases. Estimates from the mixing zone violate the FAP and so should be discarded. They are not valid estimates.

In any case the supposed microstructural changes have already occurred. The values shown by the dotted black lines are not affected by FAP violations. Interestingly, these results differ not at all from our own.

Interestingly, while the supposed microstructural changes have already occurred, they have occurred for both the estimates made using red lines, and the estimates using dotted black lines. There is no consistency to the argument that the later determined values should be discarded. The earlier values may be the result of mixing.

*However, the authors also claim to have re- evaluated the results reported by Ren and Vasconselos by applying the FAP, despite the documented phase change in alunite that is apparent in the Arrhenius plots of Ren andVasconcelos as upwards curvature (due to increased activation energies at higher temperatures). Thus, by their own admission this appears to be an inappropriate application by the authors of the FAP in the presence of a microstructural change.*
First, to say that phase changes have been documented is a misuse of a well understood terminology, despite assertions made by the authors and the reviewer. The lattice parameters did not change. Grain size reduced.

Second, the reviewer misrepresents the applicable circumstances. The microstructural changes have already occurred, before the change in slope. These results are the result of mixing, and estimates mode in the zone of mixing are indeterminate. These estimates should therefore be discarded.

*Third, the supplementary data and other materials were not made available to this reviewer, and no evidence was made available to evaluate how the authors reprocessed the results of Ren and Vasconselos.*

The submitted version of the paper explained that the estimates made by Ren & Vasconcelos (2019) were reappraised graphically. The reviewer wants us to re-examine that data and do this differently. However, there is no need for us to do this. Ren & Vasconcelos (2019) also included some FAP compliant estimates, and these differ little from our own. At least in terms of the estimated activation energy.

 A separate plot has now been included to make this more evident [page 12].

*This could have been due to mishandling of the supplementary materials, or perhaps the supplement was not submitted with the preprint.*

We apologise that the supplementary data was not submitted with the paper.

*In either case, this severely limited my ability to evaluate claims that the closure temperatures the authors derived by reprocessing the data reported by Ren and Vasconselos were in agreement with those reported by the authors.*

The paper did contain the necessary data: namely activation energy and normalised frequency factor. All that is needed then is a simple recursion in an Excel sheet, for example.

*From my own calculations of closure temperature using the diffusive parameters reported by Ren and Vasconselos (even for data above the phase transition temperature of alunite), I was unable to reproduce the high closure temperatures reported by the authors, which are generally higher than those of other authors (e.g., 150–300 C reported by Love et al., 1998, Landis et al., 2005, and Waltenberg, 2012).*

However, the reviewer is correct in that it is the activation energies that coincide, not the closure temperatures. A separate plot has now been included (on page 13) to make this more evident. Examination of these data shows that the lower group of our estimates coincide with the few FAP compliant estimates of activation energy made by Ren & Vasconcelos (2019). It is the normalised frequency factor that differs.

*The lack of access to supplementary material also made it impossible to evaluate statistical methods (e.g., the handling of uncertainties when forming weighted means from data subsets with low degrees of freedom, the computation of 95% confidence intervals, regression methods, etc.) and whether the authors followed community data reporting standards (e.g., Renne et al., 2009; full reference: Renne, P.R. et al., 2009, Data reporting norms for $^{40}Ar/^{39}Ar$ geochronology: Quaternary Geochronology, v. 4, p. 346–352, doi:10.1016/j.quageo.2009.06.005.).*

The paper uses methods documented by McDougall and Harrison in their text, which methods are incorporated in the Noble program developed by McDougall and in operation in the ANU laboratory for some decades. More information is provided in the supplementary information.

We agree with the concern expressed in respect to the difficulty in respect to evaluating statistical methods, so some revision of the manuscript is necessary in this aspect, to better document these methods.

Weighted means and regressions will be recalculated using formulae from Mahon (1996) who corrected some errors in York (1969). Our exploration of the formulae published by Mahon (1996) revealed some typographic errors. Our calculations are consistent with those of Trappitsch (2018) who appears to have come to the same conclusion, as well as identifying an error in one of the formulae for 'true' error in Mahon (1996).

*The general lack of clarity on statistical methods should be addressed, either by adding detail in the main text or supplement (if it is not there already), or by making more explicit reference to previous publications that provide sufficient descriptions of the statistics employed by the eArgon software used by the authors. I looked for publications describing the inner workings of eArgon, but could not readily find any; this software should not be a "black box" to the readers of this paper.*

This information will be included in the revised supplementary information. *eArgon* includes various options, which have been expanded to include weighted means and regressions using revised formulae from Mahon (1996). The data tables used in the original publication included methodological uncertainty in the age estimates, including uncertainty in the J-factor determination, as well as uncertainty in the estimated age for the fluence monitor. Such systematic uncertainty applies in a similar way to the entire age spectrum, so it should be removed prior to performing goodness of fit estimates, and then reapplied thereafter.

This will be done for the revised manuscript, including appropriate documentation.

*Finally, while the paper was purportedly focused on the hydrothermal history of the Martabe Au deposit, I found that the discussion was too heavily focused on experimentaland analytical procedures. Indeed, by my count only about 333 words out of a total of 3683 (just under 10%) in the discussion section appear directly related to discussing thetopic of the paper. While methods are important for this kind of work, there should be a larger fraction of the discussion section devoted to the Martabe deposit and implications for economic geology.*

Agreed, we can do this. Two additional figures are suggested, to allow the results to be put into a broader tectonic context. The first figure shows the larger-scale structural context of the Martabe deposit. The second figure shows the repeated history of stress cycling that could explain multiple short-lived fluid pulses.

The island of Sumatra lies in a northwest-southeast orientation adjacent to the Sunda subduction zone, which has a similar orientation, but ~200km from the western coast of the island. Oblique subduction causes partitioning of slip onto trench parallel strike-slip systems, with the Sumatran Fault System being the most active in recent times. However, this fault system has at least 12 step-overs, with offsets greater than one kilometre (Sieh and Natawidjaja 2000). We suggest these are caused by different offset on cross-structures during episodes of accelerated trench retreat. The second figure (below) illustrates how the pattern of faulting is re-established after one such episode, during which the Banda Aceh segment moved faster westwards, so that the Sumatran Fault was offset as shown. Renewed wrenching on the Sumatran Fault, results first in the formation of relay faults, and then thrusting as the result of the newly formed restraining bends. The stress axes flip during these events, as shown. The episodes of fluid activity recognised here provide some hints as to the frequency and metallogenic significance of these cycles.

ADDITIONAL REFERENCES

CUNNINGHAM W. D. & MANN P. 2007 Tectonics of strike-slip restraining and releasing bends, *Geological Society, London, Special Publications,* vol. 290, no. 1, pp. 1-12.

SIEH K. & NATAWIDJAJA D. 2000 Neotectonics of the Sumatran fault, Indonesia, *Journal of Geophysical Research: Solid Earth,* vol. 105, no. B12, pp. 28295-28326.

[Figure]

The Martabe deposit formed adjacent to the Sumatran Fault, near the intersection of the seismological segment boundary between the 2004 and 2005 $M_w$ 9.2 and $M_w$ 8.5 Great Earthquakes.

[Figure]

*I was initially very excited to see this paper, and I believe it would be a great service to the community for high precision $^{40}Ar/^{39}Ar$ data and diffusion information to be reported for alunite from the Martabe epithermal Au deposit. However, in its present form, I believe this manuscript is not yet ready for publication in* **Geochronology***.*

Agreed, we can make many of the changes suggested by this reviewer, and these will improve the paper.

*I strongly encourage the authors to perform additional experiments that include lower temperatures on the ten alunite samples from this preprint, followed by submission of a revised manuscript.*

We are concerned as to the implications of possible contamination introduced into the mass spectrometer. This contamination appears to have reached the mass spectrometer despite the presence of three getters (including both hot and cold), as well as copper and silver wool in the extraction line adjacent to the furnace. If our interpretations are correct, these contaminants included halogens. Hence the lab will not take the risk of damage to the instrument. Hence no further alunite samples will be analysed in the ANU laboratory.

In any case, in respect to the notion that publication be delayed until additional experiments are undertaken, we demur. Science is not served unless results are published as research proceeds.

*With improved clarity (regarding MDD theory, statistical methods, caveats of the FAP, more detailed analysis of asymptotes and limits without loss of interpretations of min/max ages,etc.) and careful application of the FAP (if and where appropriate), I think this kind of work would be an excellent contribution. I sincerely hope the authors find my comments to be constructive and helpful in generating a revised manuscript for resubmission.*

Agreed, improved clarity will benefit all. However: i) where MDD theory has already been expostulated in print, we will draw attention to those publications more directly; ii) in terms of statistical methods, we will add information to the manuscript where appropriate; iii) we differ in terms of use of terms such as min/max ages in respect to the method of asymptotes and limits, so we will explain why more precisely; and iv) applications of the Fundamental Asymmetry Principle (FAP) will be explained in a little more detail, but the attention of the reader will also be directed to publications in which this has already been done in excruciating detail.

And yes, we found the comments made by this reviewer to be constructive and helpful, and they (the comments) will be of assistance in generating a revised manuscript for resubmission.

**Specific Comments (by reviewer two)**

*Comments by topic*

*For a paper focused on unravelling the history of hydrothermal activity at the Martabe epithermal Au deposit, there is surprisingly little discussion dedicated to this topic (only about 333 words out of a total of 3683 in the discussion section, or about 10%).*

Agreed, and we note that the original paper was short, and directed towards that end. However, in review, every aspect of the work done has been challenged: i) first because the age spectra we obtained are not plateaux and one reviewer considered that therefore the work therefore should be rejected; and ii) more detail has been requested as to the method of asymptotes of limits and its application) as per this reviewer.

*Rather, most of the discussion section focuses on experimental and analytical methods, which, while they are important aspects of the current work, arguably should not take up 90% of the discussion.*

Given that the results stem from routine application of the method of asymptotes and limits, now in use by our laboratory for more than fifteen years, it is a little difficult to confront the circumstance that sees every aspect of our research called into question. Perhaps we can rewrite the discussion to emphasize that the

reviewers requested this detail because: i) the age spectra we obtained are not plateaux and reviewer I considered that therefore the work therefore should be rejected; and ii) the method of asymptotes of limits is not, as yet, in general use: more detail has been requested as to its application, as per this reviewer.

*For example, can you expand your discussion of the geometry of the hydrothermal system? It is currently limited to a small portion of the 333 words in the final section of the discussion and a single figure (Fig. 14). What are the implications of understanding how hydrothermal activity migrated spatially over a 2 Ma period? Could thishave impacted how the ore bodies formed, or provide any utility for vectoring to the ore body during exploration? What are some relevant economic geology interpretations that you can make from your dataset in addition to the timing of hydrothermal activity (e.g., how do grades of ore correlate with the hydrothermal activity?)? How significant is it that you've identified pulses of activity over a roughly 2 Ma period? How confident are you in the timing of each of those pulses, given that you dropped min/max age interpretations when aggregating asymptote and limit dates into probability density plots? More discussion is warranted that is more directly related to the topic of the paper: the history of hydrothermal activity at the Martabe Au deposit.*

Two additional figures are required, to allow the results to be put into a broader tectonic context, as requested by the reviewer. The first figure shows the larger-scale structural context of the Martabe deposit.

The main reason for doing this work was to understand the timing in respect to regional tectonic activity.

*Regarding the Fundamental Asymmetry Principal (FAP) given the context of other recent diffusion studies: it must be made more explicit early in the paper that the FAP is predicated on MDD situations (the presence of multiple diffusion domains in samples).*
Agreed, and we should emphasize this more than we have done already.

*Also, consider the work of Cassata et al., 2013 (Cassata, W.S., and Renne, P.R., 2013, Systematic variations of argon diffusion in feldspars and implications for thermochronometry: Geochimica et Cosmochimica Acta, v. 112, p. 251–287, doi:10.1016/j.gca.2013.02.030): they (and references they cite) observed Arrhenius datawith upward curvature that would violate the FAP.*
We will draw attention to our papers that have shown already how the FAP must be applied locally, in circumstances where a MDD model has different activation energies. Upward curvature does not violate the FAP if it is applied 'locally'. Such 'local' application of the FAP was considered in detail by Forster et al. (2015) <http://dx.doi.org/10.1080/08120099.2015.1114524> including how variation in activation energy explains such upward curvature. We need not repeat this work: instead, we will draw attention to the 2015 paper.

The issue is that there will be an intervening zone of mixing, which Ren & Vasconcelos (2021) have interpreted to be representative of a distinct diffusion domain. This is not necessary, nor warranted by the data. In fact, the increased resolution provided by our data shows the existence of a separate domain not to be justified.

*Cassata et al. interpreted this as an effect of structural transitions in the mineral due to laboratory heating (e.g., thermal expansion which can occur on short timescales, and lattice deformation like the transition from triclinic to monoclinic that can pose an activation energy barrier and exhibit hysteresis on retrograde heating).*
The statement made applies only to what was happening in the mass spectrometer.

However, if Cassata et al. (2013) were correct, one should not expect corresponding variation in the age spectrum. Since there is corresponding variation in the age spectrum, Cassata et al. (2013) cannot be correct.

The simplest explanation is the one that we have already published, namely that such variation in diffusion parameters reflects the existence of highly retentive diffusion domains in potassium feldspar, in the natural

environment. It is indeed an assumption to take the parameters determined in the mass spectrometer back to the time spent by the mineral in the natural environment, and we do agree that such extrapolation can be fraught. This message will be made clearer. However, the data as it stands does support our conclusions.

*These structural transformations combined with progressive exhaustion of sub-grain domains can cause non-linearity both downward (expected by MDD) and upward (not expected by MDD alone in most circumstances; see their Figs. 8 and 10 for examples).*
We have dealt with this point in previous publications, so it is not necessary to repeat this work here. Suffice to say that some of the above statements are mathematically false, as our forward have already shown.

*They included simulations of molecular dynamics coupled with diffusion modeling to arrive at this interpretation, among other lines of evidence. Temperature-dependent anisotropy could also produce upward curvature (e.g., Huber et al., 2011; see Fig 4; full reference: Huber, C., Cassata, W.S., and Renne, P.R., 2011, A lattice Boltzmann model for noble gas diffusion in solids: The importance of domain shape and diffusive anisotropy and implications for thermochronometry: Geochimica et Cosmochimica Acta, v. 75, p. 2170–2186, doi:10.1016/j.gca.2011.01.039).*
We have dealt with this point in Lister & Forster (2016), so it is not necessary to repeat this here. Suffice to say that some of the above statements are mathematically false, as our models have already shown.

*You should be crystal clear about applicability of the FAP: it applies only in MDD situationswhere the contributions from structural transformations and temperature-dependent anisotropy are negligible. To be fair, you acknowledge this (lines 249–253) when you describe the FAP, though it is very easy to miss the operative qualifier (emphasized here in bold): "The Arrhenius plot can be analysed by the selection of lines that divide the population by rank order. Such dividing lines obey the Fundamental Asymmetry Principle (or FAP) recognised by Forster & Lister (2010) and allow estimates for the activation energy of individual diffusion domains. This is a mathematical result that must be applied if one assumes argon loss takes place only as the result of solid-state diffusion* ***without microstructural modification****.*
We have already been crystal clear. We can reorganise the manuscript in order to make this point more evident.

*"Look again at the work of Ren and Vasconselos (2019); they have multiple sections of their paper dedicated to describing the structural transformations in alunite as a result of laboratory heating, which they confirmed with TEM heating experiments in ultra-high vacuum (UHV) conditions. They found that a phase transformation occurs under UHV between 430–460 C, such that the alunite structure gives way to polycrystalline aggregates (of alunite or a mixture of alum + Al2O3 nanocrystalline aggregates).*
It is a difficult topic, so more detail in the discussion is warranted. First however, one needs to be clear. Except for one case, these authors demonstrated a scattering of orientation of crystallites without any change in the lattice parameters. **Hence a phase transformation has not been documented**!

Since Ren & Vasconcelos (2019) provide no evidence that the lattice parameters have changed, the upward slope on the Arrhenius diagram must reflect the PRIOR existence of a more retentive diffusion domain. This makes sense, because the initial release documented by Ren & Vasconcelos (2019) has highly unretentive diffusion parameters. Very small volumes of gas are involved. Release may be the result of the distortions caused by initial heating steps, thereby creating fast diffusion pathways enabling release of [39]Ar liberated by recoil during irradiation for example.

Given that the first gas reservoir defined by Ren & Vasconcelos (2019) has Arrhenius parameters that suggest it would close only below ~10°C, it is not at all likely that release of lattice argon can explain its origin.

This low retentivity reservoir can only be the result of dynamic release, occasioned by the conditions of the experiment, with the Arrhenius data estimating the kinetics for grain boundary release for example as [39]Ar produced by recoil during irradiation is expelled.

As for the steepening of gradient, one could argue that the microstructural modification described by Ren & Vasconcelos (2019) should have substantially reduced the diffusion dimension? But this would not explain a change in activation energy.

Diffusion theory alone is incapable of separating the grain size term from normalized diffusivity ($D/a^2$) which means that other arguments need to be used to develop size constraints. The normalized diffusivity would quadratically increase if diffusion dimension was to reduce. Since lattice parameters did not change, there is no expectation that the observed activation energy should also increase.

*They observed that the activation energy of Ar diffusion increased above this phase transition, which appears as upward curvature with increasingtemperature and would violate the FAP as stated in Forster & Lister (2010).*
Since Ren & Vasconcelos (2019) provide no evidence that the lattice parameters have changed, a phase transformation has not been documented. The slope change is therefore best explained as the result of mixing.

When a material contains diffusion domains with quite different activation energies, Forster & Lister (2015) showed that the FAP could only be locally applied. Further they showed a zone in which mixing between the two domains was evident, in which zone it is not possible to estimate the diffusion parameters. In the mixing zone there is no physical significance to a line of best fit as applied by Ren & Vasconcelos (2019) to determine average slope. Therefore, we dispute the intermediate values of activation energy reported by these authors.

*Furthermore, you state that you started your incremental heating schedules at 450 C; this would imply that all of the diffusive parameters that you report are likely applicable to the post-phase transition polycrystalline aggregate produced when heating alunite above about 460 C. Ina way this is good news because you would expect MDD behavior, consistent with the factthat none of the Arrhenius plots you show exhibit upward curvature that would violate theFAP. But it is bad news in the sense that the diffusive parameters you derived may not berelated to crystalline alunite at temperatures relevant to epithermal systems such as the one at Martabe. Perhaps you started this work before the paper by Ren and Vasconselos was published, so your heating schedules did not make allowance for the phase transformation they documented under UHV conditions. But their paper has been out for more than two years; at a minimum you should be forthright in stating the possibility thatyour current results might not represent crystalline alunite, and you should perform additional experiments that include lower temperatures in step heating schedules to evaluate if and when phase transitions occur in Martabe alunite (after all, compositional variations could influence the temperature at which alunite experiences a phase transformation under UHV conditions). But this should not be left to the readers' imaginations; you should provide evidence. Also, you should be careful in applying the FAP to the results of Ren and Vasconselos given that they documented a phase transformation that occurs part way through their experiments, and because of your own statement that the FAP "…must be applied if one assumes argon loss takes place only as the result of solid-state diffusion without microstructural modification." Please see additional specific comments about your reassessment of the Ren and Vasconselos resultsbelow (for lines 545–546).*
The normalized diffusivity would quadratically increase if diffusion dimension was to reduce, but then there would be no expectation that the observed activation energy should also increase.

It is a difficult topic, so agreed, more detail in the discussion is warranted.

*Regarding the eArgon software: the analytical algorithms and statistical methods implemented in this software should be clearly exposed to the readers, either by explicitd escription in this paper or by reference to a paper that does include sufficient detail. At present, this software is effectively a black box. For example, do you have a publication analogous to that describing ArArCALC (Koppers, A.A.P., 2002, ArArCALC—software for $^{40}Ar/^{39}Ar$ age calculations: Computers & Geosciences, v. 28, p. 605–619, doi:10.1016/s0098-3004(01)00095-4)?*

*Or maybe another paper where you describe things like regression methods, handling of errors, outlier detection, and assessing goodness of fit? These details should be made readily available to readers.*

This information will be included in the revised supplementary information.

*eArgon* includes various options, which have been expanded to include weighted means and regressions using revised formulae from Mahon (1996). The data tables used in the original publication included methodological uncertainty in the age estimates, including uncertainty in the J-factor determination, as well as uncertainty in the estimated age for the fluence monitor. Such systematic uncertainty applies in a similar way to the entire age spectrum, so it should be removed prior to performing goodness of fit estimates, and then reapplied.

This will be done for the revised manuscript, including appropriate documentation.

*Figures 3–12: potential for improved clarity: consider modifying captions and in-figure labels to use descriptive keywords indicating how you have identified spectrum segments with age significance (e.g., label them as min/max age, upward/downward-bounding asymptote, intermediate asymptote, etc.). You can cite your 2004 publication for detailed descriptions of the various asymptote types (probably in main text where you describe your methods), but I think it would be good to use consistent terminology with the method you developed (rather than "plateau segment"). Also, it would be nice to have the upper x-axis of Arrhenius plots labeled with temperature in Celsius; this would help potential readers who may not be familiar with such plots to interpret them more effectively (e.g., by seeing in plain text that higher temperatures are to the left).*
In part or in whole this can be done

**Line-by-line comments**

*Lines 101–102: Did you also correct for decay of $^{39}Ar$ and/or $^{37}Ar$ following irradiation? Do you list what decay constants you used for that (e.g., in the supplement)?*
Correct. Each irradiation canister is dispatched along with various salts to allow measurement of these correction factors using methods as documented in McDougall and Harrison (1982, 1999).

These include corrections for neutron interferences derived from calcium and chlorine. The corrections made are shown in the data tables (in the supplementary information). The York Plots are made using the corrected ratios, thereby eliminating this aspect as a matter for concern.

All constants are listed, consistent with Schaen et al. (2021).

*Lines 105–108: Earlier (line 96) you mentioned picking a total mass of 100-150 mg per sample. Did you then analyze single grains, or did you analyze multi-grain aliquots? Please clarify in these lines.*
Multigrain aliquots were analysed.

*Lines 113–114: You are injecting interpretation very early into the paper by identifyingfluid inclusions as the carriers of KCl and CaCl₂ contaminants. I would recommend withholding interpretation until after the results are presented. Also, I have separate concerns about fluid inclusions as the hosts for contaminants that dominantly produce effects on Cl/K and Cl/Ca ratios at very high temperatures; see additional comments below (e.g., for lines 234–235).*

We can do that. The reviewer is correct. Interpretation was introduced too early. Note that fluid inclusions can contain precipitated material, however. In any case, it is included material in general that is of interest.

*Lines 145–146: The following statement seems unnecessary and is misleading: "The resultant age spectrum allows vastly more information to be ascertained than any laser spot analysis can provide."*

We agree with the reviewer that it is perhaps unnecessary. It was introduced in response to an earlier review, challenging the use of step-heating techniques. This comment can be removed, though we do feel obliged to discuss the merits of furnace *versus* laser methods in order to address comments offered by earlier reviewer.

*This is perhaps true of an individual laser spot analysis (N = 1), but to make a fair comparison you should only consider a single (N = 1) step from an incremental release spectrum.*

What we have come to realize is that there are some common misunderstandings as the merits of otherwise of different methodologies, and there are some fundamental misconceptions. This is one of them.

A laser spot analysis is indeed a single measurement (N=1). If the laser moves to another spot, and another measurement is made, N=2. And so on. This is the same methodology as used by SHRIMP for example.

Wikipedia says it all, succinctly: *a Markov chain or a Markov process is a stochastic model describing a sequence of events in which the probability of each event depends only on the state attained in the previous event*. In essence each step in the age spectrum depends on the results that have come before.

The results of a step-heating experiment must be examined in the context of an attempt to resolve the detail of a history-dependant spectrum. Individual steps are relative. An age spectrum with 43 steps is thus an entity (N=1) that can in no way shape or form be compared with the independent results of N=43 laser blasts from points scattered within a microstructure. Laser spot analysis thus produces results that are something quite different to that which transpires as the result of a single step-heating experiment.

It might be that the laser spot is held constant in its position in the microstructure and a comparison made of gas released from a succession of reheating steps. Without temperature control however, it is unlikely that it would be possible to see any fine detail. Perhaps the use of the epithet "step-heating" should be restricted to circumstances in which variation temperature during the heating step approximates a square wave. Alternatively, if "step heating" is be used loosely (e.g., to include heating during laser spot analysis where the spot is allowed to meander) then, more attention to methodological detail is warranted, to avoid misleading the non-specialist. Reviewer one makes this point. Reform is necessary. Hence attention to this detail is warranted. Perhaps it is no accident that use of a laser results in plateaux?

Summary: A statistical model is assumed *a priori* in respect to analysis of the results. There are those who mistakenly believe that argon geochronology should *a priori* yield a single age. In this case, it might be imagined that a succession of laser blasts should produce numbers that allow the application of a Gaussian statistic (*e.g.*, the mean, variance and mean square weighted distribution or MSWD). First, as demonstrated by comparison of different spectra for the same material, e.g., as provided by Ren & Vasconcelos (2019), a laser typically produces too coarse a dispersion to reveal the fine detail displayed by a temperature-controlled step-heating experiment. Second, it is a fundamental misconception to consider the results of a temperature-controlled step-heating experiment as a sequence of unconnected measurements.

*Laser spot analyses are rarely conducted in isolation; rather large numbers (dozens, N >> 1) of spot analyses are performed, which, when interpreted as an ensemble (like a full release spectrum consisting of tens of individual steps is), can be extremely useful in identifying multiple generations of materials in a single petrographic section.*

Here we differ from the opinion provided by the reviewer. A collection of laser spot analyses provides a scatter of single value averages that disperse over the sample, for example as would a collection of K/Ar analyses. Laser spot analysis can be utilized and multiple growth generations can be identified, as described by the reviewer, but if and only if the 'spot' has sufficient spatial resolution, and individual grains have specific single ages (and hence no history of argon loss after growth). In this circumstance dispersion of ages amongst a collection of laser spots can reflect dispersion of growth ages. But the two necessary conditions for this to be true are rarely attained.

*Rather than selling incremental heating as the only game in town, consider that the two methods—incremental heating and laser ablation—have different strengths and weaknesses and can complement each other.*

The reviewer is correct, and there is no need to battle this out in this paper. The offending statements can be removed. They were added only in response to earlier criticism that laser methods were not utilized. We consider laser methods are of value in reconnaissance mode, but then since laser operation involves manual operation, we would save neither time nor effort. A step-heating experiment yields far more information, and with robotic methods allowing 24/7 operation of the laboratory, such data can be far more easily obtained.

*The former method, when performed with precise temperature control (for which you advocate), can be pairedwith diffusion modeling to provide insight into grain/domain size distributions and mineral/chemical reservoirs that may represent multiple age components; but the inferred identities of the minerals/microchemical reservoirs that released gases throughout the experiment are linked with model assumptions rather than direct visual observation.*

The reviewer is correct, but only in part. Mineral separation prior to the experiment determines the aliquot to be measured, and careful selection with a microstructural focus obviates this discussion.

*Laser ablation, when many analyses are performed on a sample, allows precise targeting of textural domains and potentially individual phases (depending on grain size) while preserving petrographic context for later visual inspection, so there is less ambiguity of what material is dated.*

The reviewer is partly correct, but only for the case that coarse grained samples are present. However, there is a wealth of information suggests that diffusion domains are much smaller than currently assumed to be the case. But there is no need to battle this out in this paper.

*However, for fine-grained samples especially, the ablated volumes can represent mineral mixtures and so must be interpreted with care and ideally with the support of modeling (just as in the case for incremental heating experiments of multi- domain and/or polymineralic samples).*

The reviewer is again correct, but in our experience (with metamorphic tectonites) this case above is what generally applies. There is a wealth of information suggests that diffusion domains are much smaller than currently assumed to be the case.

*The bottom line is that both methods can yield populations of apparent ages resulting from a mixture of gases from potentially distinct mineral/chemical reservoirs, which must be interpreted accordingly while accounting for the strengths and weaknesses of each technique and with care in selecting model assumptions (e.g., using independent chemical and petrographic observations).*

There is no need to battle this out in this paper. If we are obliged in consequence of each review to debate the theory and practice of argon geochronology, the paper would continue to grow.

*The combination of these techniques can be very powerful since the strengths of one can alleviate the shortcomings of the other. Furthermore, what is the advantage of denigrating laser ablation in this context? It does not provide relevant information for interpreting your dataset; rather, it distracts from the point you are trying to make. I recommend removing this comment.*

We will remove this comparison of laser versus step-heating methods. Note that we consider laser methods are of value possibly in reconnaissance mode, but then since laser operation involves manual operation, we would save neither time nor effort. A step-heating experiment yields far more information, and with robotic methods allowing 24/7 operation of the laboratory, such data can be far more easily obtained.

*Lines 234–235: Please provide additional detail for the interpretation that inclusions (specifically identified in lines 113–114 as fluid inclusions) bearing KCl and $CaCl_2$ caused the observed increase in Ca/K and Cl/K ratios at high temperatures (apparently >800 C in most cases). Most fluid inclusion work I've seen indicates that fluid inclusions decrepitate at much lower temperatures, i.e., <400 C. There may be some potential for variation in decrepitation temperatures based on entrapment pressures, but I've not heard of fluid inclusions remaining stable to >800 C, even in alunite.*

We will clarify our uncertainty as to this puzzling aspect of our measurements. We should not have referred to inclusions as fluid inclusions. We should also make it clear that the York Plot can be used to analyze mixing between different reservoirs, and such analysis suggest small volumes (<1%) are involved, with the effects becoming evident only as lattice argon is depleted.

*What alternative hypotheses can you propose?*

Ionization behaviour after measurement of the $CaF_2$ standard (despite the presence of multiple getters) suggests small amounts of $F_2$ are capable of making it all the way to the mass spectrometer. Since $F_2$, $HF_2^-$ and $H_2F_2$ have atomic weights close to 38, 39 and 40 this is an alternative source of contamination. But since such ionisation behaviour is not observed here, this is not discussed.

*For example, Landis et al. (2005; reference: Landis, G.P., Snee, L.W., and Juliani, C., 2005, Evaluation of argon ages and integrity of fluid-inclusion compositions: stepwise noble gas heating experiments on 1.87 Ga alunite from Tapajós Province, Brazil:Chemical Geology, v. 215, p. 127–153, doi:10.1016/j.chemgeo.2004.06.036) studied noble and active gases in ca. 1.87 Ga alunite from Tapojós, Brazil, and they observed increased $^{38}Ar/^{36}Ar$ ratios between 500–700 C.*

*Notably, fluid inclusions generally released gases below 300 C, then the alunite OH cites released gases at intermediate (300–500 C) temperatures, and the K-$SO_4$ tetrahedral sites dominated gas release above 500 C. They attributed the increased $^{38}Ar/^{36}Ar$ ratios to nuclear reactions (induced by thermal neutrons produced by the decay of U and Th) on $^{41}K$ and both isotopes of trace Cl (up to several hundred ppm) present in the K-$SO_4$ sites. This would help explain the increased contribution from Cl-derived Ar at high temperatures; but perhaps such an explanation would not apply for such young alunite as you have analyzed since there is little time for accumulation of nucleogenic products... You would need to evaluate this possibility more for your samples (e.g., do you know what the Cl contents are of your samples?), but it should perhaps be included in your multiple working hypotheses until disproven.*

We agree – we should cite and briefly discuss these works.

*But, supposing the observed increases in Ca/K and Cl/K are not due to fluid inclusions nor neutron reactions on $^{41}K$ and trace Cl; what other types of inclusions could then be carriers of Cl? Perhaps some sort of mica or clay mineral, which is not uncommon in hydrothermally altered rocks?*

These minerals were difficult to separate. We noted that the effect begins at the melting temperature for salts such as KCl, which is a persuasive argument.

*Do you have independent petrographic observations that could help constrain a possible highly retentive carrier phase included in the alunite? Whatabout the XRD data you mention? Could you (or did you) include it in the supplement? (The supplement was not available for review.) There is opportunity here for you to evaluate additional hypotheses for the observed spikes in Ca/K and Cl/K in your release spectra. Perhaps you could include an additional discussion section to expand upon the implications of your results beyond the diffusive characteristics of Ar in alunite and the timing of hydrothermal alteration at the Martabe deposit. Supplemental information would be very helpful in narrowing down the field of possibilities.*

We have included an additional figure, and quantitatively assessed the composition of the impurities.

*Lines 235–236: "In many cases, early steps define the most retentive diffusion parameters for alunite." How is it possible to have less retentive domains degas after more retentive ones (especially when using a prograde heating schedule)? Does that require the less retentive domains to be shielded/enclosed by more retentive domains? Once the more retentive domains are exhausted, how could you then recognize less retentive domains? Shouldn't the diffusion be rate limited by the enclosing phase/domain?Or, perhaps the volume fraction of low-retentivity domains is much larger than those for more retentive domains? How does this work?*

However, it is indeed possible for the spectrum to release gas from domains with more retentive parameters before gas released from less retentive domains begins to overwhelm that signal. This is because the equations governing the percentage gas release are not single-value functions, e.g., release from muscovite may dominate both the early steps and the last steps of an age spectrum, while in the middle, the muscovite signal is overwhelmed by release from highly retentive phengite. Frequency factor determines the spread of gas release. If the frequency factor is higher, there is less spread. Accordingly, a signal may appear early or later.

The reviewer is thanked for pointing out two diagrams for which the labels (with values of E and $D/a_2$ for different segments of the Arrhenius plots) had been inadvertently switched.

*Lines 249–251: This is the first time in the main body of the text where you mention the Fundamental Asymmetry Principle (FAP), but it is not explicitly clear that it assumes MDD (multi-domain diffusion) behavior. Indeed, at this point in the paper, you have not explicitly mentioned or described the concept of MDD; it is not until a few lines later (line 255) that it is mentioned without definition of the acronym. It is not until the next section,around lines 281–293, that you reference work by Lovera et al. (1997) in the context of single-domain diffusion and then weakly describe MDD with a series of implicit references but no explicit definition. This is a severe lack of clarity for any readers that are not familiar with argon diffusion work in general, and MDD work in particular. However, this should be relatively straightforward to rectify; introduce the concept of MDD explicitly earlier in the paper. It might be appropriate to describe in one of your methods sections, e.g., where you describe your diffusion experiments (the MDD context would be helpful forunderstanding your choices in experimental methods, too).*

The reviewer is thanked for pointing this out. Text was moved around in addressing issues raised in an earlier review. Agreed, this can be easily remedied.

*Lines 281–293: There is an opportunity to improve clarity here. You cite Lovera et al. (1997) in describing how diffusion parameters are determined for a single diffusion domain. But then you move into discussing MDD scenarios without making this explicit to the reader; even as someone familiar with this kind of work this was confusing, so imagine how a reader that is not familiar with MDD theory might react. I suggest amending the sentence on line 285 to read something like: "There has been much debatearound interpretation of Arrhenius data for samples with multiple diffusion domains (MDD)." If you would rather mention the concept of MDD elsewhere in this section, that is*

*fine, but I think it is essential to set the context explicitly. As currently written, it is unclear from the description on lines 287–291 what the Fundamental Asymmetry Principlestates. Read naively, the description following "Simply put…" seems more like an observation that data points may lie to the left and right of a line, but does not really statehow to interpret such a situation. The final sentence then citing the FAP and uniformity/density of data in 1/T space as critical requirements for determining closure temperatures and running diffusion models therefore feels confusing. But, by making it explicit that you are talking about an MDD scenario and that the FAP is predicated on the idea that the progressive exhaustion of domains should impose a particular "behavior" for Arrhenius data and that it should govern how data are regressed to extract diffusive parameters would help the clarity of this section. The paragraph in your Forster and Lister (2010) paper where the FAP is introduced does a nice job of setting the context before describing how data should be regressed, it would be great to see that emulated here. However, you must also be clear of situations in which applying the FAP is inappropriate,e.g., when there are structural changes to the crystal lattice; see other comments regarding this above and related to your reassessment of the data published by Ren and Vasconselos.*

The reviewer is thanked for pointing this out. We can do as suggested.

*Lines 302–303: Have you considered using a kernel density estimate (KDE) plot? This is in a sense what you have done by (arbitrarily) selecting σ = 0.05 Ma in panel b of Fig. 13; but an adaptive kernel based on the clustering of the data could be useful. For example, the DensityPlotter of Vermeesch (Vermeesch, P., 2012, On the visualisation of detrital agedistributions: Chemical Geology, v. 312, p. 190–194, doi:10.1016/j.chemgeo.2012.04.021) uses the adaptive kernel of Botev et al. (Botev, Z.I.,Grotowski, J.F., and Kroese, D.P., 2010, Kernel density estimation via diffusion: The Annals of Statistics, v. 38, p. 2916–2957, doi:10.1214/10-aos799), which is designed to best replicate the unknown generating function using observations (data) generated by that function. I doubt this would fundamentally change your discussion here, but is perhaps something you can consider for future work.*

We will look into this. The key is that an argon spectrum is a Markov chain, not a sequence of independent measurements. So, in reality, methods as above cannot be applied.

*Lines 303–308: It seems that any interpretations of asymptotes and limits as min/max ages are lost when the data are aggregated into PDPs/KDEs in this way. Isn't the directionality implied by min/max ages important for interpreting when periods of alunite growth occurred at Martabe?*

We emphasize the difference between Gaussian *versus* Poisson approaches to statistical questions. In the case at hand, we consider each estimate as an unbiassed entity, and ask the question as if the dice had been rolled, and the answers accumulated in each 'age box'. This requires more data than we have, if it is to be done effectively, so data have been bunched. If too many estimates end up in the same age box for the result to be considered random, then we need to search for an explanation for each FMA individually.

*Perhaps it would be better to plot all minimum ages in one color, all maximum ages in another color, and all intermediate asymptotes in a third color? This might change the inferences that can be drawn from the figure for the hydrothermal history at Martabe, since it would retain a sense of bounding ages. By treating each asymptote/limit date as a potentially valid age of alunite formation, I think you reduce the reliability of your age interpretations. This is perhaps what your previous reviewers were driving at by commenting on the definition of a plateau (e.g., lines 318–329).*

Future work will be able to consider our estimates and then search for an explanation for each FMA individually Otherwise, we would be locked in a mindset governed by the mistaken belief that there should be a single age, and that all of the observed scatter is merely a random redistribution of ages according to a Gaussian statistic.

*While I appreciate your contention that we (as a community) should develop ways to analyze and interpret complex release spectra (which I agree with), I don't think that justifies treating means of two or three steps as totally reliable (which is effectively what you've done by plotting them together on PDPs and picking out ages from the largest peaks formed by overlapping "frequently measured ages"). Rather, I recommend treating the asymptotes and limits you identify as bounding limits where appropriate (min/max ages), and then preserve that context throughout your analysis and interpretation of the timing of hydrothermal activity at Martabe.*

The question is whether one uses a Poisson statistic as an alternative to the Gaussian statistic embedded in every aspect of the traditional approach to argon geochronology. Rather than imposing the traditional approach, and thereby attaining a traditional result (by somewhat circular reasoning I suggest) the alternative approach is to accept the significance of the "Markov chain". We need to get busy trying to work out the factors that govern the morphology of individual spectra, I suggest, and to develop explanations as to why the method of asymptotes and limits seems able to routinely identify replicable sets of frequently measured ages (FMAs).

Contrary to the assertion above, we do not use 'estimates' as reliable indicators. Rather we act in reverse. We do not treat such estimates as unreliable. Perhaps the manuscript needs to go into detail more as to the difference between Poisson statistics and the Gaussian model assumed in conventional definition of a plateau.

For example, as in the additional figure at the start of this rejoinder, let us assume the existence of a second population. In a Gaussian probability analysis this would be submerged. This would not be the case if Poisson statistics were applied. Simply stated, the existence of "mini-plateaux" has been noted in our analysis, and the ages measured have been found to be replicable. We have too little data to go much further than that.

We can safely assert, based on the limited data we have already, that these "mini-plateaux" (and other estimates as appear from application of the "method of asymptotes and limits") are not randomly distributed in age. It is a somewhat trivial application, but this can be done using a Poisson approach in the revised manuscript, as requested.

In some ways, the mindset behind application of a Gaussian statistic is the greatest hurdle for reform in respect to the way age spectra are examined in argon geochronology. The assumption *a priori* that there is a single age can routinely be tested by application of a Poisson statistic, and it can thereby be shown if the FMA for a particular "mini-plateau" is random noise or otherwise.

In reality, if there are competing populations of gas reservoirs governing the pattern of gas release, age spectra with complexity as shown will result. Diffusion ensures complex patterns of gas release. Multiple growth domains ensures that complexity is taken a new level.

*Lines 545–546: "We have scaled [Ren and Vasconselos'] diagrams and applied the FAP and obtained revised estimates for the activation energy are entirely consistent with our results." This statement referencing alunite Ar diffusion results published by Ren and Vasconselos (2019) should be accompanied by figures demonstrating your reprocessing, at least in the supplement (which was not available for review). This statement is also a bit confusing; when you say you scaled their diagrams, do you mean you worked with images of the figures? They published all their data; did you instead replot their data andperform your own analysis? The former seems insufficient to demonstrate similar results,while the latter would be reasonable and could easily yield supporting figures that you could include in the supplement.*

The reviewer is thanked for the suggestion. However, this requires a lot of effort and this seems unwarranted: i) the Ren & Vasconcelos (2019) experiments used temperatures that rose and fell too slowly to well approximate a Dirac function, and hence their experiments are intrinsically incapable of yielding accurate estimates using the analytical equations that relate normalised diffusivity to the percentage of argon loss; ii) Ren & Vasconcelos (2019) did analyse some Arrhenius data in a way that graphically seems consistent with the Fundamental Asymmetry Principle, and thereby obtained estimates compatible with those reported here. I did offer to help Vasconcelos reinterpret these data, however, in ways consistent with the FAP.

*Notably, the range of closure temperatures you report (400–560 C assuming cooling at 20 C/Ma) are significantly higher than those reported by Ren and Vasconselos (ca. 240–270 assuming faster cooling at 100 C/Ma, which should yield higher closure temperatures than those for slower cooling rates). Even the results "above phase transformation" (APT) reported by Ren and Vasconselos only yield closure temperatures as high as ca. 330 C assuming cooling at 100 C/Ma (from my own calculations using their reported activation energies and pre-exponential constants for APT regressions).*

Ren & Vasconcelos (2019) did not apply the FAP in the analysis of their data, and therefore they obtained diffusivity estimates that are too low, based on our graphical analysis of the data illustrated. Naturally, this means lower closure temperatures. Moreover, the intermediate estimates in Ren & Vasconcelos (2019) are almost certainly the result of mixing, so these estimates should be discarded.

When this is done, the estimates for activation energy provided by the Ren & Vasconcelos (2019) data are then not inconsistent with our own, except for the estimates obtained using very low percentages of [39]Ar release. These low values are so unretentive that alunite would have reset to a zero age. Since alunite in the natural environment does not reset to zero age, the estimates of low retentivity reported by Ren & Vasconcelos (2019) are unlikely to reflect the values that apply to lattice diffusion. Instead, we suggest, these values reflect dynamic release of [39]Ar produced by recoil, with distortions of the polycrystal enabling its escape.

*It is important to provide evidence that your analytical methods yield dramatically higher closure temperatures using the data of Ren and Vasconselos, assuming that application of the FAP is appropriate to their dataset (which is probably not the case for their alunite data; see separate general and specific comments regarding the FAP). Does application of the FAP to their data imply that higher-temperature steps (e.g., those they state are above the phase transition temperature) would be included in the regression of diffusion parameters on an Arrhenius plot?*

The data provided by Ren and Vasconcelos (2019) curve from one domain to the other, exactly as occurs during MDD simulation of two existing domains. They estimate three lines. The intermediate line is hard to justify. We can illustrate this in a revised version of this manuscript. It is important, I agree, so perhaps this should have been done when we first saw the results published.

*Lines 561–562: The following statement seems unrealistic: "This is the first example of overprinting events being recorded in a single step-heating experiment of alunite." Any release spectrum produced from alunite with classical signs of partial diffusive loss (i.e., younger apparent ages in the early steps of a spectrum) could indicate either slow cooling or later thermal overprinting. Are there really no such examples in the published literature? See for example Fig. 11 of Juliani et al. (Juliani, C., Rye, R.O., Nunes, C.M.D., Snee, L.W., Silva, R.H.C., Monteiro, L.V.S., Bettencourt, J.S., Neumann, R., and Neto, A.A., 2005, Paleoproterozoic high-sulfidation mineralization in the Tapajós gold province, Amazonian Craton, Brazil: geology, mineralogy, alunite argon age, and stable-isotope constraints: Chemical Geology, v. 215, p. 95–125, doi:10.1016/j.chemgeo.2004.06.035). I would recommend dropping this claim, but if you would like, perhaps it would be more appropriate to state this is the first application of the method of asymptotes and limits andthe FAP in dating alunite with complex release spectra produced by thermal overprinting?*

The reviewer is correct, we will remove the offending statements. Note the Ren & Vasconcelos (2019) data expresses argon plateau with multiple plateau segments, suggesting effects precisely as discussed in this paper.

*Lines 345–355: While I understand your position that temperature control is essential for diffusion experiments that are coupled with $^{40}Ar/^{39}Ar$ dating, I feel like this section is unnecessarily focused on semantics. Why should experiments that employ lasers to degasthe samples but that lack precise temperature control not be called step-heating experiments?*

Depends I suppose on whether heating can be supposed to have occurred in steps, and whether the same material was reheated time and time again. If both criteria are correct, then perhaps it is OK to call the process "step-heating". More precise definition of terminology is needed.

*After all, the laser power output is controlled, and in general the laser poweris increased from one step to the next (i.e., in a stepwise fashion) such that the temperature the sample experiences does indeed increase with each successive step. The temperature is simply not measured, and specific target temperatures are not specified.*

More precise definition of terminology is needed. Would it not be better to derive a more precise description of methodology for circumstances as described by the reviewer?

*For many (if not most) applications, this is sufficient to assess a precise age for a well- behaved sample (e.g., one that had a simple thermal history and behaves as a single diffusion domain). But, as you point out, this is not sufficient for deriving diffusion parameters from a sample; temperature control is required in this case. I don't think anyone argues with that. So why the comments on terminology? This feels more like an unnecessary distraction, and it takes up valuable space in your discussion section that could perhaps be better allocated to direct discussion of the economic geology implicationsof your dataset with regards to the Martabe Au deposit. I suggest eliminating this paragraph to reallocate discussion space, or at a minimum tone this section down to focuson fundamental issues (like the need for temperature control for diffusion experiments) rather than semantics (like what should or should not be called a step-heating experiment).*

Correct: we need avoid unnecessary distraction, but note that these result from different demands that have been made during peer review. We can eliminate this one, if necessary, but given the ambiguity being made evident in methodology, perhaps this is not a good idea. The discussion can be focussed, nevertheless.

*Lines 374–376: when you say you chose sample sizes that would allow enough steps at appropriate temp intervals, does this mean there are aliquots with multiple grains that you degassed? Or were all analyses conducted on single grains? Please provide more detail for clarity.*
Aliquots with multiple domains were degassed.

*Lines 388–390: You again mention the presence of KCl and CaCl₂ impurities hosted byinclusions. I think you need more support for this interpretation, or at least some additional working hypotheses that can be tested. Wouldn't KCl impurities leave the Cl/Kratio constant? (Suppose only KCl was present as an impurity.) How can fluid inclusions survive to such high temperatures?*
See detail in discussion of this very point, provided earlier in the rejoinder.

*Lines 408–418: This paragraph is somewhat confusing to me given the context that you have effectively ignored the structural transition that alunite experiences during laboratory heating, as documented by Ren and Vasconselos (2019; see additional general and specific comments above).*
The point being made is that domains in the measured age spectra correlate with domains in the Arrhenius plot. Hence it is likely that the same correlation applied in the natural environment.

*Am I right that your argument here is that: (1) it doesn't matter what the diffusion parameters are or whether they apply to the natural epithermal hydrothermal system as long as (2) diffusion parameters for each inferred diffusion domain correspond to an asymptote or limit identified in the release spectrum, which therefore implies that (3) each diffusion domain is the result of a distinct period of alunite growth?*
The paper makes the point that the estimates provided for the diffusion parameters suggested that we were measuring the effects of different episodes of mineral growth.

*If assertion (1) is true, then why bother reporting closure temperatures and interpreting the relative retentivity as a bonus for preserving $^{40}Ar/^{39}Ar$ systematics in hydrothermal alunite? Should you therefore omit such information from your paper?*
Assertion (1) is not correct, and neither is it what we meant to imply. We need only to demonstrate that the mineral is sufficiently retentive to allow the conclusion that these are growth ages. It is interesting that Ren and Vasconcelos (2019) beat us to the punch in this aspect – and we are glad that they did. But there is no need for us to labour over the detail. Many of the diagrams were brought into the body of the manuscript in consequence of an earlier reviewer demanding that they be present.

*If assertion (2) is true, then why are there visible mismatches in the steps used to determine diffusion parameters and asymptotes/limits in release spectra in Figs. 2–13?*
Assertion (2) is not correct, and neither is it what we meant to imply. The mathematics of an Arrhenius plot is non-transparent, because of the way mixing works. There is no 1:1 correlation that should ever be implied.

*For example, in Fig. 10 (chosen at random), there appear to be two domains defined in Arrhenius space using steps 7–11 and steps 17–21, but the corresponding dates inferred from the release spectrum involve steps 11–14 and steps 21–22. You have not explained this anywhere in the text as far as I can tell; perhaps that mismatch is expected. But it should probably not be left to the readers' imaginations; please explain more clearly.*
This is difficult to explain, but in essence the issue is that mixing is non-linear. Because this is true, an exact correspondence is not always to be expected. One diagram (the age plot) is in essence a spectrum that results from a linear variation in temperature, whereas the other (the Arrhenius plot) is a spectrum that

reflects a functional variation during the same sequence plotted against inverse temperature (10000/°K). The effect of mixing is thus 'felt' differently.

For example, consider that conjoint inversion of the first few steps of a UHV diffusion experiment allows prediction as to the shape of the remainder of the Arrhenius diagram. Such a result might not be expected by the human observer, because all we see is a linear array of points on an Arrhenius plot. However, by adding the constraints provided by the normalized radius plot (which after all has the same axes as the age plot) the computer is able to draw inferences from the pattern of increasing percentage [39]Ar release, and thereby reveal that the larger more retentive domains in the crystal lattice have also contributed to the gas release pattern in those first few steps.

*And finally, for assertion (3) do you have any independent evidence to suggest that the sub- grain domains present in alunite from Martabe grew at different times? It seems like classic optical petrography or some SEM/EPMA/EBSD work would go a long way in supporting such an interpretation. How can you rule out recrystallization during repeated hydrothermal events?*
The opportunity to do more work on this aspect is long gone. In any case, recrystallisation during a later hydrothermal event would almost certainly involve a dissolution/reprecipitation reaction.

*And when you consider that laboratory heating under UHV conditions does cause structural modification of alunite (as documented by Ren and Vasconselos), how does that affect this discussion?*
The structural modification during UHV experiment may affect the diffusion parameters estimated. This is a drawback of the method. This said, we do not agree that the intermediate values reported by Ren and Vasconcelos (2019) are necessarily correct, as noted in the manuscript. We can make that clearer in a revised manuscript. The bottom line though is that we think the estimated retentivity is sufficient to justify the assertion that these are growth ages.

*You've apparently admitted (perhaps unintentionally) that the diffusion parameters you've derived are not representative of the natural epithermal environment.*
The reviewer is incorrect. We routinely and freely admit that there are many opportunities for phenomena to occur that would ensure that diffusion parameters estimated by way of an UHV diffusion experiment would not be representative of the natural epithermal environment. Nevertheless, experimental results are consistently duplicated, suggesting that our methods routinely ensure metastability, at least initially.

*But what then is the utility of your results for Martabe alunite? Perhaps age information for multiple episodes of hydrothermal alunite growth canpersist during laboratory heating, but I think this could be discussed with more clarity andwould benefit from independent evidence (e.g., petrography and SEM or similar work). And if you want to report closure temperatures for Ar in alunite that are relevant to natural environments, I again recommend that you perform additional experiments that include lower temperatures so that you can regress data from below the phase transitiontemperature of alunite (as recommended by Ren and Vasconselos, 2019).*
The opportunity to do more work on this aspect is long gone. We are trying to document a result, a prediction made as the result of detailed step-heating [40]Ar/[39]Ar geochronology that can subsequently be verified or disproven by careful petrography. This is a perfectly valid way to do science.

*Lines 426–427: This is a reasonable statement given the context of studying microstructural domains that are well below the footprint/spot size of a laser system. But,as with your comment on lines 145-146, I do not understand the need for this statement.I don't think anyone would argue that diffusion information can be obtained by laser ablation when the diffusion domain size is so small, so singling out this technique feels more like an unnecessary distraction than a useful piece of information. I suggest removing this comment.*

These statements were inserted as the result of commentary made by an earlier reviewer. They can be removed (perhaps should be removed). But then we would distort the results of peer review as we know it.

*Lines 455–459: Again, what independent evidence is there for KCl and CaCl₂ salts (besides the Cl/K and Cl/Ca ratios)? The high temperatures at which the Cl/K and Cl/Ca ratios abruptly change seems too high for fluid inclusion hosts, so where is the contamination hosted in the mineral? What other possibilities are there? See more detailed comments above for lines 234–235.*

We need to indicate what we know as opposed to what we surmise, and why. This requires an extra figure, to illustrate how mixing typically takes place on a Turner three component inverse $^{40}$Ar correlation diagram, as below (a). In (b) the York (1969) regression is used to define inverse isochrons, to demonstrate the bounds for the four different reservoirs. Uncertainties are shown as 1σ, using uncertainty calculations as corrected by Mahon (1996) and then by Trappitsch (2018).

The key to our reasoning as to the nature of the last reservoir is that: i) small volumes are involved (<2% of the total $^{39}$Ar released); ii) the mixing with the unidentified reservoir begins at the KCl or CaCl₂ melting temperature; and iii) the Ca/K ratio in the unidentified reservoir approaches 6:5 whereas the Cl/K ratio is 5:1 which can be achieved by a molal composition of inclusions equivalent to 4CaCO₃ + 2CaCl₂ + KCl. Clays such as kaolinite or dickite are present and residual material may have been included, but these compositions do not typically include K, Ca or Cl so if calcite (CaCO₃), sinjarite (CaCl₂), sylvite (KCl) or chlorocalcite (KCaCl₃) mineralogy is present, these minerals are cryptocrystalline, and so have escaped notice.

We note that fluid inclusions are abundant overall (Saing *et al.* 2016) but we have no information as to the alunite nicrostructure. The minerals listed above maybe present in microcracks rather than as cryptocrystalline residues in inclusions. Clays such as kaolinite or dickite break down at >1000°C, i.e., coincidentally at the temperatures at which the unidentified reservoir is most evident.

[Figure]

(a)  (b)

We need to revise the figures in the resubmitted manuscript to ensure that the plotted Ca/K and Cl/K ratios remain on scale, whereas now the ratios in the last 2% of gas release are not shown.

*Lines 474–476: You make the statement "...the Frequently Measured Ages (FMAs) that have been recognised using Gaussian probability plots can be shown to be statistically significant using Poisson statistics: it is highly improbable that coincidences in the determined ages are the result of random processes." Is this evidence shown in the supplement (which was not accessible for review)? If it can be shown, can you please show it? It may seem trivial to you, but as written this point this is effectively an unsubstantiated claim that limits transparency to readers.*

Whereas Gaussian statistics assumes a mean and scatter thereabouts, Poisson statistics asks the question as to whether a particular distribution of values can be attributed to a random process. Therefore, one can assess the likelihood that measured values may fall randomly in a set of ranges, as shown. Such a calculation can be included in the revised manuscript, to make the point requested by the reviewer.

*Lines 547–549: "In review, we were asked to comment on the difference between [Ren and Vasconselos'] spectra and our own. There is no reason to do this. Perhaps the flat spectra obtained by these authors imply nothing more than that only one generation of crystal growth took place in their samples?" I do not think this is such an outlandish request, especially for readers who do not themselves perform diffusion experiments. Isthere a difference in the heating schedules you used compared to the work of Ren and Vasconselos? Could that contribute to the appearance of the release spectrum? Or is it ageological difference that contributes to the differences? I find it totally reasonable that you could guide the readers through that thought process, at least to provide some direction so they can evaluate the spectra for themselves for effectively.*

We initially demurred in respect to this request, but since the request has been repeated, we suggest the following commentary could be added to the discussion. The revised manuscript will thus make explicit comparisons with the results of Ren & Vasconcelos (2019).

The alunite crystals analysed by Ren & Vasconcelos (2019) are different to the alunite crystals from Martabe, so there is no reason to expect that the results obtained by these authors should mimic our own, unless of course the material they analysed had similarly complicated histories of growth and/or dissolution and regrowth. But one reviewer requested is that we need to justify why our analyses produced complex spectra, while citing Ren & Vasconcelos (2019) as having conducted analyses that produced 'plateaux'. However, it turns out that this is NOT the case. Ren & Vasconcelos (2019) also are spectra with multiple plateau segments, and the detail in the lamp-heating increments allows recognition that these are exactly the morphologies being discussed in the present manuscript as the result of multiple periods of growth. However, in the examples published by Ren & Vasconcelos (2019), the first plateau segment demonstrates release from a younger-grown alunite generation that had lesser retentivity. In the examples published from Martabe the reverse may apply.

In addition, more clarification was sought as to the differences in diffusion parameters, and the implications these differences would have in terms of computed closure temperatures. So we begin with that point, and then work through the comparison of our results with those produced by Ren & Vasconcelos (2019).

In doing this we discovered some minor miscalculations. Dodson's closure temperature formulae require recursion. We use the MacArgon program to calculate the result to an accuracy < 0.1°C. Given the parameters listed, hypogene alunite spheres with the diffusion parameters listed would exhibit closure temperatures in the range 234-287°C. Such retentivity would have allowed alunite to retain growth ages. Lesser retentivity would result in age spectra that exhibited typical diffusional loss profiles.

We can test this by examining the morphology of age spectra measured from the natural environment.

In doing this check we discovered two important facts:

i)      Ren & Vasconcelos (2019) claimed that results of their geochronology using laser heating) are indistinguishable from results of incremental heating using a heat lamp. This is not correct. There are significant differences in the detail of the morphology of the individual spectra which are invariably poorly approximated by the laser analyses. The "mini-plateaux" were obscured by the laser analysis.

ii)     Ren & Vasconcelos (2019) documented (but did not discuss) "mini-plateaux" in their spectra. My conclusion as the result of this comparison is therefore that Ren & Vasconcelos (2019) would have benefitted from application of the method of asymptotes and limits, using the methods detailed here.

**Technical Comments (from reviewer two)**

*The abstract reports activation energies in units of kJ/mol, but in figures units are given in kcal/mol. Please be consistent throughout the paper.*
Noted for correction. Values in both units will be provided.

*Section 2 appears to be missing; there is a section 1 (the introduction), and section 3 (where experimental and analytical methods are described). Please renumber sections.*
Noted for correction.

*Regarding the term "York Plot": three-isotope correlation diagrams are in wide use in isotope geochemistry. With regards to $^{40}Ar/^{39}Ar$ work, diagrams of $^{40}Ar/^{36}Ar$ vs. $^{39}Ar/^{36}Ar$ were first used by Craig Merrihue and Grenville Turner (1966); later, Turner introduced a plot of $^{36}Ar/^{40}Ar$ vs. $^{39}Ar/^{40}Ar$ (1971). These later became commonly known as "normal" and "inverse" isochron plots, respectively. Besides developing a robust linear regression method that accounts for errors in both axes and error correlations, I'm not sure how involved Derek York was in developing or using plots of $^{36}Ar/^{40}Ar$ vs. $^{39}Ar/^{40}Ar$. If you prefer not to use the term "inverse isochron" because you feel it carries too many interpretive connotations, perhaps falling back to the term "three-isotope correlation diagram" would be more appropriate and generic? Or perhaps call it a "Turner Plot"? Yes, the York regression is well known and commonly used; but the diagrams themselves should perhaps be considered separately from the regression methods applied in those data spaces. This is mostly a matter of personal preference; in the end, just be clear in defining the term that you use (preferably define it early and explicitly).*
The reviewer is thanked for this erudite discussion. Indeed, we do not refer to these diagrams as "*inverse isochron*" plots as this terminology could mislead a user as to their purpose. We find the major utility of such plots to be in facilitating the recognition of mixing lines. Although an inverse isochron can be defined, this should be done only in the case that an argument can be made that one should expect a clump of steps with the same age. Otherwise, such alignments can be spurious, *e.g.*, the result of asymptotic convergence as the result of mixing, in which case the definition of an 'inverse isochron' would lead to fallacy. From a historical perspective perhaps the solution is as the reviewer suggests, that we call each such diagram a "*Three-isotope inverse $^{40}Ar$ correlation*" plot. As these diagrams are used for York regressions, they are York Plots.

*Figure 11e: The x-axis label appears to be malformed; should this read "Percentage $^{39}Ar$ released" like the other figures?*
Noted for correction.

*Line 65: Suggestion for usage of future tense; you state that your "… study will build on [the results of Sutupo et al., 2013]…" (my emphasis in italics). Consider saying "builds" or "has built"; the latter is perhaps most correct, as you have already done the work, and are reporting on it. Future tense is least appropriate as you are no longer planning to do the work.*
Noted for correction.

*Line 140: Consider a more active voice, e.g., "Using the [method of] 'asymptotes and limits' we were able to extract relevant age information…"*
Noted for correction.

*Line 150: The reference "Forster et al., 2019" appears to be a duplicate. When looking in the references section, I found both "Forster, M., Koudashev, O., Nie, R., Yeung, S., and Lister, G.: 40Ar/39Ar thermochronology in the Ios basement terrane resolves the tectonic significance of the South Cyclades Shear Zone. Geological Society, London, Special Publications, 487, 291-313, https://doi.org/10.1144/SP487-2018-169, 2019" and "Forster, M., Koudashev, O., Nie, R., Yeung, S., Lister, G.: 40Ar/39Ar thermochronology in the Ios basement terrane resolves the tectonic significance of the South Cyclades Shear Zone. Geological Society, London, Special Publications 487, 291-313, https://doi.org/10.1144/SP487-2018-169, 2020." The publication date appears to differ when following the DOI link and selecting the Lyell Collection (2019) or GeoScienceWorld (2020). I recommend choosing one or the other, but do not include both; probably the earlier publication date would be more appropriate.*
Noted for correction.

*Line 229: Duplicate wording: "… towards a towards a…"*
Noted for correction.

*Lines 230–231: Suggestion for wording/punctuation: change "… because of mixing with a KCl/CaCl2 gas reservoir derived from inclusions (?)." to "… potentially due to mixing with a KCl/CaCl2 gas reservoir derived from inclusions." The parenthetical question mark seems to indicate this is a potential interpretation, but also makes it seem like you are unsure of the statement rather than proposing a possible cause. Also, on lines 234–235 you reinforce this interpretation based on your observation of higher Ca/K and Cl/K ratios, so the question mark is a bit confusing.*
We are unsure as to the nature of the reservoir. Noted for correction.

*Line 266: Consider use of active voice; instead of "To demonstrate the significance of the ages obtained, it was necessary to evaluate…" consider writing "To demonstrate the significance of the ages obtained, we evaluated…"*
Noted for correction.

*Line 283: Probable misspelling/wrong word; "paraments" should likely be "parameters".*
Noted for correction

*Lines 289–293: Possible grammatical improvements for clarity (**in bold**): "Simply put, **the** line used to estimate diffusion parameters must divide the sequence of steps in the Arrhenius plot by rank order. Points of a higher rank order will lie on or to the left of the line and points with a lower rank order with lie on or to the right of the line. The FAP and the need to well populate the Arrhenius plots particularly in the lower temperature of theexperiment is absolutely critical in gaining closure temperature data and in allowing modelling.".*
Noted for correction.

*Lines 464–465: "… two orders of magnitude of more…" should likely be "… two orders of magnitude or more…"*
Noted for correction.

Additional figures:
1. Spectra typical of mixing [page 2]
2. Percentage $^{39}$Ar plotted against temperature, for the first 200°C of the step-heating schedule [page 4]
3. Isotope ratios plotted against the blank [page 6]
4. Comparison of diffusion parameters with Ren & Vasconcelos (2019) [page 12]
5. Extended Arrhenius plot showing upward curvature at higher temperatures [page 12]
6. Map of large-scale faults around the Martabe deposit [page 12]
7. Multiple faulting episodes due to cross-structures formed during differential slab roll-back and the variation in the associated stress states [page 12]
8. Extended York Plot detailing mixing with the inclusion gas reservoir [page 17]

**Supplementary information for analysed alunites from Martabe epithermal gold deposit**

**Methods and procedures for $^{40}$Ar/$^{39}$Ar analysis**

**Sample irradiation details**

Irradiation of samples for $^{40}$Ar/$^{39}$Ar analysis is required, this was undertaken at the University of California Davis McClellan Nuclear Research Centre, CA, US. The samples in this study were irradiated in March 22-2018 as ANU CAN #30 batch.

The calculated amounts of grains were weighed, recorded and wrapped in labelled aluminium packets in preparation for irradiation. The sample filled foils were placed into a quartz irradiation canister together with aliquots of the flux monitor GA1550. The GA1550 standards are dispersed throughout the irradiated cannister, between the unknown age samples. In addition, packets containing $K_2SO_4$ and $CaF_2$ were placed in the middle of the canister to monitor $^{40}$Ar production from Potassium.

Irradiated samples were unwrapped upon their return to the Australian National University, and then rewrapped in tin foils in preparation for analysis in the mass spectrometer, this is done because the tin will melt in furnace and pump away the gases prior to the sample analysis.

**$^{40}$Ar/$^{39}$Ar procedures and analysis information**

Samples and standards were analysed in the Argon Laboratory at the Research School of Earth Science, The Australian National University, Canberra, Australia using a *Thermo Fisher* ARGUS-VI multi-collector mass spectrometer.

An calibrated temperature-controlled furnace was used for the step-heating technique to extract Argon isotopes from the samples to ensure 100% release of $^{39}$Ar. The furnace was degassed 4 times at 1,450°C for 15 minutes and the gas pumped away prior to the loading of the subsequent samples, while the flux monitors crystals (GA1550 biotite) were fused using a $CO_2$ continuous-wave laser. Gas released from either the flux monitors or each step of the sample's analyses, was exposed to three different Zr-Al getters for 10 minutes to remove active gases and the purified extracted gasses were isotopically analysed in the Argus VI mass spectrometer. Samples were analysed within 32-35 steps with temperatures of the overall schedule rising from 450° to 1,450°C (see excel data tables with step-heating schedules). The $^{40}$Ar/$^{39}$Ar dating technique is adapted from McDougall and Harrison (1999) and described in Forster and Lister (2009).

The background and furnace blank levels were measured and subtracted from all analysis. The nuclear interfering values for the correction factors for the isotopes are listed below. These are measured for the reactions and uncertainties of $(^{36}Ar/^{37}Ar)_{Ca}$, $(^{39}Ar/^{37}Ar)_{Ca}$, $(^{40}Ar/^{39}Ar)_K$, $(^{38}Ar/^{39}Ar)_K$ and $(^{38}Ar)_{Cl}/(^{39}Ar)_K$, and were calculated prior to sample analysis (see Tetley et al 1980).

Sample separation and preparation details have been described in the paper. Thin sections were not made due to the small overall sample size and overall unconsolidated nature and quality of these samples. XRD was done on grains chosen for analysis to verify their mineralogy and purity, which showed samples to be 99% pure. All grains were 250-420 µm in size with sample size ranging between 100 mg and 150 mg. No acids were used to clean samples, even though this may get rid of contamination, it can cause microstructural changes. It was decided to alternatively analyse each experiment where the step-heating procedures were detailed enough so as to identify contamination.

**ANU IRRADIATION CAN #30**

Flux Monitor:  GA1550 @ 98.5 ± 0.8 Ma (Spell and McDougall 2003)

| | |
|---|---|
| $(^{36}Ar/^{37}Ar)_{Ca}$ correction factor | 3.2858E-04 |
| $(^{39}Ar/^{37}Ar)_{Ca}$ correction factor | 7.9252E-04 |
| $(^{40}Ar/^{39}Ar)_{K}$  correction factor | 3.3453E-02 |
| $^{38}Ar/^{39}Ar)_{K}$  correction factor | 1.1716E-02 |
| $(^{38}Ar)_{Cl}/(^{39}Ar)_{K}$ correction factor | 8.1145E-02 |
| Ca/K conversion factor | 1.90 |
| Mass Discrimination factor | 0.98830 ± 0.103% |
| Lambda $^{40}K$ | 5.5430E-10 |
| Total irradiation power | 6.00 MW |
| Irradiation Date | March 22, 2018 |
| Irradiation shielding | Cadmium 1.0mm |

| Sample Name | Foil | J-Factor | J-Factor uncertainty | Mineral | Measurement Date |
|---|---|---|---|---|---|
| Purnama P-01 | A01 | 1.19537E-03 | 0.2420 | Alunite | 16-Jul-2018 |
| D3011643 | A02 | 1.19536E-03 | 0.2420 | Alunite | 19-Jul-2018 |
| D3150595 | A03 | 1.19534E-03 | 0.2420 | Alunite | 21-Jul-2018 |
| D3112423 | A04 | 1.19532E-03 | 0.2420 | Alunite | 23-Jul-2018 |
| D3078029 | A05 | 1.19531E-03 | 0.2420 | Alunite | 25-Jul-2018 |
| D3056884 | A06 | 1.19530E-03 | 0.2420 | Alunite | 27-Jul-2018 |
| D3067305 | A07 | 1.19528E-03 | 0.2420 | Alunite | 29-Jul-2018 |
| D3137821 | A08 | 1.19526E-03 | 0.2420 | Alunite | 01-Aug-2018 |
| D3035222 | A09 | 1.19525E-03 | 0.2420 | Alunite | 03-Aug-2018 |
| D3049860 | A10 | 1.19524E-03 | 0.2420 | Alunite | 06-Aug-2018 |

$^{40}K$ abundances and decay constants are calculated using method from Steiger and Jager (1977). Stated precisions for $^{40}Ar/^{39}Ar$ ages include all uncertainties in the measurement of isotope ratios and are quoted at the one sigma level and include errors in the J-factor. The reported data are corrected for system backgrounds, furnace blank level, mass discrimination, Neutron flux gradients and atmospheric contamination. GA1550 standards were analysed, and a linear best fit was then used for the calculation of the J-factor and J-factor uncertainty.

Data reductions were done with an adapted version of *Noble* Software (2016, developed and adapted by the Australian National University Argon Laboratory). The data reduction was based on optimising MSWD (the mean square of weighted deviates) of isotopes intensities with an exponential best fit methodology. *Noble* is documented in McDougall and Harrison (1999), the manual being available on the ANU RSES web site. *eArgon Software* has been used as a graphic analysis program which has only minimal mathematical additions.

The discrimination factor was calculated by analysing five Air Shots analysis on either side of sample analysis (see Excel Data Tables), based on the atmospheric $^{40}Ar/^{36}Ar$ ratio (295.5), and the calculation of the 1amu was used for the discrimination factor.

$^{40}Ar/^{39}Ar$ isotopic data of the sample is supplied in the Excel tables of the step-heat data, Tables A01-A10, which include details on: the heating schedule, Argon isotopes abundances and uncertainty levels, %Ar*, $^{40}Ar*/^{39}Ar(K)$, Cumulative $^{39}Ar\%$, Age and uncertainty, Ca/K, Cl/K, J-factor and J-factor uncertainty, noting that the fractional uncertainties are shown as %, and are stated in the headings of the appropriate columns. Uncertainty levels of the calculated ages are at one sigma. Blank level results and Air Shot measurements are available in the Data Table Excel Sheet.

**References**

Forster, M.A. and Lister, G.S. 2009. Core-complex-related extension of the Aegean lithosphere initiated at the Eocene-Oligocene transition. *Journal Geophysical Research*, **114**, B02401.

McDougall, I., & Harrison, T.M. (Eds.). 1999. Geochronology and Thermochronology by the $^{40}Ar/^{39}Ar$ Method, 2nd ed., 269 pp. Oxford Univ. Press, New York.

Spell, T. L., & I. McDougall. 2003. Characterization and calibration of $^{40}Ar/^{39}Ar$ dating standards. Chemical Geology, **198**, 189–211.

Steiger, R. H., & E. Jager. 1977. Subcommission on geochronology: Convention on the use of decay constants in geo- and cosmochronology. Earth Planetary Science Letters, **36**, 359–362.

Tetley, N., McDougall, I. & Heydegger, H. R. 1980. Thermal neutron interferences in the $^{40}Ar/^{39}Ar$ dating technique. *Journal Geophysical Research*, **85**, 7201–7205.

**36Ar abundances in a representative sample and in the blank in the furnace**

The plot below shows the measured difference in the abundances of $^{36}$Ar between a representative sample (A1) and the blanks analysed from the furnace. The furnace blanks have been subtracted from final isotope abundances prior to plotting of the data, this plot shows the furnace blanks are relatively low once significant release from the sample commences. The York Plots show that the initial release of gas approximates the atmospheric $^{36}$Ar/$^{40}$Ar ratio.

[Figure]

*The overall release pattern $^{36}$Ar during the release of the $^{39}$Ar from % 0.0 to % 100.0 in sample A1. LHS Y-Axis shows the intensity of the $^{36}$Ar abundances in each measured step of the sample (blue) and that of the furnace blank (red). The RHS Y-Axis shows the ratio percentage of the $^{36}$Ar in the blank to the $^{36}$Ar in the sample (green).*

**³⁹Ar abundances in a representative sample and the blank in the furnace**

Our ⁴⁰Ar/³⁹Ar experiments start at 450°C and finish at 1450°C. The ³⁹Ar release pattern plotted below shows that there are no Ar isotope abundances of any significances below approximately 650°C (sample PurnamaP-01 being the only exception with a slightly higher abundance with ~9% release at ~660°C). All nine other samples have less than 2% ³⁹Ar release up to 650°C. However, the experiments were started at 450°C so as to be sure of this characteristic.

[Figure]

*Plot shows the ³⁹Ar release during the experiment starting at 450°C to ~650°C, being the first 200°C of the experiment. Each of the samples is colour coded as in the legend. Temperature °C is on the x-axis and the % release of ³⁹Ar is on the y-axis.*

**J-Factor Regression**

ANU CAN#30 was irradiated at the University of California Davis McClellan Nuclear Research Centre, CA, US. The fluence monitor was GA1550 Biotite. The plot below shows the results for the calculation of the J-Factor regression for this irradiation batch that included these alunite samples.

[Figure]

*Plot show the J-Factor region and the sample position in the irradiation canister. Each point represents the average of the measured J-Factor for the GA1550 fluence monitor in that particular spot of the canister. The error bar on each individual point represents the average uncertainty of several measurements. The Purple line is the best fit regression for the J-Factor. The black dashed lines represent the standard deviation of the J-Factor. The blue dashed lines represent the deviation of J-Factor including their error bars.*

**Checklist for Data Reporting as set out by Schaen et al 2020**

**Minimum Required Data**

☑ Report uncertainties for all parameters (e.g., 95% confidence interval, 1σ, 2σ)

☑ Explicitly stated whether uncertainties on ages include decay constant uncertainties

☑ Report sample identifier (ideally unique, e.g., International Geo Sample Number [IGSN])

☑ Report sample location (e.g., latitude, longitude, elevation)

☑ Report sample lithology

☑ Specify material analyzed specified (e.g., single vs. multi-crystal aliquot, weight, phase type)

☑ Report relative isotope abundances[†] for $^{40}Ar$, $^{39}Ar$, $^{38}Ar$, $^{37}Ar$, and $^{36}Ar$

☑ Describe step heating schedule and/or laser power/wattage per analysis

☑ Identify reactor and port used for irradiation (and if Cd shielding or rotation was used)

☑ Describe fluence monitor details (e.g., name, age assumed, reference, $J$ value)

☑ Report decay constants used (e.g., $^{40}K$, $^{39}Ar$, $^{37}Ar$, $^{36}Cl$), references cited

☑ Identify interfering isotope production ratios (e.g., Ar produced from K, Ca, Cl), references cited

☑ Report ratios used for trapped[§] argon correction ($^{40}Ar/^{36}Ar$, $^{40}Ar/^{38}Ar$), reference cited

☑ Indicate time interval used in decay corrections (e.g., days from end of irradiation to start of analysis)

☑ Report proportion radiogenic $^{40}Ar$ (%$^{40}Ar$*)

☑ Provide model age and unit of each analysis (e.g., yr, ka, Ma, Ga)

☑ List $F$ value ($^{40}Ar$*/$^{39}Ar_K$)

☑ Distinguish which steps are included in the age spectrum/isochron

☑ Report statistics to evaluate robustness of data (e.g., MSWD, p-value)

☑ Publish data tables in tabular (e.g., CSV, XLS) or machine-readable (e.g., JSON/XML) file format

**Recommended Data**

☑ Describe sample treatment (e.g., mineral separation techniques, acid treatment used)

☑ Identify data reduction software used (e.g., Mass Spec, ArArCALC, PyChron, in-house)

☑ List grain size of material analyzed

☑ Report representative blank measurements

☑ Report frequency of blank/air/cocktail measurements
* * *
[†] Corrected for baseline, background, mass discrimination and/or detector intercalibration, reactor interferences, and radioactive decay

[§] For terrestrial samples, this is commonly the composition of atmospheric argon

*Note: that the uncertainties for the Ca/K and Cl/K are not reported here.*

9.

---

## Author Response (AR2)

**Direct dating of overprinting fluid systems in the Martabe epithermal gold deposit using highly retentive alunite**

Jack Muston[1], Marnie Forster[2], Davood Vasegh[2], Conrad Alderton[3], Shawn Crispin[4], Gordon Lister[5]

[1] AngloGold Ashanti, Perth, 2601 Australia

[2] Argon Geochronology and Structural Geology, Research School of Earth Sciences, Australian National University, Canberra, 2601 Australia

[3] C3 Metals, Toronto, Canada

[4] Eurasian Resources Group, Dubai

[5] W.H. Bryan Mining and Geology Research Centre, Sustainable Minerals Institute, The University of Queensland, Brisbane 4068, Australia

*Correspondence to*: Jack Muston <jemuston@gmail.com> and Gordon Lister <g.lister@uq.edu.au>

Dear Klaus

Please accept my apologies, for the tracking option was not turned on and I did not notice. However, changes made fall into the category of improving English usage, avoiding emotive language, etc. as well as directly addressing the requested corrections.

All the best
Gordon

Corrections are as noted [including on the PDFs of individual figures, as required].

1) with the writing style of the main text. It reads like someone is telling a story to a live audience in a casual environment. It lacks scientific rigor and clarity.
THIS IS NO LONGER THE CASE

2) The paper needs some rewriting to make it more scientific. Words like: interestingly, clearly, yet, obviously, we remark, we see, in our view, one can be sure, we must conclude, we doubt are too emotional/personal for a scientific text and should be eliminated to present an objective scientific story rather than a subjective evaluation.
THIS HAS BEEN ADDRESSED

3) It is not acceptable in the paper to respond to reviewer comments. (e.g. 222, 425, 538..)
THIS HAS BEEN ADDRESSED

4) Place figure captions below figures, not above and below at random
DONE

5) Decide if you want to use "e" or "E" for exponential expressions and keep it consistent throughout the main text, also in the Supplement
DONE

6) Report data properly. Error usually two significant digits. Data: enough and not more digits so the error is in the last or two last digits reported.
WE THINK THE DATA IS ALREADY REPORTED CORRECTLY, BASED ON THE PRECISION OF THE MEASUREMENT (BUT PLEASE LET US KNOW IF YOU DIFFER)

7) Use the chemical symbol for the element if not specified. Use name only at the beginning of a sentence and when you mean the pure element (e.g. argon gas, rhenium filament).
DONE EXCEPT WHEN WOULD INTERRUPT THE FLOW OF THE SENTENCE

8) Kelvin (K) does not have a degree symbol (°), only Celsius or Centigrade do.
CORRECTED

9) Every map needs a scale. Not everybody knows that 1°latitude is ca. 111.1 km
DONE

10) 113 K content: do you mean mol% or wt%, K or K2O
DONE

11) 144: unwanted gases: very imprecise statement, what are they?
DONE

12) 221: this is very strange text for a scientific paper. Hear-say should not be part of a scientific paper. Try to remain factual and make sure all statements are backed by accessible data and publications. Referring to a comment from a review is not appropriate.
REMOVED

13) 406: attitudes are not scientific criteria. Delete sentence
DONE

14) 425: one reviewer cited Ren and425 Vasconcelos (2019): this statement cannot be part of the paper. It is only suitable in the response to a review that is directed to the editor.
REMOVED

15) Fig 16: asympotes: should be asymptotes
CORRECTED – THANKS – WELL SPOTTED

16) Fig 18 released not release ?
CORRECTED

17) Fig 19: random use of capital letters and lower case for the same word
CORRECTED

18) 791: very poor writing style. You can talk like this, but not write a scientific text this way
CORRECTED

19) 704: very arbitrary statement. Doubts are not a robust statement in a scientific text
REMOVED

---

## Author Response (AR3)

Author's response

The reference list has been updated according to the journal instructions.